# Mixed-curvature Variational Autoencoders

**Ondrej Skopek[1], Octavian-Eugen Ganea[1,2] & Gary Bécigneul[1,2]**
[1] Department of Computer Science, ETH Zürich
[2] Computer Science and Artificial Intelligence Laboratory, Massachusetts Institute of Technology
`oskopek@oskopek.com, oct@mit.edu, gary.becigneul@inf.ethz.ch`

## Abstract

Euclidean geometry has historically been the typical "workhorse" for machine learning applications due to its power and simplicity. However, it has recently been shown that geometric spaces with constant non-zero curvature improve representations and performance on a variety of data types and downstream tasks. Consequently, generative models like Variational Autoencoders (VAEs) have been successfully generalized to elliptical and hyperbolic latent spaces. While these approaches work well on data with particular kinds of biases e.g. tree-like data for a hyperbolic VAE, there exists no generic approach unifying and leveraging all three models. We develop a Mixed-curvature Variational Autoencoder, an efficient way to train a VAE whose latent space is a product of constant curvature Riemannian manifolds, where the per-component curvature is fixed or learnable. This generalizes the Euclidean VAE to curved latent spaces and recovers it when curvatures of all latent space components go to 0.

## 1 Introduction

Generative models, a growing area of unsupervised learning, aim to model the data distribution $p(\boldsymbol{x})$ over data points $\boldsymbol{x}$ from a space $\mathcal{X}$, which is usually a high-dimensional Euclidean space $\mathbb{R}^n$. This has desirable benefits like a naturally definable inner-product, vector addition, or a closed-form distance function. Yet, many types of data have a strongly non-Euclidean latent structure (Bronstein et al., 2017), e.g. the set of human-interpretable images. They are usually thought to live on a "natural image manifold" (Zhu et al., 2016), a continuous lower-dimensional subset of the space in which they are represented. By moving along the manifold, one can continuously change the content and appearance of interpretable images. As noted in Nickel & Kiela (2017), changing the geometry of the underlying latent space enables better representations of specific data types compared to Euclidean spaces of any dimensions, e.g. tree structures and scale-free networks.

Motivated by these observations, a range of recent methods learn representations in different spaces of constant curvatures: spherical or elliptical (Batmanghelich et al., 2016), hyperbolic (Nickel & Kiela, 2017; Sala et al., 2018; Tifrea et al., 2019) and even in products of these spaces (Gu et al., 2019; Bachmann et al., 2020). Using a combination of different constant curvature spaces, Gu et al. (2019) aim to match the underlying geometry of the data better. However, an open question remains: how to choose the dimensionality and curvatures of each of the partial spaces?

A popular approach to generative modeling is the Variational Autoencoder (Kingma & Welling, 2014). VAEs provide a way to sidestep the intractability of marginalizing a joint probability model of the input and latent space $p(\boldsymbol{x}, \boldsymbol{z})$, while allowing for a prior $p(\boldsymbol{z})$ on the latent space. Recently, variants of the VAE have been introduced for spherical (Davidson et al., 2018; Xu & Durrett, 2018) and hyperbolic (Mathieu et al., 2019; Nagano et al., 2019) latent spaces.

Our approach, the Mixed-curvature Variational Autoencoder, is a generalization of VAEs to products of constant curvature spaces[1]. It has the advantage of a better reduction in dimensionality, while maintaining efficient optimization. The resulting latent space is a "non-constantly" curved manifold that is more flexible than a single constant curvature manifold.

---

[1]Code is available on GitHub at `https://github.com/oskopek/mvae`.

Our contributions are the following: (i) we develop a principled framework for manipulating representations and modeling probability distributions in products of constant curvature spaces that smoothly transitions across curvatures of different signs, (ii) we generalize Variational Autoencoders to learn latent representations on products of constant curvature spaces with generalized Gaussian-like priors, and (iii) empirically, our models outperform current benchmarks on a synthetic tree dataset (Mathieu et al., 2019) and on image reconstruction on the MNIST (LeCun, 1998), Omniglot (Lake et al., 2015), and CIFAR (Krizhevsky, 2009) datasets for some latent space dimensions.

## 2 GEOMETRY AND PROBABILITY IN RIEMANNIAN MANIFOLDS

To define constantly curved spaces, we first need to define the notion of *sectional curvature* $K(\tau_{\boldsymbol{x}})$ of two linearly independent vectors in the tangent space at a point $\boldsymbol{x} \in \mathcal{M}$ spanning a two-dimensional plane $\tau_{\boldsymbol{x}}$ (Berger, 2012). Since we deal with constant curvature spaces where all the sectional curvatures are equal, we denote a manifold's curvature as $K$. Instead of curvature $K$, we sometimes use the generalized notion of a *radius*: $R = 1/\sqrt{|K|}$.

There are three different types of manifolds $\mathcal{M}$ we can define with respect to the sign of the curvature: a positively curved space, a "flat" space, and a negatively curved space. Common realizations of those manifolds are the hypersphere $\mathbb{S}_K$, the Euclidean space $\mathbb{E}$, and the hyperboloid $\mathbb{H}_K$:

$$\mathcal{M} = \begin{cases} \mathbb{S}_K^n = \{\boldsymbol{x} \in \mathbb{R}^{n+1} : \langle \boldsymbol{x}, \boldsymbol{x} \rangle_2 = 1/K\}, & \text{for } K > 0 \\ \mathbb{E}^n = \mathbb{R}^n, & \text{for } K = 0 \\ \mathbb{H}_K^n = \{\boldsymbol{x} \in \mathbb{R}^{n+1} : \langle \boldsymbol{x}, \boldsymbol{x} \rangle_{\mathcal{L}} = 1/K\}, & \text{for } K < 0 \end{cases}$$

where $\langle \cdot, \cdot \rangle_2$ is the standard Euclidean inner product, and $\langle \cdot, \cdot \rangle_{\mathcal{L}}$ is the Lorentz inner product,

$$\langle \boldsymbol{x}, \boldsymbol{y} \rangle_{\mathcal{L}} = -x_1 y_1 + \sum_{i=2}^{n+1} x_i y_i \quad \forall \boldsymbol{x}, \boldsymbol{y} \in \mathbb{R}^{n+1}.$$

We will need to define the exponential map, logarithmic map, and parallel transport in all spaces we consider. The exponential map in Euclidean space is defined as $\exp_{\boldsymbol{x}}(\boldsymbol{v}) = \boldsymbol{x} + \boldsymbol{v}$, for all $\boldsymbol{x} \in \mathbb{E}^n$ and $\boldsymbol{v} \in \mathcal{T}_{\boldsymbol{x}} \mathbb{E}^n$. Its inverse, the logarithmic map is $\log_{\boldsymbol{x}}(\boldsymbol{y}) = \boldsymbol{y} - \boldsymbol{x}$, for all $\boldsymbol{x}, \boldsymbol{y} \in \mathbb{E}^n$. Parallel transport in Euclidean space is simply an identity $\mathrm{PT}_{\boldsymbol{x} \to \boldsymbol{y}}(\boldsymbol{v}) = \boldsymbol{v}$, for all $\boldsymbol{x}, \boldsymbol{y} \in \mathbb{E}^n$ and $\boldsymbol{v} \in \mathcal{T}_{\boldsymbol{x}} \mathbb{E}^n$. An overview of these operations in the hyperboloid $\mathbb{H}_K^n$ and the hypersphere $\mathbb{S}_K^n$ can be found in Table 1. For more details, refer to Petersen et al. (2006), Cannon et al. (1997), or Appendix A.

### 2.1 STEREOGRAPHICALLY PROJECTED SPACES

The above spaces are enough to cover any possible value of the curvature, and they define all the necessary operations we will need to train VAEs in them. However, both the hypersphere and the hyperboloid have an unsuitable property, namely the non-convergence of the norm of points as the curvature goes to 0. Both spaces grow as $K \to 0$ and become locally "flatter", but to do that, their points have to go away from the origin of the coordinate space $\boldsymbol{0}$ to be able to satisfy the manifold's definition. A good example of a point that diverges is the origin of the hyperboloid (or a pole of the hypersphere) $\boldsymbol{\mu}_0 = (1/\sqrt{|K|}, 0, \ldots, 0)^T$. In general, we can see that $\|\boldsymbol{\mu}_0\|^2 = 1/K \xrightarrow{K \to 0} \pm\infty$. Additionally, the distance and the metric tensors of these spaces do not converge to their Euclidean variants as $K \to 0$, hence the spaces themselves do not converge to $\mathbb{R}^d$. This makes both of these spaces unsuitable for trying to learn sign-agnostic curvatures.

Luckily, there exist well-defined non-Euclidean spaces that inherit most properties from the hyperboloid and the hypersphere, yet do not have these properties – namely, the Poincaré ball and the projected sphere, respectively. We obtain them by applying stereographic conformal projections, meaning that angles are preserved by this transformation. Since the distance function on the hyperboloid and hypersphere only depend on the radius and angles between points, they are isometric.

We first need to define the projection function $\rho_K$. For a point $(\xi; \boldsymbol{x}^T)^T \in \mathbb{R}^{n+1}$ and curvature $K \in \mathbb{R}$, where $\xi \in \mathbb{R}, \boldsymbol{x}, \boldsymbol{y} \in \mathbb{R}^n$

$$\rho_K((\xi; \boldsymbol{x}^T)^T) = \frac{\boldsymbol{x}}{1 + \sqrt{|K|}\xi}, \qquad \rho_K^{-1}(\boldsymbol{y}) = \left( \frac{1}{\sqrt{|K|}} \frac{1 - K \|\boldsymbol{y}\|_2^2}{1 + K \|\boldsymbol{y}\|_2^2}; \frac{2\boldsymbol{y}^T}{1 + K \|\boldsymbol{y}\|_2^2} \right)^T.$$

The formulas correspond to the classical stereographic projections defined for these models (Lee, 1997). Note that both of these projections map the point $\boldsymbol{\mu}_0 = (1/\sqrt{|K|}, 0, \ldots, 0)$ in the original space to $\boldsymbol{\mu}_0 = \mathbf{0}$ in the projected space, and back.

Since the stereographic projection is conformal, the metric tensors of both spaces will be conformal. In this case, the metric tensors of both spaces are the same, except for the sign of $K$: $\mathfrak{g}_{\boldsymbol{x}}^{\mathbb{D}_K} = \mathfrak{g}_{\boldsymbol{x}}^{\mathbb{P}_K} = (\lambda_{\boldsymbol{x}}^K)^2 \mathfrak{g}^{\mathbb{E}}$, for all $\boldsymbol{x}$ in the respective manifold (Ganea et al., 2018a), and $\mathfrak{g}_{\boldsymbol{y}}^{\mathbb{E}} = \boldsymbol{I}$ for all $\boldsymbol{y} \in \mathbb{E}$. The conformal factor $\lambda_{\boldsymbol{x}}^K$ is then defined as $\lambda_{\boldsymbol{x}}^K = 2/(1 + K \|\boldsymbol{x}\|_2^2)$. Among other things, this form of the metric tensor has the consequence that we unfortunately cannot define a single unified inner product in all tangent spaces at all points. The inner product at $\boldsymbol{x} \in \mathcal{M}$ has the form of $\langle \boldsymbol{u}, \boldsymbol{v} \rangle_{\boldsymbol{x}} = (\lambda_{\boldsymbol{x}}^K)^2 \langle \boldsymbol{u}, \boldsymbol{v} \rangle_2$ for all $\boldsymbol{u}, \boldsymbol{v} \in \mathcal{T}_{\boldsymbol{x}} \mathcal{M}$.

We can now define the two models corresponding to $K > 0$ and $K < 0$. The curvature of the projected manifold is the same as the original manifold. An $n$-dimensional *projected hypersphere* ($K > 0$) is defined as the set $\mathbb{D}_K^n = \rho_K(\mathbb{S}_K^n \setminus \{-\boldsymbol{\mu}_0\}) = \mathbb{R}^n$, where $\boldsymbol{\mu}_0 = (1/\sqrt{|K|}, 0, \ldots, 0)^T \in \mathbb{S}_K^n$, along with the induced distance function. The $n$-dimensional *Poincaré ball* $\mathbb{P}_K^n$ (also called the Poincaré disk when $n = 2$) for a given curvature $K < 0$ is defined as $\mathbb{P}_K^n = \rho_K(\mathbb{H}_K^n) = \left\{ \boldsymbol{x} \in \mathbb{R}^n : \langle \boldsymbol{x}, \boldsymbol{x} \rangle_2 < -\frac{1}{K} \right\}$, with the induced distance function.

## 2.2 GYROVECTOR SPACES

An important analogy to vector spaces (vector addition and scalar multiplication) in non-Euclidean geometry is the notion of gyrovector spaces (Ungar, 2008). Both of the above spaces $\mathbb{D}_K$ and $\mathbb{P}_K$ (jointly denoted as $\mathcal{M}_K$) share the same structure, hence they also share the following definition of addition. The *Möbius addition* $\oplus_K$ of $\boldsymbol{x}, \boldsymbol{y} \in \mathcal{M}_K$ (for both signs of $K$) is defined as

$$\boldsymbol{x} \oplus_K \boldsymbol{y} = \frac{(1 - 2K \langle \boldsymbol{x}, \boldsymbol{y} \rangle_2 - K \|\boldsymbol{y}\|_2^2)\boldsymbol{x} + (1 + K \|\boldsymbol{x}\|_2^2)\boldsymbol{y}}{1 - 2K \langle \boldsymbol{x}, \boldsymbol{y} \rangle_2 + K^2 \|\boldsymbol{x}\|_2^2 \|\boldsymbol{y}\|_2^2}.$$

We can, therefore, define "gyrospace distances" for both of the above spaces, which are alternative curvature-aware distance functions

$$d_{\mathbb{D}_{\mathrm{gyr}}}(\boldsymbol{x}, \boldsymbol{y}) = \frac{2}{\sqrt{K}} \tan^{-1}(\sqrt{K} \|-\boldsymbol{x} \oplus_K \boldsymbol{y}\|_2), \quad d_{\mathbb{P}_{\mathrm{gyr}}}(\boldsymbol{x}, \boldsymbol{y}) = \frac{2}{\sqrt{-K}} \tanh^{-1}(\sqrt{-K} \|-\boldsymbol{x} \oplus_K \boldsymbol{y}\|_2).$$

These two distances are equivalent to their non-gyrospace variants $d_{\mathcal{M}}(\boldsymbol{x}, \boldsymbol{y}) = d_{\mathcal{M}_{\mathrm{gyr}}}(\boldsymbol{x}, \boldsymbol{y})$, as is shown in Theorem A.4 and its hypersphere equivalent. Additionally, Theorem A.5 shows that

$$d_{\mathcal{M}_{\mathrm{gyr}}}(\boldsymbol{x}, \boldsymbol{y}) \xrightarrow{K \to 0} 2 \|\boldsymbol{x} - \boldsymbol{y}\|_2,$$

which means that the distance functions converge to the Euclidean distance function as $K \to 0$.

We can notice that most statements and operations in constant curvature spaces have a dual statement or operation in the corresponding space with the opposite curvature sign. The notion of duality is one which comes up very often and in our case is based on Euler's formula $e^{ix} = \cos(x) + i\sin(x)$ and the notion of principal square roots $\sqrt{-K} = i\sqrt{K}$. This provides a connection between trigonometric, hyperbolic trigonometric, and exponential functions. Thus, we can convert all the hyperbolic formulas above to their spherical equivalents and vice-versa.

Since Ganea et al. (2018a) and Tifrea et al. (2019) used the same gyrovector spaces to define an exponential map, its inverse logarithmic map, and parallel transport in the Poincaré ball, we can reuse them for the projected hypersphere by applying the transformations above, as they share the same formalism. For parallel transport, we additionally need the notion of gyration (Ungar, 2008)

$$\mathrm{gyr}[\boldsymbol{x}, \boldsymbol{y}]\boldsymbol{v} = \ominus_K (\boldsymbol{x} \oplus_K \boldsymbol{y}) \oplus_K (\boldsymbol{x} \oplus_K (\boldsymbol{y} \oplus_K \boldsymbol{v})).$$

Parallel transport in the both the projected hypersphere and the Poincaré ball then is $\mathrm{PT}_{\boldsymbol{x} \to \boldsymbol{y}}^K(\boldsymbol{v}) = (\lambda_{\boldsymbol{x}}^K / \lambda_{\boldsymbol{y}}^K)\mathrm{gyr}[\boldsymbol{y}, -\boldsymbol{x}]\boldsymbol{v}$, for all $\boldsymbol{x}, \boldsymbol{y} \in \mathcal{M}_K^n$ and $\boldsymbol{v} \in \mathcal{T}_{\boldsymbol{x}} \mathcal{M}_K^n$. Using a curvature-aware definition scalar products ($\langle \boldsymbol{x}, \boldsymbol{y} \rangle_K = \langle \boldsymbol{x}, \boldsymbol{y} \rangle_2$ if $K \geq 0$, $\langle \boldsymbol{x}, \boldsymbol{y} \rangle_{\mathcal{L}}$ if $K < 0$) and of trigonometric functions

$$\sin_K = \begin{cases} \sin & \text{if } K > 0 \\ \sinh & \text{if } K < 0 \end{cases} \qquad \cos_K = \begin{cases} \cos & \text{if } K > 0 \\ \cosh & \text{if } K < 0 \end{cases} \qquad \tan_K = \begin{cases} \tan & \text{if } K > 0 \\ \tanh & \text{if } K < 0 \end{cases}$$

we can summarize all the necessary operations in all manifolds compactly in Table 1 and Table 2.

Table 1: Summary of operations in $\mathbb{S}_K$ and $\mathbb{H}_K$.

| | |
|---|---|
| Distance | $d(\boldsymbol{x}, \boldsymbol{y}) = \dfrac{1}{\sqrt{|K|}} \cos_K^{-1}(|K| \langle \boldsymbol{x}, \boldsymbol{y} \rangle_K)$ |
| Exponential map | $\exp_{\boldsymbol{x}}^K(\boldsymbol{v}) = \cos_K\left(\sqrt{|K|}\,\|\boldsymbol{v}\|_K\right) \boldsymbol{x} + \sin_K\left(\sqrt{|K|}\,\|\boldsymbol{v}\|_K\right) \dfrac{\boldsymbol{v}}{\sqrt{|K|}\,\|\boldsymbol{v}\|_K}$ |
| Logarithmic map | $\log_{\boldsymbol{x}}^K(\boldsymbol{y}) = \dfrac{\cos_K^{-1}(K \langle \boldsymbol{x}, \boldsymbol{y} \rangle_K)}{\sin_K\left(\cos_K^{-1}(K \langle \boldsymbol{x}, \boldsymbol{y} \rangle_K)\right)} (\boldsymbol{y} - K \langle \boldsymbol{x}, \boldsymbol{y} \rangle_K \boldsymbol{x})$ |
| Parallel transport | $\mathrm{PT}_{\boldsymbol{x} \to \boldsymbol{y}}^K(\boldsymbol{v}) = \boldsymbol{v} - \dfrac{K \langle \boldsymbol{y}, \boldsymbol{v} \rangle_K}{1 + K \langle \boldsymbol{x}, \boldsymbol{y} \rangle_K} (\boldsymbol{x} + \boldsymbol{y})$ |

Table 2: Summary of operations in projected spaces $\mathbb{D}_K$ and $\mathbb{P}_K$.

| | |
|---|---|
| Distance | $d(\boldsymbol{x}, \boldsymbol{y}) = \dfrac{1}{\sqrt{|K|}} \cos_K^{-1}\left(1 - \dfrac{2K \|\boldsymbol{x} - \boldsymbol{y}\|_2^2}{(1 + K \|\boldsymbol{x}\|_2^2)(1 + K \|\boldsymbol{y}\|_2^2)}\right)$ |
| Gyrospace distance | $d_{\mathrm{gyr}}(\boldsymbol{x}, \boldsymbol{y}) = \dfrac{2}{\sqrt{|K|}} \tan_K^{-1}(\sqrt{|K|}\,\|-\boldsymbol{x} \oplus_K \boldsymbol{y}\|_2)$ |
| Exponential map | $\exp_{\boldsymbol{x}}^K(\boldsymbol{v}) = \boldsymbol{x} \oplus_K \left( \tan_K\left(\sqrt{|K|}\,\dfrac{\lambda_{\boldsymbol{x}}^K \|\boldsymbol{v}\|_2}{2}\right) \dfrac{\boldsymbol{v}}{\sqrt{|K|}\,\|\boldsymbol{v}\|_2}\right)$ |
| Logarithmic map | $\log_{\boldsymbol{x}}^K(\boldsymbol{y}) = \dfrac{2}{\sqrt{|K|}\lambda_{\boldsymbol{x}}^K} \tan_K^{-1}\left(\sqrt{|K|}\,\|-\boldsymbol{x} \oplus_K \boldsymbol{y}\|_2\right) \dfrac{-\boldsymbol{x} \oplus_K \boldsymbol{y}}{\|-\boldsymbol{x} \oplus_K \boldsymbol{y}\|_2}$ |
| Parallel transport | $\mathrm{PT}_{\boldsymbol{x} \to \boldsymbol{y}}^K(\boldsymbol{v}) = \dfrac{\lambda_{\boldsymbol{x}}^K}{\lambda_{\boldsymbol{y}}^K} \mathrm{gyr}[\boldsymbol{y}, -\boldsymbol{x}] \boldsymbol{v}$ |

## 2.3 PRODUCTS OF SPACES

Previously, our space consisted of only one manifold of varying dimensionality and fixed curvature. Like Gu et al. (2019), we propose learning latent representations in products of constant curvature spaces, contrary to existing VAE approaches which are limited to a single Riemannian manifold.

Our latent space $\mathcal{M}'$ consists of several *component spaces* $\mathcal{M}' = \bigtimes_{i=1}^k \mathcal{M}_{K_i}^{n_i}$, where $n_i$ is the dimensionality of the space, $K_i$ is its curvature, and $\mathcal{M} \in \{\mathbb{E}, \mathbb{S}, \mathbb{D}, \mathbb{H}, \mathbb{P}\}$ is the model choice. Even though all components have constant curvature, the resulting manifold $\mathcal{M}'$ has non-constant curvature. Its distance function decomposes based on its definition $d_{\mathcal{M}}^2(\boldsymbol{x}, \boldsymbol{y}) = \sum_{i=1}^k d_{\mathcal{M}_{K_i}^{n_i}}^2\left(\boldsymbol{x}^{(i)}, \boldsymbol{y}^{(i)}\right)$,

where $\boldsymbol{x}^{(i)}$ represents a vector in $\mathcal{M}_{K_i}^{n_i}$, corresponding to the part of the latent space representation of $\boldsymbol{x}$ belonging to $\mathcal{M}_{K_i}^{n_i}$. All other operations we defined on our manifolds are element-wise. Therefore, we again decompose the representations into parts $\boldsymbol{x}^{(i)}$, apply the operation on that part $\tilde{\boldsymbol{x}}^{(i)} = f_{K_i}^{(n_i)}(\boldsymbol{x}^{(i)})$ and concatenate the resulting parts back $\tilde{\boldsymbol{x}} = \bigodot_{i=1}^k \tilde{\boldsymbol{x}}^{(i)}$.

The *signature* of the product space, i.e. its parametrization, has several degrees of freedom per component: (i) the model $\mathcal{M}$, (ii) the dimensionality $n_i$, and (iii) the curvature $K_i$. We need to select all of the above for every component in our product space. To simplify, we use a shorthand notation for repeated components: $(\mathcal{M}_{K_i}^{n_i})^j = \bigtimes_{l=1}^j \mathcal{M}_{K_i}^{n_i}$. In Euclidean spaces, the notation is redundant. For $n_1, \ldots, n_k \in \mathbb{Z}$, such that $\sum_{i=1}^k n_i = n \in \mathbb{Z}$, it holds that the Cartesian product of Euclidean spaces $\mathbb{E}^{n_i}$ is $\mathbb{E}^n = \bigtimes_{i=1}^k \mathbb{E}^{n_i}$. However, the equality does *not* hold for the other considered manifolds. This is due to the additional constraints posed on the points in the definitions of individual models of curved spaces.

### 2.4 PROBABILITY DISTRIBUTIONS ON RIEMANNIAN MANIFOLDS

To be able to train Variational Autoencoders, we need to chose a probability distribution $p$ as a prior and a corresponding posterior distribution family $q$. Both of these distributions have to be differentiable with respect to their parametrization, they need to have a differentiable Kullback-Leiber (KL) divergence, and be "reparametrizable" (Kingma & Welling, 2014). For distributions where the KL does not have a closed-form solution independent on $z$, or where this integral is too hard to compute, we can estimate it using Monte Carlo estimation

$$D_{\mathrm{KL}}\left(q \,\|\, p\right) \approx \frac{1}{L} \sum_{l=1}^{L} \log\left(\frac{q(z^{(l)})}{p(z^{(l)})}\right) \stackrel{\text{if } L=1}{=} \log\left(\frac{q(z^{(1)})}{p(z^{(1)})}\right),$$

where $z^{(l)} \sim q$ for all $l = 1, \ldots, L$. The Euclidean VAE uses a natural choice for a prior on its latent representations – the Gaussian distribution (Kingma & Welling, 2014). Apart from satisfying the requirements for a VAE prior and posterior distribution, the Gaussian distribution has additional properties, like being the maximum entropy distribution for a given variance (Jaynes, 1957). There exist several fundamentally different approaches to generalizing the Normal distribution to Riemannian manifolds. We discuss the following three generalizations based on the way they are constructed (Mathieu et al., 2019).

**Wrapping**  This approach leverages the fact that all manifolds define a tangent vector space at every point. We simply sample from a Gaussian distribution in the tangent space at $\mu_0$ with mean $\mathbf{0}$, and use parallel transport and the exponential map to map the sampled point onto the manifold. The PDF can be obtained using the multivariate chain rule if we can compute the determinant of the Jacobian of the parallel transport and the exponential map. This is very computationally effective at the expense of losing some theoretical properties.

**Restriction**  The "Restricted Normal" approach is conceptually antagonal – instead of expanding a point to a dimensionally larger point, we restrict a point of the ambient space sampled from a Gaussian to the manifold. The consequence is that the distributions constructed this way are based on the "flat" Euclidean distance. An example of this is the von Mises-Fisher (vMF) distribution (Davidson et al., 2018). A downside of this approach is that vMF only has a single scalar covariance parameter $\kappa$, while other approaches can parametrize covariance in different dimensions separately.

**Maximizing entropy**  Assuming a known mean and covariance matrix, we want to maximize the entropy of the distribution (Pennec, 2006). This approach is usually called the Riemannian Normal distribution. Mathieu et al. (2019) derive it for the Poincaré ball, and Hauberg (2018) derive the Spherical Normal distribution on the hypersphere. Maximum entropy distributions resemble the Gaussian distribution's properties the closest, but it is usually very hard to sample from such distributions, compute their normalization constants, and even derive the specific form. Since the gains for VAE performance using this construction of Normal distributions over wrapping is only marginal, as reported by Mathieu et al. (2019), we have chosen to focus on Wrapped Normal distributions.

To summarize, Wrapped Normal distributions are very computationally efficient to sample from and also efficient for computing the log probability of a sample, as detailed by Nagano et al. (2019). The Riemannian Normal distributions (based on geodesic distance in the manifold directly) could also be used, however they are more computationally expensive for sampling, because the only methods available are based on rejection sampling (Mathieu et al., 2019).

#### 2.4.1 WRAPPED NORMAL DISTRIBUTION

First of all, we need to define an "origin" point on the manifold, which we will denote as $\mu_0 \in \mathcal{M}_K$. What this point corresponds to is manifold-specific: in the hyperboloid and hypersphere it corresponds to the point $\mu_0 = (1/\sqrt{|K|}, 0, \ldots, 0)^T$, and in the Poincaré ball, projected sphere, and Euclidean space it is simply $\mu_0 = \mathbf{0}$, the origin of the coordinate system.

Sampling from the distribution $\mathcal{WN}(\mu, \Sigma)$ has been described in detail in Nagano et al. (2019) and Mathieu et al. (2019), and we have extended all the necessary operations and procedures to arbitrary curvature $K$. Sampling then corresponds to:

$$v \sim \mathcal{N}(\mu_0, \Sigma) \in \mathcal{T}_{\mu_0}\mathcal{M}_K, \ u = \mathrm{PT}_{\mu_0 \to \mu}^K(v) \in \mathcal{T}_\mu \mathcal{M}_K, \ z = \exp{\mu_x^K}(u) \in \mathcal{M}_K.$$

The log-probability of samples can be computed by the reverse procedure:

$$\boldsymbol{u} = \log_{\boldsymbol{\mu}}^K(\boldsymbol{z}) \in \mathcal{T}_{\boldsymbol{\mu}}\mathcal{M}_K, \ \boldsymbol{v} = \mathrm{PT}_{\boldsymbol{\mu} \to \boldsymbol{\mu}_0}^K(\boldsymbol{u}) \in \mathcal{T}_{\boldsymbol{\mu}_0}\mathcal{M}_K,$$

$$\log \mathcal{WN}(\boldsymbol{\mu}, \boldsymbol{\Sigma}) = \log \mathcal{N}(\boldsymbol{v}; \boldsymbol{\mu}_0, \boldsymbol{\Sigma}) - \log \det \left( \frac{\partial f}{\partial \boldsymbol{v}} \right),$$

where $f = \exp_{\boldsymbol{\mu}}^K \circ \mathrm{PT}_{\boldsymbol{\mu}_0 \to \boldsymbol{\mu}}^K$. The distribution can be applied to all manifolds that we have introduced. The only differences are the specific forms of operations and the log-determinant in the PDF. The specific forms of the log-PDF for the four spaces $\mathbb{H}$, $\mathbb{S}$, $\mathbb{D}$, and $\mathbb{P}$ are derived in Appendix B. All variants of this distribution are reparametrizable, differentiable, and the KL can be computed using Monte Carlo estimation. As a consequence of the distance function and operations convergence theorems for the Poincaré ball A.5 (analogously for the projected hypersphere), A.17, A.18, and A.20, we see that the Wrapped Normal distribution converges to the Gaussian distribution as $K \to 0$.

## 3 VARIATIONAL AUTOENCODERS IN PRODUCTS SPACES

To be able to learn latent representations in Riemannian manifolds instead of in $\mathbb{R}^d$ as above, we only need to change the parametrization of the mean and covariance in the VAE forward pass, and the choice of prior and posterior distributions. The prior and posterior have to be chosen depending on the chosen manifold and are essentially treated as hyperparameters of our VAE. Since we have defined the Wrapped Normal family of distributions for all spaces, we can use $\mathcal{WN}(\boldsymbol{\mu}_0, \boldsymbol{\sigma}^2 \boldsymbol{I})$ as the posterior family, and $\mathcal{WN}(\boldsymbol{\mu}_0, \boldsymbol{I})$ as the prior distribution. The forms of the distributions depend on the chosen space type. The mean is parametrized using the exponential map $\exp_{\boldsymbol{\mu}_0}^K$ as an activation function. Hence, all the model's parameters live in Euclidean space and can be optimized directly.

In experiments, we sometimes use $\mathrm{vMF}(\boldsymbol{\mu}, \kappa)$ for the hypersphere $\mathbb{S}_K^n$ (or a backprojected variant of vMF for $\mathbb{D}_K^n$) with the associated hyperspherical uniform distribution $U(\mathbb{S}_K^n)$ as a prior (Davidson et al., 2018), or the Riemannian Normal distribution $\mathcal{RN}(\boldsymbol{\mu}, \sigma^2)$ and the associated prior $\mathcal{RN}\boldsymbol{\mu}_0, 1$ for the Poincare ball $\mathbb{P}_K^n$ (Mathieu et al., 2019).

### 3.1 LEARNING CURVATURE

We have already seen approaches to learning VAEs in products of spaces of constant curvature. However, we can also change the curvature constant in each of the spaces during training. The individual spaces will still have constant curvature at each point, we just allow changing the constant in between training steps. To differentiate between these training procedures, we will call them *fixed* curvature and *learnable* curvature VAEs respectively.

The motivation behind changing curvature of non-Euclidean constant curvature spaces might not be clear, since it is apparent from the definition of the distance function in the hypersphere and hyperboloid $d(\boldsymbol{x}, \boldsymbol{y}) = R \cdot \theta_{\boldsymbol{x}, \boldsymbol{y}}$, that the distances between two points that stay at the same angle only get rescaled when changing the radius of the space. Same applies for the Poincaré ball and the projected spherical space. However, the decoder does not only depend on pairwise distances, but rather on the specific positions of points in the space. It can be conjectured that the KL term of the ELBO indeed is only "rescaled" when we change the curvature, however, the reconstruction process is influenced in non-trivial ways. Since that is hard to quantify and prove, we devise a series of practical experiments to show overall model performance is enhanced when learning curvature.

**Fixed curvature VAEs** In *fixed* curvature VAEs, all component latent spaces have a fixed curvature that is selected a priori and fixed for the whole duration of the training procedure, as well as during evaluation. For Euclidean components it is 0, for positively or negatively curved spaces any positive or negative number can be chosen, respectively. For stability reasons, we select curvature values from the range $[0.25, 1.0]$, which corresponds to radii in $[1.0, 2.0]$. The exact curvature value does not have a significant impact on performance when training a fixed curvature VAE, as motivated by the distance rescaling remark above. In the following, we refer to fixed curvature components with a constant subscript, e.g. $\mathbb{H}_1^n$.

**Learnable curvature VAEs**   In all our manifolds, we can differentiate the ELBO with respect to the curvature $K$. This enables us to treat $K$ as a parameter of the model and learn it using gradient-based optimization, exactly like we learn the encoder/decoder maps in a VAE.

Learning curvature directly is badly conditioned – we are trying to learn one scalar parameter that influences the resulting decoder and hence the ELBO quite heavily. Empirically, we have found that Stochastic Gradient Descent works well to optimize the radius of a component. We constrain the radius to be strictly positive in all non-Euclidean spaces by applying a ReLU activation function before we use it in operations.

**Universal curvature VAEs**   However, we must still a priori select the "partitioning" of our latent space – the number of components and for each of them select the dimension and at least the sign of the curvature of that component (signature estimation).

The simplest approach would be to just try all possibilities and compare the results on a specific dataset. This procedure would most likely be optimal, but does not scale well.

To eliminate this, we propose an approximate method – we partition our space into 2-dimensional components (if the number of dimensions is odd, one component will have 3 dimensions). We initialize all of them as Euclidean components and train for half the number of maximal epochs we are allowed. Then, we split the components into 3 approximately equal-sized groups and make one group into hyperbolic components, one into spherical, and the last remains Euclidean. We do this by changing the curvature of a component by a very small $\epsilon$. We then train just the encoder/decoder maps for a few epochs to stabilize the representations after changing the curvatures. Finally, we allow learning the curvatures of all non-Euclidean components and train for the rest of the allowed epochs. The method is not completely general, as it never uses components bigger than dimension 2, but the approximation has empirically performed satisfactorily.

We also do not constrain the curvature of the components to a specific sign in the last stage of training. Therefore, components may change their type of space from a positively curved to a negatively curved one, or vice-versa.

Because of the divergence of points as $K \to 0$ for the hyperboloid and hypersphere, the universal curvature VAE assumes the positively curved space is $\mathbb{D}$ and the negatively curved space is $\mathbb{P}$. In all experiments, this universal approach is denoted as $\mathbb{U}^n$.

## 4   EXPERIMENTS

For our experiments, we use four datasets: (i) Branching diffusion process (Mathieu et al., 2019, BDP) – a synthetic tree-like dataset with injected noise, (ii) Dynamically-binarized MNIST digits (LeCun, 1998) – we binarize the images similarly to Burda et al. (2016); Salakhutdinov & Murray (2008): the training set is binarized dynamically (uniformly sampled threshold per-sample $\mathrm{bin}(\boldsymbol{x}) \in \{0,1\}^D = \boldsymbol{x} > \mathcal{U}[0,1], \boldsymbol{x} \in \boldsymbol{x} \subseteq [0,1]^D$), and the evaluation set is done with a fixed binarization ($\boldsymbol{x} > 0.5$), (iii) Dynamically-binarized Omniglot characters (Lake et al., 2015) downsampled to $28 \times 28$ pixels, and (iv) CIFAR-10 (Krizhevsky, 2009).

All models in all datasets are trained with early stopping on training ELBO with a lookahead of 50 epochs and a warmup of 100 epochs (Bowman et al., 2016). All BDP models are trained for a 1000 epochs, MNIST and Omniglot models are trained for 300 epochs, and CIFAR for 200 epochs. We compare models with a given latent space dimension using marginal log-likelihood with importance sampling (Burda et al., 2016) with 500 samples, except for CIFAR, which uses 50 due to memory constraints. In all tables, we denote it as LL. We run all experiments at least 3 times to get an estimate of variance when using different initial values.

In all the BDP, MNIST, and Omniglot experiments below, we use a simple feed-forward encoder and decoder architecture consisting of a single dense layer with 400 neurons and element-wise ReLU activation. Since all the VAE parameters $\{\theta, \phi\}$ live in Euclidean manifolds, we can use standard gradient-based optimization methods. Specifically, we use the Adam (Kingma & Ba, 2015) optimizer with a learning rate of $10^{-3}$ and standard settings for $\beta_1 = 0.9, \beta_2 = 0.999$, and $\epsilon = 10^{-8}$.

For the CIFAR encoder map, we use a simple convolutional neural networks with three convolutional layers with 64, 128, and 512 channels respectively. For the decoder map, we first use a dense layer

Table 3: Summary of results (mean and standard deviation), latent space dimension 6, spherical covariance, on the BDP (left) and MNIST (right) datasets.

| Model | LL | Model | LL |
|---|---|---|---|
| $\mathbb{S}_1^6$ | $-55.81_{\pm 0.35}$ | $\mathbb{S}_1^6$ | $-96.71_{\pm 0.17}$ |
| $\mathbb{D}_1^6$ | $-55.78_{\pm 0.07}$ | vMF $\mathbb{S}_1^6$ | $-97.03_{\pm 0.14}$ |
| $\mathbb{E}^6$ | $-56.28_{\pm 0.56}$ | $\mathbb{D}_1^6$ | $-98.21_{\pm 0.23}$ |
| $(\mathbb{H}_{-1}^2)^3$ | $-56.08_{\pm 0.52}$ | $\mathbb{E}^6$ | $-97.16_{\pm 0.15}$ |
| $\mathbb{H}_{-1}^6$ | $-56.18_{\pm 0.32}$ | $\mathbb{H}_{-1}^6$ | $-97.10_{\pm 0.44}$ |
| $(\mathbb{P}_{-1}^2)^3$ | $-55.98_{\pm 0.62}$ | $(\mathbb{P}_{-1}^2)^3$ | $-97.56_{\pm 0.04}$ |
| $\mathbb{P}_{-1}^6$ | $-56.74_{\pm 0.55}$ | $(\mathcal{RN}\ \mathbb{P}_{-1}^2)^3$ | $\mathbf{-92.54}_{\pm 0.19}$ |
| $(\mathcal{RN}\ \mathbb{P}_{-1}^2)^3$ | $\mathbf{-54.99}_{\pm 0.12}$ | | |
| $(\mathbb{S}^2)^3$ | $-56.05_{\pm 0.21}$ | $(\mathbb{S}^2)^3$ | $-96.46_{\pm 0.12}$ |
| $(\mathbb{D}^2)^3$ | $-56.06_{\pm 0.36}$ | $\mathbb{S}^6$ | $-96.72_{\pm 0.15}$ |
| $(\mathbb{H}^2)^3$ | $\mathbf{-55.80}_{\pm 0.32}$ | vMF $\mathbb{S}^6$ | $-96.72_{\pm 0.18}$ |
| $(\mathbb{P}^2)^3$ | $-56.29_{\pm 0.05}$ | $\mathbb{D}^6$ | $-97.72_{\pm 0.15}$ |
| $(\mathcal{RN}\ \mathbb{P}^2)^3$ | $-56.25_{\pm 0.56}$ | $(\mathbb{H}^2)^3$ | $-97.37_{\pm 0.13}$ |
| | | $(\mathcal{RN}\ \mathbb{P}^2)^3$ | $\mathbf{-94.16}_{\pm 0.68}$ |
| $\mathbb{D}^2 \times \mathbb{E}^2 \times \mathbb{P}^2$ | $-55.87_{\pm 0.22}$ | $\mathbb{D}^2 \times \mathbb{E}^2 \times \mathbb{P}^2$ | $-97.48_{\pm 0.18}$ |
| $\mathbb{E}^2 \times \mathbb{H}^2 \times \mathbb{S}^2$ | $-55.92_{\pm 0.42}$ | $\mathbb{D}^2 \times \mathbb{E}^2 \times (\mathcal{RN}\ \mathbb{P}^2)$ | $-96.43_{\pm 0.47}$ |
| $\mathbb{E}^2 \times \mathbb{H}^2 \times (\text{vMF}\ \mathbb{S}^2)$ | $-55.82_{\pm 0.43}$ | $\mathbb{D}_1^2 \times \mathbb{E}^2 \times (\mathcal{RN}\ \mathbb{P}_{-1}^2)$ | $\mathbf{-96.18}_{\pm 0.21}$ |
| $\mathbb{E}^2 \times \mathbb{H}_{-1}^2 \times (\text{vMF}\ \mathbb{S}_1^2)$ | $\mathbf{-55.77}_{\pm 0.51}$ | $\mathbb{E}^2 \times \mathbb{H}^2 \times \mathbb{S}^2$ | $-96.80_{\pm 0.20}$ |
| $(\mathbb{U}^2)^3$ | $\mathbf{-55.56}_{\pm 0.15}$ | $\mathbb{E}^2 \times \mathbb{H}_{-1}^2 \times \mathbb{S}_1^2$ | $-96.76_{\pm 0.09}$ |
| $\mathbb{U}^6$ | $-55.84_{\pm 0.38}$ | $\mathbb{E}^2 \times \mathbb{H}^2 \times (\text{vMF}\ \mathbb{S}^2)$ | $-96.56_{\pm 0.27}$ |
| | | $(\mathbb{U}^2)^3$ | $\mathbf{-97.12}_{\pm 0.04}$ |
| | | $\mathbb{U}^6$ | $-97.26_{\pm 0.16}$ |

of dimension 2048, and then three consecutive transposed convolutional layers with 256, 64, and 3 channels. All layers are followed by a ReLU activation function, except for the last one. All convolutions have $4 \times 4$ kernels with stride 2, and padding of size 1.

The first 10 epochs for all models are trained with a fixed curvature starting at 0 and increasing in absolute value each epoch. This corresponds to a burn-in period similarly to Nickel & Kiela (2017). For learnable curvature approaches we then use Stochastic Gradient Descent with learning rate $10^{-4}$ and let the optimizers adjust the value freely, for fixed curvature approaches it stays at the last burn-in value. All our models use the Wrapped Normal distribution, or equivalently Gaussian in Euclidean components, unless specified otherwise. All fixed curvature components are denoted with a $\mathcal{M}_1$ or $\mathcal{M}_{-1}$ subscript, learnable curvature components do not have a subscript. The observation model for the reconstruction loss term were Bernoulli distributions for MNIST and Omniglot, and standard Gaussian distributions for BDP and CIFAR.

As baselines, we train VAEs with spaces that have a fixed constant curvature, i.e. assume a single Riemannian manifold (potentially a product of them) as their latent space. It is apparent that our models with a single component, like $\mathbb{S}_1^n$ correspond to Davidson et al. (2018) and Xu & Durrett (2018), $\mathbb{H}_{-1}^n$ is equivalent to the Hyperbolic VAE of Nagano et al. (2019), $\mathbb{P}_{-c}^n$ corresponds to the $\mathcal{P}^c$-VAE of Mathieu et al. (2019), and $\mathbb{E}^n$ is equivalent to the Euclidean VAE. In the following, we present a selection of all the obtained results. For more information see Appendix E. Bold numbers represent values that are particularly interesting. Since the Riemannian Normal and the von Mises-Fischer distribution only have a spherical covariance matrix, i.e. a single scalar variance parameter per component, we evaluate all our approaches with a spherical covariance parametrization as well.

**Binary diffusion process** For the BDP dataset and latent dimension 6 (Table 3), we observe that all VAEs that only use the von Mises-Fischer distribution perform worse than a Wrapped Normal. However, when a VMF spherical component was paired with other component types, it performed better than if a Wrapped Normal spherical component was used instead. Riemannian Normal VAEs did very well on their own – the fixed Poincaré VAE $(\mathcal{RN}\ \mathbb{P}_{-1}^2)^3$ obtains the best score. It did not fare as well when we tried to learn curvature with it, however.

Table 4: Summary of selected models (mean and standard deviation), latent space dimension 6 (left) and 12 (right), diagonal covariance, on the MNIST dataset.

| Model | LL | Model | LL |
|---|---|---|---|
| $\mathbb{S}_1^6$ | $-\mathbf{96.51}_{\pm 0.09}$ | $(\mathbb{S}_1^2)^6$ | $-79.92_{\pm 0.21}$ |
| $\mathbb{D}_1^6$ | $-97.89_{\pm 0.10}$ | $(\mathbb{D}_1^2)^6$ | $-80.53_{\pm 0.10}$ |
| $\mathbb{E}^6$ | $-96.88_{\pm 0.16}$ | $\mathbb{E}^{12}$ | $-\mathbf{79.51}_{\pm 0.09}$ |
| $\mathbb{H}_{-1}^6$ | $-97.38_{\pm 0.73}$ | $(\mathbb{H}_{-1}^2)^6$ | $-80.54_{\pm 0.23}$ |
| $\mathbb{P}_{-1}^6$ | $-97.33_{\pm 0.15}$ | $\mathbb{H}_{-1}^{12}$ | $\textbf{-79.37}_{\pm 0.14}$ |
| | | $(\mathbb{P}_{-1}^2)^6$ | $-80.39_{\pm 0.07}$ |
| $\mathbb{S}^6$ | $-\mathbf{96.44}_{\pm 0.20}$ | $\mathbb{S}^{12}$ | $-79.99_{\pm 0.27}$ |
| $\mathbb{D}^6$ | $-97.53_{\pm 0.22}$ | $\mathbb{D}^{12}$ | $-80.37_{\pm 0.16}$ |
| $(\mathbb{H}^2)^3$ | $-\mathbf{96.86}_{\pm 0.31}$ | $\mathbb{H}^{12}$ | $-79.77_{\pm 0.10}$ |
| $\mathbb{H}^6$ | $-96.90_{\pm 0.26}$ | $(\mathbb{P}^2)^6$ | $-80.31_{\pm 0.08}$ |
| $\mathbb{P}^6$ | $-97.26_{\pm 0.16}$ | | |
| $\mathbb{D}^2 \times \mathbb{E}^2 \times \mathbb{P}^2$ | $-97.37_{\pm 0.14}$ | $(\mathbb{D}_1^2)^2 \times (\mathbb{E}^2)^2 \times (\mathbb{P}_{-1}^2)^2$ | $-80.14_{\pm 0.11}$ |
| $\mathbb{D}_1^2 \times \mathbb{E}^2 \times \mathbb{P}_{-1}^2$ | $-97.29_{\pm 0.16}$ | $\mathbb{D}_1^4 \times \mathbb{E}^4 \times \mathbb{P}_{-1}^4$ | $-80.14_{\pm 0.20}$ |
| $\mathbb{E}^2 \times \mathbb{H}^2 \times \mathbb{S}^2$ | $-\mathbf{96.71}_{\pm 0.19}$ | $(\mathbb{E}^2)^2 \times (\mathbb{H}^2)^2 \times (\mathbb{S}^2)^2$ | $-\mathbf{79.59}_{\pm 0.25}$ |
| $\mathbb{E}^2 \times \mathbb{H}_{-1}^2 \times \mathbb{S}_1^2$ | $-\mathbf{96.66}_{\pm 0.27}$ | $\mathbb{E}^4 \times \mathbb{H}^4 \times \mathbb{S}^4$ | $-\mathbf{79.69}_{\pm 0.14}$ |
| $(\mathbb{U}^2)^3$ | $-97.06_{\pm 0.13}$ | $(\mathbb{U}^2)^6$ | $-\mathbf{79.61}_{\pm 0.06}$ |
| $\mathbb{U}^6$ | $-\mathbf{96.90}_{\pm 0.10}$ | $\mathbb{U}^{12}$ | $-80.01_{\pm 0.30}$ |

An interesting observation is that all single-component VAEs $\mathcal{M}^6$ performed worse than product VAEs $(\mathcal{M}^2)^3$ when curvature was learned, across all component types. Our universal curvature VAE $(\mathbb{U}^2)^3$ managed to get better results than all other approaches except for the Riemannian Normal baseline, but it is within the margin of error of some other models. It also outperformed its single-component variant $\mathbb{U}^6$. However, we did not find that it converged to specific curvature values, only that they were in the approximate range of $(-0.1, +0.1)$.

**Dynamically-binarized MNIST reconstruction** On MNIST (Table 3) with spherical covariance, we noticed that VMF again under-performed Wrapped Normal, except when it was part of a product like $\mathbb{E}^2 \times \mathbb{H}^2 \times (\text{vMF } \mathbb{S}^2)$. When paired with another Euclidean and a Riemannian Normal Poincaré disk component, it performed well, but that might be because the $\mathcal{RN} \; \mathbb{P}_{-1}$ component achieved best results across the board on MNIST. It achieved good results even compared to diagonal covariance VAEs on 6-dimensional MNIST. Several approaches are better than the Euclidean baseline. That applies mainly to the above mentioned Riemannian Normal Poincaré ball components, but also $\mathbb{S}^6$ both with Wrapped Normal and VMF, as well as most product space VAEs with different curvatures (third section of the table). Our $(\mathbb{U}^2)^3$ performed similarly to the Euclidean baseline VAE.

With diagonal covariance parametrization (Table 4), we observe similar trends. With a latent dimension of 6, the Riemannian Normal Poincaré ball VAE is still the best performer. The Euclidean baseline VAE achieved better results than its spherical covariance counterpart. Overall, the best result is achieved by the single-component spherical model, with learnable curvature $\mathbb{S}_6$. Interestingly, all single-component VAEs performed better than their $(\mathcal{M}^2)^3$ counterparts, except for the $\mathbb{H}^6$ hyperboloid, but only by a tiny margin. Products of different component types also achieve good results. Noteworthy is that their fixed curvature variants seem to perform marginally better than learnable curvature ones. Our universal VAEs perform at around the Euclidean baseline VAE performance. Interestingly, all of them end up with negative curvatures $-0.3 < K < 0$.

Secondly, we run our models with a latent space dimension of 12. We immediately notice, that not many models can beat the Euclidean VAE baseline $\mathbb{E}^{12}$ consistently, but several are within the margin of error. Notably, the product VAEs of $\mathbb{H}$, $\mathbb{S}$, and $\mathbb{E}$, fixed and learnable $\mathbb{H}^{12}$, and our universal VAE $(\mathbb{U}^2)^6$. Interestingly, products of small components perform better when curvature is fixed, whereas single big component VAEs are better when curvature is learned.

**Dynamically-binarized Omniglot reconstruction** For a latent space of dimension 6 (Table 5), the best of the baseline models is the Poincaré VAE of (Mathieu et al., 2019). Our models that come

Table 5: Summary of selected models (mean and standard deviation), latent space dimension 6, diagonal covariance, on the Omniglot (left) and CIFAR-10 dataset (right), respectively.

| Model | LL | | Model | LL |
|---|---|---|---|---|
| $\mathbb{S}_1^6$ | $-136.69_{\pm 0.94}$ | | $\mathbb{E}^6$ | $-1896.19_{\pm 2.54}$ |
| $\mathbb{D}_1^6$ | $-137.42_{\pm 1.20}$ | | $\mathbb{H}_{-1}^6$ | $\mathbf{-1888.23}_{\pm 2.12}$ |
| $\mathbb{E}^6$ | $-136.05_{\pm 0.29}$ | | $\mathbb{P}_{-1}^6$ | $-1893.27_{\pm 0.61}$ |
| $\mathbb{H}_{-1}^6$ | $-137.09_{\pm 0.06}$ | | $\mathbb{D}^6$ | $-1893.85_{\pm 0.36}$ |
| $\mathbb{P}_{-1}^6$ | $\mathbf{-135.86}_{\pm 0.20}$ | | $\mathbb{S}^6$ | $\mathbf{-1889.76}_{\pm 1.62}$ |
| $(\mathbb{S}^2)^3$ | $\mathbf{-136.14}_{\pm 0.27}$ | | $\mathbb{P}^6$ | $-1891.40_{\pm 2.14}$ |
| $\mathbb{S}^6$ | $-136.20_{\pm 0.44}$ | | $\mathbb{D}^2 \times \mathbb{E}^2 \times \mathbb{P}^2$ | $-1899.90_{\pm 4.60}$ |
| $(\mathbb{D}^2)^3$ | $-136.13_{\pm 0.17}$ | | $\mathbb{E}^2 \times \mathbb{H}^2 \times \mathbb{S}^2$ | $-1895.46_{\pm 0.92}$ |
| $\mathbb{D}^6$ | $-136.30_{\pm 0.08}$ | | $(\mathbb{U}^2)^3$ | $-1895.09_{\pm 4.27}$ |
| $(\mathbb{H}^2)^3$ | $-136.17_{\pm 0.09}$ | | | |
| $\mathbb{H}^6$ | $-136.24_{\pm 0.32}$ | | | |
| $(\mathbb{P}^2)^3$ | $-136.09_{\pm 0.07}$ | | | |
| $\mathbb{P}^6$ | $-136.05_{\pm 0.44}$ | | | |
| $\mathbb{D}^2 \times \mathbb{E}^2 \times \mathbb{P}^2$ | $\mathbf{-135.89}_{\pm 0.40}$ | | | |
| $\mathbb{E}^2 \times \mathbb{H}^2 \times \mathbb{S}^2$ | $-135.93_{\pm 0.48}$ | | | |
| $(\mathbb{U}^2)^3$ | $-136.21_{\pm 0.07}$ | | | |
| $\mathbb{U}^6$ | $\mathbf{-136.04}_{\pm 0.17}$ | | | |

very close to the average estimated marginal log-likelihood, and are definitely within the margin of error, are mainly $(\mathbb{S}^2)^3$, $\mathbb{D}^2 \times \mathbb{E}^2 \times \mathbb{P}^2$, and $\mathbb{U}^6$. However, with the variance of performance across different runs, we cannot draw a clear conclusion. In general, hyperbolic VAEs seem to be doing a bit better on this dataset than spherical VAEs, which is also confirmed by the fact that almost all universal curvature models finished with negative curvature components.

**CIFAR-10 reconstruction**  For a latent space of dimension 6, we can observe that almost all non-Euclidean models perform better than the euclidean baseline $\mathbb{E}^6$. Especially well-performing is the fixed hyperboloid $\mathbb{H}_{-1}^6$, and the learnable hypersphere $\mathbb{S}^6$. Curvatures for all learnable models on this dataset converge to values in the approximate range of $(-0.15, +0.15)$.

**Summary**  In conclusion, a very good model seems to be the Riemannian Normal Poincaré ball VAE $\mathcal{RN}\ \mathbb{P}^n$. However, it has practical limitations due to a rejection sampling algorithm and an unstable implementation. On the contrary, von Mises-Fischer spherical VAEs have almost consistently performed worse than their Wrapped Normal equivalents. Overall, Wrapped Normal VAEs in all constant curvature manifolds seem to perform well at modeling the latent space.

A key takeaway is that our universal curvature models $\mathbb{U}^n$ and $(\mathbb{U}^2)^{\lfloor n/2 \rfloor}$ seem to generally outperform their corresponding Euclidean VAE baselines in lower-dimensional latent spaces and, with minor losses, manage to keep most of the competitive performance as the dimensionality goes up, contrary to VAEs with other non-Euclidean components.

## 5 CONCLUSION

By transforming the latent space and associated prior distributions onto Riemannian manifolds of constant curvature, it has previously been shown that we can learn representations on curved space. Generalizing on the above ideas, we have extended the theory of learning VAEs to products of constant curvature spaces. To do that, we have derived the necessary operations in several models of constant curvature spaces, extended existing probability distribution families to these manifolds, and generalized VAEs to latent spaces that are products of smaller "component" spaces, with learnable curvature. On various datasets, we show that our approach is competitive and additionally has the property that it generalizes the Euclidean variational autoencoder – if the curvatures of all components go to 0, we recover the VAE of Kingma & Welling (2014).

ACKNOWLEDGMENTS

We would like to thank Andreas Bloch for help in verifying some of the formulas for constant curvature spaces and for many insightful discussions; Prof. Thomas Hofmann and the Data Analytics Lab, the Leonhard cluster, and ETH Zürich for GPU access.

Work was done while all authors were at ETH Zürich.

Ondrej Skopek (`oskopek@google.com`) is now at Google.

Octavian-Eugen Ganea (`oct@mit.edu`) is now at the Computer Science and Artificial Intelligence Laboratory, Massachusetts Institute of Technology.

Gary Bécigneul (`gary.becigneul@inf.ethz.ch`) is funded by the Max Planck ETH Center for Learning Systems.

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

# A GEOMETRICAL DETAILS

## A.1 RIEMANNIAN GEOMETRY

An elementary notion in Riemannian geometry is that of a real, smooth *manifold* $\mathcal{M} \subseteq \mathbb{R}^n$, which is a collection of real vectors $\boldsymbol{x}$ that is locally similar to a linear space, and lives in the *ambient space* $\mathbb{R}^n$. At each point of the manifold $\boldsymbol{x} \in \mathcal{M}$ a real vector space of the same dimensionality as $\mathcal{M}$ is defined, called the *tangent space at point $\boldsymbol{x}$*: $\mathcal{T}_{\boldsymbol{x}}\mathcal{M}$. Intuitively, the tangent space contains all the directions and speeds at which one can pass through $\boldsymbol{x}$. Given a matrix representation $G(\boldsymbol{x}) \in \mathbb{R}^{n \times n}$ of the *Riemannian metric tensor* $\mathfrak{g}(\boldsymbol{x})$, we can define a *scalar product on the tangent space*: $\langle \cdot, \cdot \rangle_{\boldsymbol{x}} : \mathcal{T}_{\boldsymbol{x}}\mathcal{M} \times \mathcal{T}_{\boldsymbol{x}}\mathcal{M} \to \mathcal{M}$, where $\langle \boldsymbol{a}, \boldsymbol{b} \rangle_{\boldsymbol{x}} = \mathfrak{g}(\boldsymbol{x})(\boldsymbol{a}, \boldsymbol{b}) = \boldsymbol{a}^T G(\boldsymbol{x}) \boldsymbol{b}$ for any $\boldsymbol{a}, \boldsymbol{b} \in \mathcal{T}_{\boldsymbol{x}}\mathcal{M}$. A *Riemannian manifold* is then the tuple $(\mathcal{M}, \mathfrak{g})$. The scalar product induces a norm on the tangent space $\mathcal{T}_{\boldsymbol{x}}\mathcal{M}$: $||\boldsymbol{a}||_{\boldsymbol{x}} = \sqrt{\langle \boldsymbol{a}, \boldsymbol{a} \rangle_{\boldsymbol{x}}} \; \forall \boldsymbol{a} \in \mathcal{T}_{\boldsymbol{x}}\mathcal{M}$ (Petersen et al., 2006).

Although it seems like the manifold only defines a local geometry, it induces global quantities by integrating the local contributions. The metric tensor induces a local infinitesimal volume element on each tangent space $\mathcal{T}_{\boldsymbol{x}}\mathcal{M}$ and hence a measure is induced as well $d\mathcal{M}(\boldsymbol{x}) = \sqrt{|G(\boldsymbol{x})|}d\boldsymbol{x}$ where $d\boldsymbol{x}$ is the Lebesgue measure. The *length* of a curve $\gamma : t \mapsto \gamma(t) \in \mathcal{M}$, $t \in [0, 1]$ is given by $L(\gamma) = \int_0^1 \sqrt{\left\| \frac{d}{dt}\gamma(t) \right\|_{\gamma(t)}} dt$.

Straight lines are generalized to constant speed curves giving the shortest path between pairs of points $\boldsymbol{x}, \boldsymbol{y} \in \mathcal{M}$, so called *geodesics*, for which it holds that $\gamma^* = \arg\min_\gamma L(\gamma)$, such that $\gamma(0) = \boldsymbol{x}$, $\gamma(1) = \boldsymbol{y}$, and $\left\| \frac{d}{dt}\gamma(t) \right\|_{\gamma(t)} = 1$. Global distances are thus induced on $\mathcal{M}$ by $d_{\mathcal{M}}(\boldsymbol{x}, \boldsymbol{y}) = \inf_\gamma L(\gamma)$. Using this metric, we can go on to define a metric space $(\mathcal{M}, d_{\mathcal{M}})$. Moving from a point $\boldsymbol{x} \in \mathcal{M}$ in a given direction $\boldsymbol{v} \in \mathcal{T}_{\boldsymbol{x}}\mathcal{M}$ with constant velocity is formalized by the *exponential map*: $\exp_{\mathbf{x}} : \mathcal{T}_{\boldsymbol{x}}\mathcal{M} \to \mathcal{M}$. There exists a unique unit speed geodesic $\gamma$ such that $\gamma(0) = \boldsymbol{x}$ and $\frac{d\gamma(t)}{dt}\Big|_{t=0} = \boldsymbol{v}$, where $\boldsymbol{v} \in \mathcal{T}_{\boldsymbol{x}}\mathcal{M}$.

The corresponding exponential map is then defined as $\exp_{\mathbf{x}}(\boldsymbol{v}) = \gamma(1)$. The logarithmic map is the inverse $\log_{\boldsymbol{x}} = \exp_{\mathbf{x}}^{-1} : \mathcal{M} \to \mathcal{T}_{\boldsymbol{x}}\mathcal{M}$. For geodesically complete manifolds, i.e. manifolds in which there exists a length-minimizing geodesic between every $\boldsymbol{x}, \boldsymbol{y} \in \mathcal{M}$, such as the Lorentz model, hypersphere, and many others, $\exp_{\mathbf{x}}$ is well-defined on the full tangent space $\mathcal{T}_{\boldsymbol{x}}\mathcal{M}$.

To connect vectors in tangent spaces, we use parallel transport $\mathrm{PT}_{\boldsymbol{x} \to \boldsymbol{y}} : \mathcal{T}_{\boldsymbol{x}}\mathcal{M} \to \mathcal{T}_{\boldsymbol{y}}\mathcal{M}$, which is an isomorphism between the two tangent spaces, so that the transported vectors stay parallel to the connection. It corresponds to moving tangent vectors along geodesics and defines a canonical way to connect tangent spaces.

## A.2 BRIEF COMPARISON OF CONSTANT CURVATURE SPACE MODELS

We have seen five different models of constant curvature space, each of which has advantages and disadvantages when applied to learning latent representations in them using VAEs.

A big advantage of the hyperboloid and hypersphere is that optimization in the spaces does not suffer from as many numerical instabilities as it does in the respective projected spaces. On the other hand, we have seen that when $K \to 0$, the norms of points go to infinity. As we will see in experiments, this is not a problem when optimizing curvature within these spaces in practice, except if we're trying to cross the boundary at $K = 0$ and go from a hyperboloid to a sphere, or vice versa. Intuitively, the points are just positioned very differently in the ambient space of $\mathbb{H}_{-\epsilon}$ and $\mathbb{S}_\epsilon$, for a small $\epsilon > 0$.

Since points in the $n$-dimensional projected hypersphere and Poincaré ball models can be represented using a real vector of length $n$, it enables us to visualize points in these manifolds directly for $n = 2$ or even $n = 3$. On the other hand, optimizing a function over these models is not very well-conditioned. In the case of the Poincaré ball, a significant amount of points lie close to the boundary of the ball (i.e. with a squared norm of almost $1/K$), which causes numerical instabilities even when using 64-bit float precision in computations.

A similar problem occurs with the projected hypersphere with points that are far away from the origin $\mathbf{0}$ (i.e. points that are close to the "South pole" on the backprojected sphere). Unintuitively,

all points that are far away from the origin are actually very close to each other with respect to the induced distance function and very far away from each other in terms of Euclidean distance.

Both distance conversion theorems (A.5 and its projected hypersphere counterpart) rely on the points being *fixed* when changing curvature. If they are somehow dependent on curvature, the convergence theorem does not hold. We conjecture that if points stay close to the boundary in $\mathbb{P}$ or far away from $\mathbf{0}$ in $\mathbb{D}$ as $K \to 0$, this is exactly the reason for numerical instabilities (apart from the standard numerical problem of representing large numbers in floating-point notation).

Because of the above reasons, we do some of our experiments with the projected spaces and others with the hyperboloid and hypersphere, and aim to compare the performance of these empirically as well.

## A.3 EUCLIDEAN GEOMETRY

### A.3.1 EUCLIDEAN SPACE

**Distance function**   The distance function in $\mathbb{E}^n$ is

$$d_{\mathbb{E}}(\boldsymbol{x}, \boldsymbol{y}) = \|\boldsymbol{x} - \boldsymbol{y}\|_2 .$$

Due to the Pythagorean theorem, we can derive that

$$\|\boldsymbol{x} - \boldsymbol{y}\|_2^2 = \langle \boldsymbol{x} - \boldsymbol{y}, \boldsymbol{x} - \boldsymbol{y} \rangle_2 = \|\boldsymbol{x}\|_2^2 - 2 \langle \boldsymbol{x}, \boldsymbol{y} \rangle_2 + \|\boldsymbol{y}\|_2^2$$
$$= \|\boldsymbol{x}\|_2^2 + \|\boldsymbol{y}\|_2^2 - 2 \|\boldsymbol{x}\|_2 \|\boldsymbol{y}\|_2 \cos^{-1} \theta_{\boldsymbol{x}, \boldsymbol{y}}$$

**Exponential map**   The exponential map in $\mathbb{E}^n$ is

$$\exp_{\boldsymbol{x}}(\boldsymbol{v}) = \boldsymbol{x} + \boldsymbol{v}.$$

The fact that the resulting points belong to the space is trivial. Deriving the inverse function, i.e. the logarithmic map, is also trivial:

$$\log_{\boldsymbol{x}}(\boldsymbol{y}) = \boldsymbol{y} - \boldsymbol{x}.$$

**Parallel transport**   We do not need parallel transport in the Euclidean space, as we can directly sample from a Normal distribution. In other words, we can just define parallel transport to be an identity function.

## A.4 HYPERBOLIC GEOMETRY

### A.4.1 HYPERBOLOID

Do note, that all the theorems for the hypersphere are essentially trivial corollaries of their equivalents in the hypersphere (and vice-versa) (Section A.5.1). Notable differences include the fact that $R^2 = -\frac{1}{K}$, not $R^2 = \frac{1}{K}$, and all the operations use the hyperbolic trigonometric functions $\sinh$, $\cosh$, and $\tanh$, instead of their Euclidean counterparts. Also, we often leverage the "hyperbolic" Pythagorean theorem, in the form $\cosh^2(\alpha) - \sinh^2(\alpha) = 1$.

**Projections**   Due to the definition of the space as a retraction from the ambient space, we can project a generic vector in the ambient space to the hyperboloid using the shortest Euclidean distance by normalization:

$$\text{proj}_{\mathbb{H}_K^n}(\boldsymbol{x}) = R \frac{\boldsymbol{x}}{\|\boldsymbol{x}\|_{\mathcal{L}}} = \frac{\boldsymbol{x}}{\sqrt{K} \|\boldsymbol{x}\|_{\mathcal{L}}}.$$

Secondly, the $n + 1$ coordinates of a point on the hyperboloid are co-dependent; they satisfy the relation $\langle \boldsymbol{x}, \boldsymbol{x} \rangle_{\mathcal{L}} = 1/K$. This implies, that if we are given a vector with $n$ coordinates $\tilde{\boldsymbol{x}} = (x_2, \ldots, x_{n+1})$, we can compute the missing coordinate to place it onto the hyperboloid:

$$x_1 = \sqrt{\|\tilde{\boldsymbol{x}}\|_2^2 - \frac{1}{K}}.$$

This is useful for example in the case of orthogonally projecting points from $\mathcal{T}_{\boldsymbol{\mu}_0} \mathbb{H}_K^n$ onto the manifold.

**Distance function**  The distance function in $\mathbb{H}_K^n$ is

$$d_{\mathbb{H}}^K(\boldsymbol{x}, \boldsymbol{y}) = R \cdot \theta_{\boldsymbol{x},\boldsymbol{y}} = R \cosh^{-1}\left(-\frac{\langle \boldsymbol{x}, \boldsymbol{y}\rangle_{\mathcal{L}}}{R^2}\right) = \frac{1}{\sqrt{-K}} \cosh^{-1}\left(-K \langle \boldsymbol{x}, \boldsymbol{y}\rangle_{\mathcal{L}}\right).$$

**Remark A.1** (About the divergence of points in $\mathbb{H}_K^n$). *Since the points on the hyperboloid $\boldsymbol{x} \in \mathbb{H}_K^n$ are norm-constrained to*

$$\langle \boldsymbol{x}, \boldsymbol{x}\rangle_{\mathcal{L}} = \frac{1}{K},$$

*all the points on the hyperboloid go to infinity as $K$ goes to $0^-$ from below:*

$$\lim_{K \to 0^-} \langle \boldsymbol{x}, \boldsymbol{x}\rangle_{\mathcal{L}} = -\infty.$$

This confirms the intuition that the hyperboloid grows "flatter", but to do that, it has to go away from the origin of the coordinate space $\boldsymbol{0}$. A good example of a point that diverges is the origin of the hyperboloid $\boldsymbol{\mu}_0^K = (1/K, 0, \ldots, 0)^T = (R, 0, \ldots, 0)^T$. That makes this model unsuitable for trying to learn sign-agnostic curvatures, similarly to the hypersphere.

**Exponential map**  The exponential map in $\mathbb{H}_K^n$ is

$$\exp_{\boldsymbol{x}}^K(\boldsymbol{v}) = \cosh\left(\frac{||\boldsymbol{v}||_{\mathcal{L}}}{R}\right) \boldsymbol{x} + \sinh\left(\frac{||\boldsymbol{v}||_{\mathcal{L}}}{R}\right) \frac{R\boldsymbol{v}}{||\boldsymbol{v}||_{\mathcal{L}}},$$

and in the case of $\boldsymbol{x} := \boldsymbol{\mu}_0 = (R, 0, \ldots, 0)^T$:

$$\exp_{\boldsymbol{\mu}_0}^K(\boldsymbol{v}) = \left(\cosh\left(\frac{||\tilde{\boldsymbol{v}}||_2}{R}\right) R;\ \sinh\left(\frac{||\tilde{\boldsymbol{v}}||_2}{R}\right) \frac{R}{||\tilde{\boldsymbol{v}}||_2} \tilde{\boldsymbol{v}}^T\right)^T,$$

where $\boldsymbol{v} = (0; \tilde{\boldsymbol{v}}^T)^T$ and $||\boldsymbol{v}||_{\mathcal{L}} = ||\boldsymbol{v}||_2 = ||\tilde{\boldsymbol{v}}||_2$.

**Theorem A.2** (Logarithmic map in $\mathbb{H}_K^n$). *For all $\boldsymbol{x}, \boldsymbol{y} \in \mathbb{H}_K^n$, the logarithmic map in $\mathbb{H}_K^n$ maps $\boldsymbol{y}$ to a tangent vector at $\boldsymbol{x}$:*

$$\log_{\boldsymbol{x}}^K(\boldsymbol{y}) = \frac{\cosh^{-1}(\alpha)}{\sqrt{\alpha^2 - 1}}(\boldsymbol{y} - \alpha\boldsymbol{x}),$$

*where $\alpha = K \langle \boldsymbol{x}, \boldsymbol{y}\rangle_{\mathcal{L}}$.*

*Proof.* We show the detailed derivation of the logarithmic map as an inverse function to the exponential map $\log_{\boldsymbol{x}}(\boldsymbol{y}) = \exp_{\boldsymbol{x}}^{-1}(\boldsymbol{y})$, adapted from (Nagano et al., 2019).

As mentioned previously,

$$\boldsymbol{y} = \exp_{\boldsymbol{x}}^K(\boldsymbol{v}) = \cosh\left(\frac{||\boldsymbol{v}||_{\mathcal{L}}}{R}\right) \boldsymbol{x} + \sinh\left(\frac{||\boldsymbol{v}||_{\mathcal{L}}}{R}\right) \frac{R\boldsymbol{v}}{||\boldsymbol{v}||_{\mathcal{L}}}.$$

Solving for $\boldsymbol{v}$, we obtain

$$\boldsymbol{v} = \frac{||\boldsymbol{v}||_{\mathcal{L}}}{R \sinh\left(\frac{||\boldsymbol{v}||_{\mathcal{L}}}{R}\right)}\left(\boldsymbol{y} - \cosh\left(\frac{||\boldsymbol{v}||_{\mathcal{L}}}{R}\right) \boldsymbol{x}\right).$$

However, we still need to rewrite $||\boldsymbol{v}||_{\mathcal{L}}$ in evaluatable terms:

$$0 = \langle \boldsymbol{x}, \boldsymbol{v}\rangle_{\mathcal{L}} = \frac{||\boldsymbol{v}||_{\mathcal{L}}}{R \sinh\left(\frac{||\boldsymbol{v}||_{\mathcal{L}}}{R}\right)}\left(\langle \boldsymbol{x}, \boldsymbol{y}\rangle_{\mathcal{L}} - \cosh\left(\frac{||\boldsymbol{v}||_{\mathcal{L}}}{R}\right) \underbrace{\langle \boldsymbol{x}, \boldsymbol{x}\rangle_{\mathcal{L}}}_{-R^2}\right),$$

hence

$$\cosh\left(\frac{||\boldsymbol{v}||_{\mathcal{L}}}{R}\right) = -\frac{1}{R^2} \langle \boldsymbol{x}, \boldsymbol{y}\rangle_{\mathcal{L}},$$

and therefore

$$||\boldsymbol{v}||_{\mathcal{L}} = R \cosh^{-1}\left(-\frac{1}{R^2} \langle \boldsymbol{x}, \boldsymbol{y}\rangle_{\mathcal{L}}\right) = \frac{1}{\sqrt{-K}} \cosh^{-1}(K \langle \boldsymbol{x}, \boldsymbol{y}\rangle_{\mathcal{L}}) = d_{\mathbb{H}}^K(\boldsymbol{x}, \boldsymbol{y}).$$

Plugging the result back into the first equation, we obtain

$$
\begin{aligned}
\boldsymbol{v} &= \frac{\|\boldsymbol{v}\|_{\mathcal{L}}}{R\sinh\left(\frac{\|\boldsymbol{v}\|_{\mathcal{L}}}{R}\right)}\left(\boldsymbol{y}-\cosh\left(\frac{\|\boldsymbol{v}\|_{\mathcal{L}}}{R}\right)\boldsymbol{x}\right) \\
&= \frac{R\cosh^{-1}(\alpha)}{R\sinh\left(\frac{1}{R}R\cosh^{-1}(\alpha)\right)}\left(\boldsymbol{y}-\cosh\left(\frac{1}{R}R\cosh^{-1}(\alpha)\right)\boldsymbol{x}\right) \\
&= \frac{\cosh^{-1}(\alpha)}{\sinh(\cosh^{-1}(\alpha))}(\boldsymbol{y}-\cosh(\cosh^{-1}(\alpha))\boldsymbol{x}) \\
&= \frac{\cosh^{-1}(\alpha)}{\sqrt{\alpha^2-1}}(\boldsymbol{y}-\alpha\boldsymbol{x}),
\end{aligned}
$$

where $\alpha = -\frac{1}{R^2}\langle\boldsymbol{x},\boldsymbol{y}\rangle_{\mathcal{L}} = K\langle\boldsymbol{x},\boldsymbol{y}\rangle_{\mathcal{L}}$, and the last equality assumes $|\alpha| > 1$. This assumption holds, since for all points $\boldsymbol{x},\boldsymbol{y}\in\mathbb{H}_K^n$ it holds that $\langle\boldsymbol{x},\boldsymbol{y}\rangle_{\mathcal{L}}\leq -R^2$, and $\langle\boldsymbol{x},\boldsymbol{y}\rangle_{\mathcal{L}} = -R^2$ if and only if $\boldsymbol{x}=\boldsymbol{y}$, due to Cauchy-Schwarz (Ratcliffe, 2006, Theorem 3.1.6). Hence, the only case where this would be a problem would be if $\boldsymbol{x}=\boldsymbol{y}$, but it is clear that the result in that case is $\boldsymbol{u}=\boldsymbol{0}$. $\qquad\square$

**Parallel transport**   Using the generic formula for parallel transport in manifolds for $\boldsymbol{x},\boldsymbol{y}\in\mathcal{M}$ and $\boldsymbol{v}\in\mathcal{T}_{\boldsymbol{x}}\mathcal{M}$

$$
\mathrm{PT}_{\boldsymbol{x}\to\boldsymbol{y}}^K(\boldsymbol{v}) = \boldsymbol{v} - \frac{\left\langle\log_{\boldsymbol{x}}^K(\boldsymbol{y}),\boldsymbol{v}\right\rangle_{\boldsymbol{x}}}{d_{\mathcal{M}}(\boldsymbol{x},\boldsymbol{y})}(\log_{\boldsymbol{x}}^K(\boldsymbol{y})+\log_{\boldsymbol{y}}^K(\boldsymbol{x})), \tag{1}
$$

and the logarithmic map formula from Theorem A.2

$$
\log_{\boldsymbol{x}}^K(\boldsymbol{y}) = \frac{\cosh^{-1}(\alpha)}{\sqrt{\alpha^2-1}}(\boldsymbol{y}-\alpha\boldsymbol{x}),
$$

where $\alpha = -\frac{1}{R^2}\langle\boldsymbol{x},\boldsymbol{y}\rangle_{\mathcal{L}}$, we derive parallel transport in $\mathbb{H}_K^n$:

$$
\mathrm{PT}_{\boldsymbol{x}\to\boldsymbol{y}}^K(\boldsymbol{v}) = \boldsymbol{v} + \frac{\langle\boldsymbol{y},\boldsymbol{v}\rangle_{\mathcal{L}}}{R^2-\langle\boldsymbol{x},\boldsymbol{y}\rangle_{\mathcal{L}}}(\boldsymbol{x}+\boldsymbol{y}).
$$

A special form of parallel transport exists for when the source vector is $\boldsymbol{\mu}_0 = (R,0,\ldots,0)^T$:

$$
\mathrm{PT}_{\boldsymbol{\mu}_0\to\boldsymbol{y}}^K(\boldsymbol{v}) = \boldsymbol{v} + \frac{\langle\boldsymbol{y},\boldsymbol{v}\rangle_2}{R^2+Ry_1}\begin{pmatrix}y_1+R\\y_2\\\vdots\\y_{n+1}\end{pmatrix}.
$$

### A.4.2   POINCARÉ BALL

Do note, that all the theorems for the projected hypersphere are essentially trivial corollaries of their equivalents in the Poincaré ball (and vice-versa) (Section A.5.2). Notable differences include the fact that $R^2 = -\frac{1}{K}$, not $R^2 = \frac{1}{K}$, and all the operations use the hyperbolic trigonometric functions $\sinh$, $\cosh$, and $\tanh$, instead of their Euclidean counterparts. Also, we often leverage the "hyperbolic" Pythagorean theorem, in the form $\cosh^2(\alpha)-\sinh^2(\alpha) = 1$.

**Stereographic projection**

**Theorem A.3** (Stereographic backprojected points of $\mathbb{P}_K^n$ belong to $\mathbb{H}_K^n$). *For all $\boldsymbol{y}\in\mathbb{P}_K^n$,*

$$
\left\|\rho_K^{-1}(\boldsymbol{y})\right\|_{\mathcal{L}}^2 = \frac{1}{K}.
$$

*Proof.*

$$
\left\|\rho_K^{-1}(\boldsymbol{y})\right\|_{\mathcal{L}}^2 = \left\|\left(\frac{1}{\sqrt{|K|}}\frac{K\|\boldsymbol{y}\|_2^2-1}{K\|\boldsymbol{y}\|_2^2+1};\frac{2\boldsymbol{y}^T}{K\|\boldsymbol{y}\|_2^2+1}\right)^T\right\|_{\mathcal{L}}^2
$$

$$= -\left(\frac{1}{\sqrt{|K|}}\frac{K\|\boldsymbol{y}\|_2^2 - 1}{K\|\boldsymbol{y}\|_2^2 + 1}\right)^2 + \frac{4\|\boldsymbol{y}\|_2^2}{(K\|\boldsymbol{y}\|_2^2 + 1)^2}$$

$$= \frac{1}{|K|}\frac{-(K\|\boldsymbol{y}\|_2^2 - 1)^2 + 4|K|\|\boldsymbol{y}\|_2^2}{(K\|\boldsymbol{y}\|_2^2 + 1)^2}$$

$$= \frac{1}{-K}\frac{-(K\|\boldsymbol{y}\|_2^2 - 1)^2 - 4K\|\boldsymbol{y}\|_2^2}{(K\|\boldsymbol{y}\|_2^2 + 1)^2}$$

$$= \frac{1}{K}\frac{(K\|\boldsymbol{y}\|_2^2 - 1)^2 + 4K\|\boldsymbol{y}\|_2^2}{(K\|\boldsymbol{y}\|_2^2 + 1)^2}$$

$$= \frac{1}{K}\frac{K^2\|\boldsymbol{y}\|_2^4 + 2K\|\boldsymbol{y}\|_2^2 + 1}{(K\|\boldsymbol{y}\|_2^2 + 1)^2}$$

$$= \frac{1}{K}\frac{(K\|\boldsymbol{y}\|_2^2 + 1)^2}{(K\|\boldsymbol{y}\|_2^2 + 1)^2} = \frac{1}{K}.$$

$\square$

**Distance function**   The distance function in $\mathbb{P}_K^n$ is (derived from the hyperboloid distance function using the stereographic projection $\rho_K$):

$$d_{\mathbb{P}}(\boldsymbol{x}, \boldsymbol{y}) = d_{\mathbb{H}}(\rho_K^{-1}(\boldsymbol{x}), \rho_K^{-1}(\boldsymbol{y}))$$

$$= \frac{1}{\sqrt{-K}}\cosh^{-1}\left(1 - \frac{2K\|\boldsymbol{x} - \boldsymbol{y}\|_2^2}{(1 + K\|\boldsymbol{x}\|_2^2)(1 + K\|\boldsymbol{y}\|_2^2)}\right)$$

$$= R\cosh^{-1}\left(1 + \frac{2R^2\|\boldsymbol{x} - \boldsymbol{y}\|_2^2}{(R^2 - \|\boldsymbol{x}\|_2^2)(R^2 - \|\boldsymbol{y}\|_2^2)}\right)$$

**Theorem A.4** (Distance equivalence in $\mathbb{P}_K^n$)**.**  *For all $K < 0$ and for all pairs of points $\boldsymbol{x}, \boldsymbol{y} \in \mathbb{P}_K^n$, the Poincaré distance between them equals the gyrospace distance*

$$d_{\mathbb{P}}(\boldsymbol{x}, \boldsymbol{y}) = d_{\mathbb{P}_{gyr}}(\boldsymbol{x}, \boldsymbol{y}).$$

*Proof.*  Proven using Mathematica (File: `distance_limits.ws`), proof involves heavy algebra.
$\square$

**Theorem A.5** (Gyrospace distance converges to Euclidean in $\mathbb{P}_K^n$)**.**  *For any fixed pair of points $\boldsymbol{x}, \boldsymbol{y} \in \mathbb{P}_K^n$, the Poincaré gyrospace distance between them converges to the Euclidean distance in the limit (up to a constant) as $K \to 0^-$:*

$$\lim_{K \to 0^-} d_{\mathbb{P}_{gyr}}(\boldsymbol{x}, \boldsymbol{y}) = 2\|\boldsymbol{x} - \boldsymbol{y}\|_2.$$

*Proof.*

$$\lim_{K \to 0^-} d_{\mathbb{P}_{gyr}}(\boldsymbol{x}, \boldsymbol{y}) = 2\lim_{K \to 0^-}\left[\frac{\tanh^{-1}(\sqrt{-K}\|-\boldsymbol{x} \oplus_K \boldsymbol{y}\|_2)}{\sqrt{-K}}\right]$$

$$= 2\lim_{K \to 0^-}\left[\frac{\tanh^{-1}(\sqrt{-K}\|\boldsymbol{y} - \boldsymbol{x}\|_2)}{\sqrt{-K}}\right]$$

$$= 2\|\boldsymbol{y} - \boldsymbol{x}\|_2,$$

where the second equality holds because of the theorem of limits of composed functions, where

$$f(a) = \frac{\tanh^{-1}(a\sqrt{-K})}{\sqrt{-K}}$$

$$g(K) = \|-\boldsymbol{x} \oplus_K \boldsymbol{y}\|_2 \,.$$

We see that

$$\lim_{K \to 0^-} g(K) = \|\boldsymbol{y} - \boldsymbol{x}\|_2$$

due to Theorem A.14, and

$$\lim_{a \to \|\boldsymbol{x}-\boldsymbol{y}\|_2} f(a) = \frac{\tanh^{-1}(a\sqrt{-K})}{\sqrt{-K}}$$

Additionally for the last equality, we need the fact that

$$\lim_{x \to 0} \frac{\tanh^{-1}(a\sqrt{|x|})}{\sqrt{|x|}} = a.$$

$\square$

**Theorem A.6** (Distance converges to Euclidean as $K \to 0^-$ in $\mathbb{P}_K^n$). *For any* fixed *pair of points* $\boldsymbol{x}, \boldsymbol{y} \in \mathbb{P}_K^n$, *the Poincaré distance between them converges to the Euclidean distance in the limit (up to a constant) as* $K \to 0^-$:

$$\lim_{K \to 0^-} d_{\mathbb{P}}(\boldsymbol{x}, \boldsymbol{y}) = 2 \|\boldsymbol{x} - \boldsymbol{y}\|_2 \,.$$

*Proof.* Theorem A.4 and A.5. $\square$

**Exponential map** As derived and proven in Ganea et al. (2018a), the exponential map in $\mathbb{P}_K^n$ and its inverse is

$$\exp_{\boldsymbol{x}}^K(\boldsymbol{v}) = \boldsymbol{x} \oplus_K \left( \tanh\left(\sqrt{-K}\frac{\lambda_{\boldsymbol{x}}^K \|\boldsymbol{v}\|_2}{2}\right) \frac{\boldsymbol{v}}{\sqrt{-K} \|\boldsymbol{v}\|_2} \right)$$

$$\log_{\boldsymbol{x}}^K(\boldsymbol{y}) = \frac{2}{\sqrt{-K}\lambda_{\boldsymbol{x}}^K} \tanh^{-1}\left(\sqrt{-K} \|-\boldsymbol{x} \oplus_K \boldsymbol{y}\|_2\right) \frac{-\boldsymbol{x} \oplus_K \boldsymbol{y}}{\|-\boldsymbol{x} \oplus_K \boldsymbol{y}\|_2}$$

In the case of $\boldsymbol{x} := \boldsymbol{\mu}_0 = (0, \dots, 0)^T$ they simplify to:

$$\exp_{\boldsymbol{\mu}_0}^K(\boldsymbol{v}) = \tanh\left(\sqrt{-K} \|\boldsymbol{v}\|_2\right) \frac{\boldsymbol{v}}{\sqrt{-K} \|\boldsymbol{v}\|_2}$$

$$\log_{\boldsymbol{\mu}_0}^K(\boldsymbol{y}) = \tanh^{-1}\left(\sqrt{-K} \|\boldsymbol{y}\|_2\right) \frac{\boldsymbol{y}}{\|\boldsymbol{y}\|_2}.$$

**Parallel transport** Kochurov et al. (2019); Ganea et al. (2018a) have also derived and implemented the parallel transport operation for the Poincaré ball:

$$\mathrm{PT}_{\boldsymbol{x} \to \boldsymbol{y}}^K(\boldsymbol{v}) = \frac{\lambda_{\boldsymbol{x}}^K}{\lambda_{\boldsymbol{y}}^K} \mathrm{gyr}[\boldsymbol{y}, -\boldsymbol{x}]\boldsymbol{v},$$

$$\mathrm{PT}_{\boldsymbol{\mu}_0 \to \boldsymbol{y}}^K(\boldsymbol{v}) = \frac{2}{\lambda_{\boldsymbol{y}}^K}\boldsymbol{v},$$

$$\mathrm{PT}_{\boldsymbol{x} \to \boldsymbol{\mu}_0}^K(\boldsymbol{v}) = \frac{\lambda_{\boldsymbol{x}}^K}{2}\boldsymbol{v},$$

where

$$\mathrm{gyr}[\boldsymbol{x}, \boldsymbol{y}]\boldsymbol{v} = -(\boldsymbol{x} \oplus_K \boldsymbol{y}) \oplus_K (\boldsymbol{x} \oplus_K (\boldsymbol{y} \oplus_K \boldsymbol{v}))$$

is the gyration operation (Ungar, 2008, Definition 1.11).

Unfortunately, on the Poincaré ball, $\langle \cdot, \cdot \rangle_{\boldsymbol{x}}$ has a form that changes with respect to $\boldsymbol{x}$, unlike in the hyperboloid.

## A.5 SPHERICAL GEOMETRY

### A.5.1 HYPERSPHERE

All the theorems for the hypersphere are essentially trivial corollaries of their equivalents in the hyperboloid (Section A.4.1). Notable differences include the fact that $R^2 = \frac{1}{K}$, not $R^2 = -\frac{1}{K}$, and all the operations use the Euclidean trigonometric functions $\sin$, $\cos$, and $\tan$, instead of their hyperbolic counterparts. Also, we often leverage the Pythagorean theorem, in the form $\sin^2(\alpha) + \cos^2(\alpha) = 1$.

**Projections**   Due to the definition of the space as a retraction from the ambient space, we can project a generic vector in the ambient space to the hypersphere using the shortest Euclidean distance by normalization:

$$\mathrm{proj}_{\mathbb{S}_K^{n-1}}(\boldsymbol{x}) = R\frac{\boldsymbol{x}}{||\boldsymbol{x}||_2} = \frac{\boldsymbol{x}}{\sqrt{K}\,||\boldsymbol{x}||_2}.$$

Secondly, the $n+1$ coordinates of a point on the sphere are co-dependent; they satisfy the relation $\langle \boldsymbol{x}, \boldsymbol{x} \rangle_2 = 1/K$. This implies, that if we are given a vector with $n$ coordinates $\tilde{\boldsymbol{x}} = (x_2, \ldots, x_{n+1})$, we can compute the missing coordinate to place it onto the sphere:

$$x_1 = \sqrt{\frac{1}{K} - ||\tilde{\boldsymbol{x}}||_2^2}.$$

This is useful for example in the case of orthogonally projecting points from $\mathcal{T}_{\boldsymbol{\mu}_0}\mathbb{S}_K^n$ onto the manifold.

**Distance function**   The distance function in $\mathbb{S}_K^n$ is

$$d_{\mathbb{S}}^K(\boldsymbol{x}, \boldsymbol{y}) = R \cdot \theta_{\boldsymbol{x},\boldsymbol{y}} = R\cos^{-1}\left(\frac{\langle \boldsymbol{x}, \boldsymbol{y} \rangle_2}{R^2}\right) = \frac{1}{\sqrt{K}}\cos^{-1}\left(K \langle \boldsymbol{x}, \boldsymbol{y} \rangle_2\right).$$

**Remark A.7** (About the divergence of points in $\mathbb{S}_K^n$). *Since the points on the hypersphere $\boldsymbol{x} \in \mathbb{S}_K^n$ are norm-constrained to*

$$\langle \boldsymbol{x}, \boldsymbol{x} \rangle_2 = \frac{1}{K},$$

*all the points on the sphere go to infinity as $K$ goes to $0^+$ from above:*

$$\lim_{K \to 0^+} \langle \boldsymbol{x}, \boldsymbol{x} \rangle_2 = \infty.$$

This confirms the intuition that the sphere grows "flatter", but to do that, it has to go away from the origin of the coordinate space $\boldsymbol{0}$. A good example of a point that diverges is the north pole of the sphere $\boldsymbol{\mu}_0^K = (1/K, 0, \ldots, 0)^T = (R, 0, \ldots, 0)^T$. That makes this model unsuitable for trying to learn sign-agnostic curvatures, similarly to the hyperboloid.

**Exponential map**   The exponential map in $\mathbb{S}_K^n$ is

$$\exp_{\boldsymbol{x}}^K(\boldsymbol{v}) = \cos\left(\frac{||\boldsymbol{v}||_2}{R}\right)\boldsymbol{x} + \sin\left(\frac{||\boldsymbol{v}||_2}{R}\right)\frac{R\boldsymbol{v}}{||\boldsymbol{v}||_2}.$$

**Theorem A.8** (Logarithmic map in $\mathbb{S}_K^n$). *For all $\boldsymbol{x}, \boldsymbol{y} \in \mathbb{S}_K^n$, the logarithmic map in $\mathbb{S}_K^n$ maps $\boldsymbol{y}$ to a tangent vector at $\boldsymbol{x}$:*

$$\log_{\boldsymbol{x}}^K(\boldsymbol{y}) = \frac{\cos^{-1}(\alpha)}{\sqrt{1 - \alpha^2}}(\boldsymbol{y} - \alpha\boldsymbol{x}),$$

*where $\alpha = K \langle \boldsymbol{x}, \boldsymbol{y} \rangle_2$.*

*Proof.* Analogous to the proof of Theorem A.2.

As mentioned previously,

$$\boldsymbol{y} = \exp_{\boldsymbol{x}}^K(\boldsymbol{v}) = \cos\left(\frac{||\boldsymbol{v}||_2}{R}\right)\boldsymbol{x} + \sin\left(\frac{||\boldsymbol{v}||_2}{R}\right)\frac{R\boldsymbol{v}}{||\boldsymbol{v}||_2}.$$

Solving for $v$, we obtain

$$v = \frac{||v||_2}{R \sin\left(\frac{||v||_2}{R}\right)} \left(y - \cos\left(\frac{||v||_2}{R}\right) x\right).$$

However, we still need to rewrite $||v||_2$ in evaluatable terms:

$$0 = \langle x, v \rangle_2 = \frac{||v||_2}{R \sin\left(\frac{||v||_2}{R}\right)} \left(\langle x, y \rangle_2 - \cos\left(\frac{||v||_2}{R}\right) \underbrace{\langle x, x \rangle_2}_{R^2}\right),$$

hence

$$\cos\left(\frac{||v||_2}{R}\right) = \frac{1}{R^2} \langle x, y \rangle_2,$$

and therefore

$$||v||_2 = R \cos^{-1}\left(\frac{1}{R^2} \langle x, y \rangle_2\right) = \frac{1}{\sqrt{K}} \cos^{-1}(K \langle x, y \rangle_2) = d_{\mathbb{S}}^K(x, y).$$

Plugging the result back into the first equation, we obtain

$$\begin{aligned}
v &= \frac{||v||_2}{R \sin\left(\frac{||v||_2}{R}\right)} \left(y - \cos\left(\frac{||v||_2}{R}\right) x\right) \\
&= \frac{R \cos^{-1}(\alpha)}{R \sin\left(\frac{1}{R} R \cos^{-1}(\alpha)\right)} \left(y - \cos\left(\frac{1}{R} R \cos^{-1}(\alpha)\right) x\right) \\
&= \frac{\cos^{-1}(\alpha)}{\sin(\cos^{-1}(\alpha))} (y - \cos(\cos^{-1}(\alpha))x) \\
&= \frac{\cos^{-1}(\alpha)}{\sqrt{1 - \alpha^2}} (y - \alpha x),
\end{aligned}$$

where $\alpha = \frac{1}{R^2} \langle x, y \rangle_2 = K \langle x, y \rangle_2$, and the last equality assumes $|\alpha| > 1$. This assumption holds, since for all points $x, y \in \mathbb{S}_K^n$ it holds that $\langle x, y \rangle_2 \le R^2$, and $\langle x, y \rangle_2 = R^2$ if and only if $x = y$, due to Cauchy-Schwarz (Ratcliffe, 2006, Theorem 3.1.6). Hence, the only case where this would be a problem would be if $x = y$, but it is clear that the result in that case is $u = 0$. □

**Parallel transport**  Using the generic formula for parallel transport in manifolds (Equation A.4.1) for $x, y \in \mathbb{S}_K^n$ and $v \in \mathcal{T}_x \mathbb{S}_K^n$ and the spherical logarithmic map formula

$$\log_x^K(y) = \frac{\cos^{-1}(\alpha)}{\sqrt{1 - \alpha^2}} (y - \alpha x),$$

where $\alpha = K \langle x, y \rangle_2$, we derive parallel transport in $\mathbb{S}_K^n$:

$$\begin{aligned}
\mathrm{PT}_{x \to y}^K(v) &= v - \frac{\langle y, v \rangle_2}{R^2 + \langle x, y \rangle_2} (x + y) \\
&= v - \frac{K \langle y, v \rangle_2}{1 + K \langle x, y \rangle_2} (x + y).
\end{aligned}$$

A special form of parallel transport exists for when the source vector is $\mu_0 = (R, 0, \ldots, 0)^T$:

$$\mathrm{PT}_{\mu_0 \to y}^K(v) = v - \frac{\langle y, v \rangle_2}{R^2 + R y_1} \begin{pmatrix} y_1 + R \\ y_2 \\ \vdots \\ y_{n+1} \end{pmatrix}.$$

### A.5.2  PROJECTED HYPERSPHERE

Do note, that all the theorems for the projected hypersphere are essentially trivial corollaries of their equivalents in the Poincaré ball (and vice-versa) (Section A.4.2). Notable differences include the fact that $R^2 = \frac{1}{K}$, not $R^2 = -\frac{1}{K}$, and all the operations use the Euclidean trigonometric functions $\sin$, $\cos$, and $\tan$, instead of their hyperbolic counterparts. Also, we often leverage the Pythagorean theorem, in the form $\sin^2(\alpha) + \cos^2(\alpha) = 1$.

**Stereographic projection**

**Remark A.9** (Homeomorphism between $\mathbb{S}_K^n$ and $\mathbb{R}^n$). *We notice that $\rho_K$ is not a homeomorphism between the $n$-dimensional sphere and $\mathbb{R}^n$, as it is not defined at $-\boldsymbol{\mu}_0 = (-R; \mathbf{0}^T)^T$. If we additionally changed compactified the plane by adding a point "at infinity" and set it equal to $\rho_K(\boldsymbol{\mu}_0)$, $\rho_K$ would become a homeomorphism.*

**Theorem A.10** (Stereographic backprojected points of $\mathbb{D}_K^n$ belong to $\mathbb{S}_K^n$). *For all $\boldsymbol{y} \in \mathbb{D}_K^n$,*

$$\left\| \rho_K^{-1}(\boldsymbol{y}) \right\|_2^2 = \frac{1}{K}.$$

*Proof.*

$$
\begin{aligned}
\left\| \rho_K^{-1}(\boldsymbol{y}) \right\|_2^2 &= \left\| \left( \frac{1}{\sqrt{|K|}} \frac{K \|\boldsymbol{y}\|_2^2 - 1}{K \|\boldsymbol{y}\|_2^2 + 1}; \frac{2\boldsymbol{y}^T}{K \|\boldsymbol{y}\|_2^2 + 1} \right)^T \right\|_2^2 \\
&= \left( \frac{1}{\sqrt{|K|}} \frac{K \|\boldsymbol{y}\|_2^2 - 1}{K \|\boldsymbol{y}\|_2^2 + 1} \right)^2 + \frac{4 \|\boldsymbol{y}\|_2^2}{(K \|\boldsymbol{y}\|_2^2 + 1)^2} \\
&= \frac{1}{|K|} \frac{(K \|\boldsymbol{y}\|_2^2 - 1)^2 + 4|K| \|\boldsymbol{y}\|_2^2}{(K \|\boldsymbol{y}\|_2^2 + 1)^2} \\
&= \frac{1}{K} \frac{(K \|\boldsymbol{y}\|_2^2 - 1)^2 + 4K \|\boldsymbol{y}\|_2^2}{(K \|\boldsymbol{y}\|_2^2 + 1)^2} \\
&= \frac{1}{K} \frac{K^2 \|\boldsymbol{y}\|_2^4 + 2K \|\boldsymbol{y}\|_2^2 + 1}{(K \|\boldsymbol{y}\|_2^2 + 1)^2} \\
&= \frac{1}{K} \frac{(K \|\boldsymbol{y}\|_2^2 + 1)^2}{(K \|\boldsymbol{y}\|_2^2 + 1)^2} = \frac{1}{K}.
\end{aligned}
$$

$\square$

**Distance function**    The distance function in $\mathbb{D}_K^n$ is (derived from the spherical distance function using the stereographic projection $\rho_K$):

$$
\begin{aligned}
d_{\mathbb{D}}(\boldsymbol{x}, \boldsymbol{y}) &= d_{\mathbb{S}}(\rho_K^{-1}(\boldsymbol{x}), \rho_K^{-1}(\boldsymbol{y})) \\
&= \frac{1}{\sqrt{K}} \cos^{-1} \left( 1 - \frac{2K \|\boldsymbol{x} - \boldsymbol{y}\|_2^2}{(1 + K \|\boldsymbol{x}\|_2^2)(1 + K \|\boldsymbol{y}\|_2^2)} \right) \\
&= R \cos^{-1} \left( 1 - \frac{2R^2 \|\boldsymbol{x} - \boldsymbol{y}\|_2^2}{(R^2 + \|\boldsymbol{x}\|_2^2)(R^2 + \|\boldsymbol{y}\|_2^2)} \right)
\end{aligned}
$$

**Theorem A.11** (Distance equivalence in $\mathbb{D}_K^n$). *For all $K > 0$ and for all pairs of points $\boldsymbol{x}, \boldsymbol{y} \in \mathbb{D}_K^n$, the spherical projected distance between them equals the gyrospace distance*

$$d_{\mathbb{D}}(\boldsymbol{x}, \boldsymbol{y}) = d_{\mathbb{D}_{gyr}}(\boldsymbol{x}, \boldsymbol{y}).$$

*Proof.* Proven using Mathematica (File: `distance_limits.ws`), proof involves heavy algebra.
$\square$

**Theorem A.12** (Gyrospace distance converges to Euclidean in $\mathbb{D}_K^n$). *For any* fixed *pair of points $\boldsymbol{x}, \boldsymbol{y} \in \mathbb{D}_K^n$, the spherical projected gyrospace distance between them converges to the Euclidean distance in the limit (up to a constant) as $K \to 0^+$:*

$$\lim_{K \to 0^+} d_{\mathbb{D}_{gyr}}(\boldsymbol{x}, \boldsymbol{y}) = 2 \|\boldsymbol{x} - \boldsymbol{y}\|_2 .$$

*Proof.*

$$\lim_{K \to 0^+} d_{\mathbb{D}_{\text{gyr}}}(\boldsymbol{x}, \boldsymbol{y}) = 2 \lim_{K \to 0^+} \left[ \frac{\tan^{-1}(\sqrt{K} \, \|-\boldsymbol{x} \oplus_K \boldsymbol{y}\|_2)}{\sqrt{K}} \right]$$

$$= 2 \lim_{K \to 0^+} \left[ \frac{\tan^{-1}(\sqrt{K} \, \|\boldsymbol{y} - \boldsymbol{x}\|_2)}{\sqrt{K}} \right]$$

$$= 2 \, \|\boldsymbol{y} - \boldsymbol{x}\|_2 \,,$$

where the second equality holds because of the theorem of limits of composed functions, where

$$f(a) = \frac{\tan^{-1}(a\sqrt{K})}{\sqrt{K}}$$

$$g(K) = \|-\boldsymbol{x} \oplus_K \boldsymbol{y}\|_2 \,.$$

We see that

$$\lim_{K \to 0^-} g(K) = \|\boldsymbol{y} - \boldsymbol{x}\|_2$$

due to Theorem A.14, and

$$\lim_{a \to \|\boldsymbol{x} - \boldsymbol{y}\|_2} f(a) = \frac{\tan^{-1}(a\sqrt{K})}{\sqrt{K}}$$

Additionally for the last equality, we need the fact that

$$\lim_{x \to 0} \frac{\tanh^{-1}(a\sqrt{|x|})}{\sqrt{|x|}} = a.$$

$\square$

**Theorem A.13** (Distance converges to Euclidean as $K \to 0^+$ in $\mathbb{D}_K^n$)**.** *For any* fixed *pair of points* $\boldsymbol{x}, \boldsymbol{y} \in \mathbb{D}_K^n$, *the spherical projected distance between them converges to the Euclidean distance in the limit (up to a constant) as* $K \to 0^+$:

$$\lim_{K \to 0^+} d_{\mathbb{D}}(\boldsymbol{x}, \boldsymbol{y}) = 2 \, \|\boldsymbol{x} - \boldsymbol{y}\|_2 \,.$$

*Proof.* Theorem A.11 and A.12. $\square$

**Exponential map** Analogously to the derivation of the exponential map in $\mathbb{P}_K^n$ in Ganea et al. (2018a, Section 2.3–2.4), we can derive Möbius scalar multiplication in $\mathbb{D}_K^n$:

$$r \otimes_K \boldsymbol{x} = \frac{1}{i\sqrt{K}} \tanh(r \tanh^{-1}(i\sqrt{K} \, \|\boldsymbol{x}\|_2)) \frac{\boldsymbol{x}}{\|\boldsymbol{x}\|_2}$$

$$= \frac{1}{i\sqrt{K}} \tanh(ri \tan^{-1}(\sqrt{K} \, \|\boldsymbol{x}\|_2)) \frac{\boldsymbol{x}}{\|\boldsymbol{x}\|_2}$$

$$= \frac{1}{\sqrt{K}} \tan(r \tan^{-1}(\sqrt{K} \, \|\boldsymbol{x}\|_2)) \frac{\boldsymbol{x}}{\|\boldsymbol{x}\|_2} \,,$$

where we use the fact that $\tanh^{-1}(ix) = i \tan^{-1}(x)$ and $\tanh(ix) = i \tan(x)$. We can easily see that $1 \otimes_K \boldsymbol{x} = \boldsymbol{x}$.

Hence, the geodesic has the form of

$$\gamma_{\boldsymbol{x} \to \boldsymbol{y}}(t) = \boldsymbol{x} \oplus_K t \otimes_K (-\boldsymbol{x} \oplus_K \boldsymbol{y}),$$

and therefore the exponential map in $\mathbb{D}_K^n$ is:

$$\exp_{\boldsymbol{x}}^K(\boldsymbol{v}) = \boldsymbol{x} \oplus_K \left( \tan\left( \sqrt{K} \frac{\lambda_{\boldsymbol{x}}^K \|\boldsymbol{v}\|_2}{2} \right) \frac{\boldsymbol{v}}{\sqrt{K} \, \|\boldsymbol{v}\|_2} \right).$$

The inverse formula can also be computed:

$$\log_{\boldsymbol{x}}^K(\boldsymbol{y}) = \frac{2}{\sqrt{K}\lambda_{\boldsymbol{x}}^K} \tan^{-1}\left(\sqrt{K} \left\|-\boldsymbol{x} \oplus_K \boldsymbol{y}\right\|_2\right) \frac{-\boldsymbol{x} \oplus_K \boldsymbol{y}}{\left\|-\boldsymbol{x} \oplus_K \boldsymbol{y}\right\|_2}$$

In the case of $\boldsymbol{x} := \boldsymbol{\mu}_0 = (0, \ldots, 0)^T$ they simplify to:

$$\exp_{\boldsymbol{\mu}_0}^K(\boldsymbol{v}) = \tan\left(\sqrt{K} \left\|\boldsymbol{v}\right\|_2\right) \frac{\boldsymbol{v}}{\sqrt{K} \left\|\boldsymbol{v}\right\|_2}$$

$$\log_{\boldsymbol{\mu}_0}^K(\boldsymbol{y}) = \tan^{-1}\left(\sqrt{K} \left\|\boldsymbol{y}\right\|_2\right) \frac{\boldsymbol{y}}{\sqrt{K} \left\|\boldsymbol{y}\right\|_2}.$$

**Parallel transport**  Similarly to the Poincaré ball, we can derive the parallel transport operation for the projected sphere:

$$\text{PT}_{\boldsymbol{x} \to \boldsymbol{y}}^K(\boldsymbol{v}) = \frac{\lambda_{\boldsymbol{x}}^K}{\lambda_{\boldsymbol{y}}^K} \text{gyr}[\boldsymbol{y}, -\boldsymbol{x}]\boldsymbol{v},$$

$$\text{PT}_{\boldsymbol{\mu}_0 \to \boldsymbol{y}}^K(\boldsymbol{v}) = \frac{2}{\lambda_{\boldsymbol{y}}^K} \boldsymbol{v},$$

$$\text{PT}_{\boldsymbol{x} \to \boldsymbol{\mu}_0}^K(\boldsymbol{v}) = \frac{\lambda_{\boldsymbol{x}}^K}{2} \boldsymbol{v},$$

where

$$\text{gyr}[\boldsymbol{x}, \boldsymbol{y}]\boldsymbol{v} = -(\boldsymbol{x} \oplus_K \boldsymbol{y}) \oplus_K (\boldsymbol{x} \oplus_K (\boldsymbol{y} \oplus_K \boldsymbol{v}))$$

is the gyration operation (Ungar, 2008, Definition 1.11).

Unfortunately, on the projected sphere, $\langle \cdot, \cdot \rangle_{\boldsymbol{x}}$ has a form that changes with respect to $\boldsymbol{x}$, similarly to the Poincaré ball and unlike in the hypersphere.

### A.6  MISCELLANEOUS PROPERTIES

**Theorem A.14** (Möbius addition converges to Eucl. vector addition)**.**

$$\lim_{K \to 0} (\boldsymbol{x} \oplus_K \boldsymbol{y}) = \boldsymbol{x} + \boldsymbol{y}.$$

*Note: This theorem works from both sides, hence applies to the Poincaré ball as well as the projected spherical space. Observe that the Möbius addition has the same form for both spaces.*

*Proof.*

$$\lim_{K \to 0} (\boldsymbol{x} \oplus_K \boldsymbol{y}) = \lim_{K \to 0} \left[\frac{(1 - 2K \langle \boldsymbol{x}, \boldsymbol{y} \rangle_2 - K \left\|\boldsymbol{y}\right\|_2^2)\boldsymbol{x} + (1 + K \left\|\boldsymbol{x}\right\|_2^2)\boldsymbol{y}}{1 - 2K \langle \boldsymbol{x}, \boldsymbol{y} \rangle_2 + K^2 \left\|\boldsymbol{x}\right\|_2^2 \left\|\boldsymbol{y}\right\|_2^2}\right]$$

$$= \boldsymbol{x} + \boldsymbol{y}.$$

$\square$

**Theorem A.15** ($\rho_K^{-1}$ is the inverse stereographic projection)**.**
*For all $(\xi; \boldsymbol{x}^T)^T \in \mathcal{M}_K^n$, $\xi \in \mathbb{R}$*

$$\rho_K^{-1}(\rho((\xi; \boldsymbol{x}^T)^T)) = \boldsymbol{x},$$

*where $\mathcal{M} \in \{\mathbb{S}, \mathbb{H}\}$.*

*Proof.*

$$\rho_K^{-1}(\rho_K((\xi; \boldsymbol{x}^T)^T)) = \rho_K^{-1}\left(\frac{\boldsymbol{x}}{1 - \sqrt{|K|}\xi}\right)$$

$$
= \left( \frac{1}{\sqrt{|K|}} \frac{K \left\| \frac{\boldsymbol{x}}{1-\sqrt{|K|}\xi} \right\|_2^2 - 1}{K \left\| \frac{\boldsymbol{x}}{1-\sqrt{|K|}\xi} \right\|_2^2 + 1} ; \frac{\frac{2\boldsymbol{x}^T}{1-\sqrt{|K|}\xi}}{K \left\| \frac{\boldsymbol{x}}{1-\sqrt{|K|}\xi} \right\|_2^2 + 1} \right)^T
$$

$$
= \frac{1/\sqrt{|K|}}{K \left\| \frac{\boldsymbol{x}}{1-\sqrt{|K|}\xi} \right\|_2^2 + 1} \left( K \left\| \frac{\boldsymbol{x}}{1 - \sqrt{|K|}\xi} \right\|_2^2 - 1 ; \frac{2\sqrt{|K|}\boldsymbol{x}^T}{1 - \sqrt{|K|}\xi} \right)^T
$$

$$
= \frac{1/\sqrt{|K|}}{\frac{K\|\boldsymbol{x}\|_2^2}{(1-\sqrt{|K|}\xi)^2} + 1} \left( \frac{K\|\boldsymbol{x}\|_2^2}{(1 - \sqrt{|K|}\xi)^2} - 1 ; \frac{2\sqrt{|K|}\boldsymbol{x}^T}{1 - \sqrt{|K|}\xi} \right)^T
$$

We observe that $\|\boldsymbol{x}\|_2^2 = \frac{1}{K} - \xi^2$, because $\boldsymbol{x} \in \mathcal{M}_K^n$. Therefore

$$
\rho_K^{-1}(\rho_K((\xi; \boldsymbol{x}^T)^T)) =
$$
$$
= \dots \quad \text{(above)}
$$

$$
= \frac{1/\sqrt{|K|}}{K\frac{\frac{1}{K} - \xi^2}{(1-\sqrt{|K|}\xi)^2} + 1} \left( K\frac{\frac{1}{K} - \xi^2}{(1 - \sqrt{|K|}\xi)^2} - 1 ; \frac{2\sqrt{|K|}\boldsymbol{x}^T}{1 - \sqrt{|K|}\xi} \right)^T
$$

$$
= \frac{1/\sqrt{|K|}}{\frac{(1-\sqrt{|K|}\xi)(1+\sqrt{|K|}\xi)}{(1-\sqrt{|K|}\xi)^2} + 1} \left( \frac{(1 - \sqrt{|K|}\xi)(1 + \sqrt{|K|}\xi)}{(1 - \sqrt{|K|}\xi)^2} - 1 ; \frac{2\sqrt{|K|}\boldsymbol{x}^T}{1 - \sqrt{|K|}\xi} \right)^T
$$

$$
= \frac{1/\sqrt{|K|}}{\frac{1+\sqrt{|K|}\xi}{1-\sqrt{|K|}\xi} + 1} \left( \frac{1 + \sqrt{|K|}\xi}{1 - \sqrt{|K|}\xi} - 1 ; \frac{2\sqrt{|K|}\boldsymbol{x}^T}{1 - \sqrt{|K|}\xi} \right)^T
$$

$$
= \frac{1/\sqrt{|K|}}{\frac{1+\sqrt{|K|}\xi+1-\sqrt{|K|}\xi}{1-\sqrt{|K|}\xi}} \left( \frac{1 + \sqrt{|K|}\xi - 1 + \sqrt{|K|}\xi}{1 - \sqrt{|K|}\xi} ; \frac{2\sqrt{|K|}\boldsymbol{x}^T}{1 - \sqrt{|K|}\xi} \right)^T
$$

$$
= \frac{1}{2\sqrt{|K|}} \left( 2\sqrt{|K|}\xi ; 2\sqrt{|K|}\boldsymbol{x}^T \right)^T = (\xi; \boldsymbol{x}^T)^T .
$$

$\square$

**Lemma A.16** ($\lambda_{\boldsymbol{x}}^K$ converges to 2 as $K \to 0$). *For all $\boldsymbol{x}$ in $\mathbb{P}_K^n$ or $\mathbb{D}_K^n$, it holds that*

$$
\lim_{K \to 0} \lambda_{\boldsymbol{x}}^K = 2.
$$

*Proof.*

$$
\lim_{K \to 0} \lambda_{\boldsymbol{x}}^K = \lim_{K \to 0} \frac{2}{1 + K \|\boldsymbol{x}\|_2^2} = 2.
$$

$\square$

**Theorem A.17** ($\exp_{\boldsymbol{x}}^K(\boldsymbol{v})$ converges to $\boldsymbol{x} + \boldsymbol{v}$ as $K \to 0$). *For all $\boldsymbol{x}$ in the Poincaré ball $\mathbb{P}_K^n$ or the projected sphere $\mathbb{D}_K^n$ and $\boldsymbol{v} \in \mathcal{T}_{\boldsymbol{x}}\mathcal{M}$, it holds that*

$$
\lim_{K \to 0} \exp_{\boldsymbol{x}}^K(\boldsymbol{v}) = \exp_{\boldsymbol{x}}(\boldsymbol{v}) = \boldsymbol{x} + \boldsymbol{v},
$$

*hence the exponential map converges to its Euclidean variant.*

*Proof.* For the positive case $K > 0$

$$\lim_{K \to 0^+} \exp_{\boldsymbol{x}}^K(\boldsymbol{v}) = \lim_{K \to 0^+} \left( \boldsymbol{x} \oplus_K \left( \tan_K \left( \sqrt{|K|} \frac{\lambda_{\boldsymbol{x}}^K \|\boldsymbol{v}\|_2}{2} \right) \frac{\boldsymbol{v}}{\sqrt{|K|} \|\boldsymbol{v}\|_2} \right) \right)$$

$$= \boldsymbol{x} + \lim_{K \to 0^+} \left( \tan_K \left( \sqrt{|K|} \frac{\lambda_{\boldsymbol{x}}^K \|\boldsymbol{v}\|_2}{2} \right) \frac{\boldsymbol{v}}{\sqrt{|K|} \|\boldsymbol{v}\|_2} \right)$$

$$= \boldsymbol{x} + \frac{\boldsymbol{v}}{\|\boldsymbol{v}\|_2} \lim_{K \to 0^+} \frac{\tan \left( \sqrt{K} \frac{\lambda_{\boldsymbol{x}}^K \|\boldsymbol{v}\|_2}{2} \right)}{\sqrt{K} \|\boldsymbol{v}\|_2}$$

$$= \boldsymbol{x} + \boldsymbol{v},$$

due to several applications of the theorem of limits of composed functions, Lemma A.16, and the fact that

$$\lim_{\alpha \to 0} \frac{\tan(\sqrt{\alpha} a)}{\sqrt{\alpha}} = a.$$

The negative case $K < 0$ is analogous. $\qquad \square$

**Theorem A.18** ($\log_{\boldsymbol{x}}^K(\boldsymbol{y})$ converges to $\boldsymbol{y} - \boldsymbol{x}$ as $K \to 0$)**.** *For all $\boldsymbol{x}, \boldsymbol{y}$ in the Poincaré ball $\mathbb{P}_K^n$ or the projected sphere $\mathbb{D}_K^n$, it holds that*

$$\lim_{K \to 0} \log_{\boldsymbol{x}}^K(\boldsymbol{y}) = \log_{\boldsymbol{x}}(\boldsymbol{v}) = \boldsymbol{y} - \boldsymbol{x},$$

*hence the logarithmic map converges to its Euclidean variant.*

*Proof.* Firstly,

$$\boldsymbol{z} = -\boldsymbol{x} \oplus_K \boldsymbol{y} \xrightarrow{K \to 0} \boldsymbol{y} - \boldsymbol{x},$$

due to Theorem A.14. For the positive case $K > 0$

$$\lim_{K \to 0^+} \log_{\boldsymbol{x}}^K(\boldsymbol{y}) = \lim_{K \to 0^+} \left( \frac{2}{\sqrt{|K|} \lambda_{\boldsymbol{x}}^K} \tan_K^{-1} \left( \sqrt{|K|} \|\boldsymbol{z}\|_2 \right) \frac{\boldsymbol{z}}{\|\boldsymbol{z}\|_2} \right)$$

$$= \lim_{K \to 0^+} \left( \frac{2}{\lambda_{\boldsymbol{x}}^K} \frac{\tan_K^{-1} \left( \sqrt{|K|} \|\boldsymbol{z}\|_2 \right)}{\sqrt{|K|} \|\boldsymbol{z}\|_2} \boldsymbol{z} \right)$$

$$= \lim_{K \to 0^+} \frac{2}{\lambda_{\boldsymbol{x}}^K} \cdot \lim_{K \to 0^+} \frac{\tan^{-1} \left( \sqrt{K} \|\boldsymbol{z}\|_2 \right)}{\sqrt{K} \|\boldsymbol{z}\|_2} \cdot \lim_{K \to 0^+} \boldsymbol{z}$$

$$= 1 \cdot 1 \cdot (\boldsymbol{x} - vy) = \boldsymbol{x} - \boldsymbol{y},$$

due to several applications of the theorem of limits of composed functions, product rule for limits, Lemma A.16, and the fact that

$$\lim_{\alpha \to 0} \frac{\tan^{-1}(\sqrt{\alpha} a)}{\sqrt{\alpha}} = a.$$

The negative case $K < 0$ is analogous. $\qquad \square$

**Lemma A.19** (gyr$[\boldsymbol{x}, \boldsymbol{y}]\boldsymbol{v}$ converges to $\boldsymbol{v}$ as $K \to 0$)**.** *For all $\boldsymbol{x}, \boldsymbol{y}$ in the Poincaré ball $\mathbb{P}_K^n$ or the projected sphere $\mathbb{D}_K^n$ and $\boldsymbol{v} \in \mathcal{T}_{\boldsymbol{x}} \mathcal{M}$, it holds that*

$$\lim_{K \to 0} \mathrm{gyr}[\boldsymbol{x}, \boldsymbol{y}]\boldsymbol{x} = \boldsymbol{v},$$

*hence gyration converges to an identity function.*

*Proof.*

$$\lim_{K \to 0} \text{gyr}[\boldsymbol{x}, \boldsymbol{y}]\boldsymbol{v} = \lim_{K \to 0} \left( \ominus_K (\boldsymbol{x} \oplus_K \boldsymbol{y}) \oplus_K (\boldsymbol{x} \oplus_K (\boldsymbol{y} \oplus_K \boldsymbol{v})) \right)$$
$$= -(\boldsymbol{x} + \boldsymbol{y}) + (\boldsymbol{x} + (\boldsymbol{y} + \boldsymbol{v}))$$
$$= -\boldsymbol{x} - \boldsymbol{y} + \boldsymbol{x} + \boldsymbol{y} + \boldsymbol{v} = \boldsymbol{v},$$

due to Theorem A.14 and the theorem of limits of composed functions. $\qquad\square$

**Theorem A.20** ($\text{PT}_{\boldsymbol{x} \to \boldsymbol{y}}^K(\boldsymbol{v})$ converges to $\boldsymbol{v}$ as $K \to 0$). *For all $\boldsymbol{x}, \boldsymbol{y}$ in the Poincaré ball $\mathbb{P}_K^n$ or the projected sphere $\mathbb{D}_K^n$ and $\boldsymbol{v} \in \mathcal{T}_{\boldsymbol{x}}\mathcal{M}$, it holds that*

$$\lim_{K \to 0} PT_{\boldsymbol{x} \to \boldsymbol{y}}^K(\boldsymbol{v}) = \boldsymbol{v}.$$

*Proof.*

$$\lim_{K \to 0} PT_{\boldsymbol{x} \to \boldsymbol{y}}^K(\boldsymbol{v}) = \lim_{K \to 0} \left( \frac{\lambda_{\boldsymbol{x}}^K}{\lambda_{\boldsymbol{y}}^K} \text{gyr}[\boldsymbol{y}, -\boldsymbol{x}]\boldsymbol{v} \right)$$
$$= \lim_{K \to 0} \underbrace{\frac{\lambda_{\boldsymbol{x}}^K}{\lambda_{\boldsymbol{y}}^K}}_{\xrightarrow{K \to 0} 1} \cdot \lim_{K \to 0} \underbrace{\text{gyr}[\boldsymbol{y}, -\boldsymbol{x}]\boldsymbol{v}}_{\xrightarrow{K \to 0} \boldsymbol{v}}$$
$$= \boldsymbol{v},$$

due to the product rule for limits, Lemma A.16, and Lemma A.19. $\qquad\square$

# B  PROBABILITY DETAILS

## B.1  WRAPPED NORMAL DISTRIBUTIONS

**Theorem B.1** (Probability density function of $\mathcal{WN}(\boldsymbol{z}; \boldsymbol{\mu}, \boldsymbol{\Sigma})$ in $\mathbb{H}_K^n$).

$$\log \mathcal{WN}(\boldsymbol{z}; \boldsymbol{\mu}, \boldsymbol{\Sigma}) = \log \mathcal{N}(\boldsymbol{v}; \boldsymbol{0}, \boldsymbol{\Sigma}) - (n-1) \log \left( \frac{R \sinh \left( \frac{\|\boldsymbol{u}\|_{\mathcal{L}}}{R} \right)}{\|\boldsymbol{u}\|_{\mathcal{L}}} \right),$$

*where $\boldsymbol{u} = \log_{\boldsymbol{\mu}}^K(\boldsymbol{z})$, $\boldsymbol{v} = \text{PT}_{\boldsymbol{\mu} \to \boldsymbol{\mu}_0}^K(\boldsymbol{u})$, and $R = 1/\sqrt{-K}$.*

*Proof.* This was shown for the case $K = 1$ by Nagano et al. (2019). The difference is that we do not assume unitary radius $R = 1 = 1/\sqrt{-K}$. Hence, our tranformation function has the form $f = \exp_{\boldsymbol{\mu}}^K \circ \text{PT}_{\boldsymbol{\mu}_0 \to \boldsymbol{\mu}}^K$, and $f^{-1} = \text{PT}_{\boldsymbol{\mu} \to \boldsymbol{\mu}_0}^K \circ \log_{\boldsymbol{\mu}}^K$.

The derivative of parallel transport $PT_{\boldsymbol{x} \to \boldsymbol{y}}^K(\boldsymbol{v})$ for any $\boldsymbol{x}, \boldsymbol{y} \in \mathbb{H}_K^n$ and $\boldsymbol{v} \in \mathcal{T}_{\boldsymbol{x}}\mathbb{H}_K^n$ is a map $\text{d}\,\text{PT}_{\boldsymbol{x} \to \boldsymbol{y}}^K(\boldsymbol{v}) : \mathcal{T}_{\boldsymbol{v}}(\mathcal{T}_{\boldsymbol{x}}\mathbb{H}_K^n)$. Using the orthonormal basis (with respect to the Lorentz product) $\{\boldsymbol{\xi}_1, \dots \boldsymbol{\xi}_n\}$, we can compute the determinant by computing the change with respect to each basis vector.

$$\text{d}\,\text{PT}_{\boldsymbol{x} \to \boldsymbol{y}}^K(\boldsymbol{\xi}) = \frac{\partial}{\partial \epsilon}\bigg|_{\epsilon=0} \text{PT}_{\boldsymbol{x} \to \boldsymbol{y}}^K(\boldsymbol{v} + \epsilon\boldsymbol{\xi})$$
$$= \frac{\partial}{\partial \epsilon}\bigg|_{\epsilon=0} \left[ (\boldsymbol{v} + \epsilon\boldsymbol{\xi}) + \frac{\langle \boldsymbol{y}, \boldsymbol{v} + \epsilon\boldsymbol{\xi} \rangle_{\mathcal{L}}}{R^2 - \langle \boldsymbol{x}, \boldsymbol{y} \rangle_{\mathcal{L}}} (\boldsymbol{x} + \boldsymbol{y}) \right]$$
$$= \left[ \boldsymbol{\xi} + \frac{\langle \boldsymbol{y}, \boldsymbol{\xi} \rangle_{\mathcal{L}}}{R^2 - \langle \boldsymbol{x}, \boldsymbol{y} \rangle_{\mathcal{L}}} (\boldsymbol{x} + \boldsymbol{y}) \right]_{\epsilon=0}$$
$$= \text{PT}_{\boldsymbol{x} \to \boldsymbol{y}}^K(\boldsymbol{\xi}).$$

Since parallel transport preserves norms and vectors in the orthonormal basis have norm 1, the change is $\left\| \text{d}\,\text{PT}_{\boldsymbol{x} \to \boldsymbol{y}}^K(\boldsymbol{\xi}) \right\|_{\mathcal{L}} = \left\| \text{PT}_{\boldsymbol{x} \to \boldsymbol{y}}^K(\boldsymbol{\xi}) \right\|_{\mathcal{L}} = 1$.

For computing the determinant of the exponential map Jacobian, we choose the orthonormal basis $\{\boldsymbol{\xi}_1 = \boldsymbol{u}/\|\boldsymbol{u}\|_{\mathcal{L}}, \boldsymbol{\xi}_2, \dots, \boldsymbol{\xi}_n\}$, where we just completed the basis based on the first vector. We again look at the change with respect to each basis vector. For the basis vector $\boldsymbol{\xi}_1$:

$$\mathrm{d}\exp_{\boldsymbol{x}}^K(\boldsymbol{\xi}_1) =$$

$$= \left.\frac{\partial}{\partial\epsilon}\right|_{\epsilon=0} \exp_{\boldsymbol{x}}^K\left(\boldsymbol{u} + \epsilon\frac{\boldsymbol{u}}{\|\boldsymbol{u}\|_{\mathcal{L}}}\right)$$

$$= \left.\frac{\partial}{\partial\epsilon}\right|_{\epsilon=0}\left[\cosh\left(\frac{|\|\boldsymbol{u}\|_{\mathcal{L}} + \epsilon|}{R}\right)\boldsymbol{x} + \frac{R\sinh\left(\frac{|\|\boldsymbol{u}\|_{\mathcal{L}}+\epsilon|}{R}\right)}{\|\boldsymbol{u}\|_{\mathcal{L}}\,|\|\boldsymbol{u}\|_{\mathcal{L}} + \epsilon|}\left(\|\boldsymbol{u}\|_{\mathcal{L}} + \epsilon\right)\boldsymbol{u}\right]$$

$$= \left[\frac{(\|\boldsymbol{u}\|_{\mathcal{L}} + \epsilon)\sinh\left(\frac{|\|\boldsymbol{u}\|_{\mathcal{L}}+\epsilon|}{R}\right)}{R|\|\boldsymbol{u}\|_{\mathcal{L}} + \epsilon|}\boldsymbol{x} + \frac{\cosh\left(\frac{|\|\boldsymbol{u}\|_{\mathcal{L}}+\epsilon|}{R}\right)}{\|\boldsymbol{u}\|_{\mathcal{L}}}\boldsymbol{u}\right]_{\epsilon=0}$$

$$= \sinh\left(\frac{\|\boldsymbol{u}\|_{\mathcal{L}}}{R}\right)\frac{\boldsymbol{x}}{R} + \cosh\left(\frac{\|\boldsymbol{u}\|_{\mathcal{L}}}{R}\right)\frac{\boldsymbol{u}}{\|\boldsymbol{u}\|_{\mathcal{L}}},$$

where the second equality is due to

$$\left\|\boldsymbol{u} + \epsilon\frac{\boldsymbol{u}}{\|\boldsymbol{u}\|_{\mathcal{L}}}\right\|_{\mathcal{L}} = \left\|\left(1 + \frac{\epsilon}{\|\boldsymbol{u}\|_{\mathcal{L}}}\right)\boldsymbol{u}\right\|_{\mathcal{L}} = \left|1 + \frac{\epsilon}{\|\boldsymbol{u}\|_{\mathcal{L}}}\right|\|\boldsymbol{u}\|_{\mathcal{L}} = |\|\boldsymbol{u}\|_{\mathcal{L}} + \epsilon|.$$

For every other basis vector $\boldsymbol{\xi}_k$ where $k > 1$:

$$\mathrm{d}\exp_{\boldsymbol{x}}^K(\boldsymbol{\xi}) =$$

$$= \left.\frac{\partial}{\partial\epsilon}\right|_{\epsilon=0} \exp_{\boldsymbol{x}}^K(\boldsymbol{u} + \epsilon\boldsymbol{\xi})$$

$$= \left.\frac{\partial}{\partial\epsilon}\right|_{\epsilon=0}\left[\cosh\left(\frac{\|\boldsymbol{u}+\epsilon\boldsymbol{\xi}\|_{\mathcal{L}}}{R}\right)\boldsymbol{x} + \frac{R\sinh\left(\frac{\|\boldsymbol{u}+\epsilon\boldsymbol{\xi}\|_{\mathcal{L}}}{R}\right)}{\|\boldsymbol{u}+\epsilon\boldsymbol{\xi}\|_{\mathcal{L}}}(\boldsymbol{u} + \epsilon\boldsymbol{\xi})\right]$$

$$= \left.\frac{\partial}{\partial\epsilon}\right|_{\epsilon=0}\left[\cosh\left(\frac{\sqrt{\|\boldsymbol{u}\|_{\mathcal{L}}^2 + \epsilon^2}}{R}\right)\boldsymbol{x} + \frac{R\sinh\left(\frac{\sqrt{\|\boldsymbol{u}\|_{\mathcal{L}}^2 + \epsilon^2}}{R}\right)}{\sqrt{\|\boldsymbol{u}\|_{\mathcal{L}}^2 + \epsilon^2}}(\boldsymbol{u} + \epsilon\boldsymbol{\xi})\right]$$

$$= \left[\frac{\epsilon\cosh\left(\frac{\sqrt{\|\boldsymbol{u}\|_{\mathcal{L}}^2 + \epsilon^2}}{R}\right)}{\|\boldsymbol{u}\|_{\mathcal{L}}^2 + \epsilon^2}(\boldsymbol{u} + \epsilon\boldsymbol{\xi})\right.$$

$$\left.+ \frac{(R^2\|\boldsymbol{u}\|_{\mathcal{L}}^2\boldsymbol{\xi} - R^2\epsilon\boldsymbol{u} + \epsilon(\|\boldsymbol{u}\|_{\mathcal{L}}^2 + \epsilon^2)\boldsymbol{x})\sinh\left(\frac{\sqrt{\|\boldsymbol{u}\|_{\mathcal{L}}^2 + \epsilon^2}}{R}\right)}{R(\|\boldsymbol{u}\|_{\mathcal{L}}^2 + \epsilon^2)^{3/2}}\right]_{\epsilon=0}$$

$$= \frac{R^2\|\boldsymbol{u}\|_{\mathcal{L}}^2\sinh\left(\frac{\|\boldsymbol{u}\|_{\mathcal{L}}}{R}\right)}{R(\|\boldsymbol{u}\|_{\mathcal{L}}^2)^{3/2}}\boldsymbol{\xi} = \frac{R\sinh\left(\frac{\|\boldsymbol{u}\|_{\mathcal{L}}}{R}\right)}{\|\boldsymbol{u}\|_{\mathcal{L}}}\boldsymbol{\xi},$$

where the third equality holds because

$$\|\boldsymbol{u} + \epsilon\boldsymbol{\xi}\|_{\mathcal{L}}^2 = \|\boldsymbol{u}\|_{\mathcal{L}}^2 + \epsilon^2\|\boldsymbol{\xi}\|_{\mathcal{L}}^2 - 2\langle\boldsymbol{u}, \epsilon\boldsymbol{\xi}\rangle_{\mathcal{L}}$$
$$= \|\boldsymbol{u}\|_{\mathcal{L}}^2 + \epsilon^2 - 2\epsilon\langle\boldsymbol{u}, \boldsymbol{\xi}\rangle_{\mathcal{L}}$$
$$= \|\boldsymbol{u}\|_{\mathcal{L}}^2 + \epsilon^2,$$

where the last equality relies on the fact that the basis is orthogonal, and $\boldsymbol{u}$ is parallel to $\boldsymbol{\xi}_1 = \boldsymbol{u}/\|\boldsymbol{u}\|_{\mathcal{L}}$, hence it is orthogonal to all the other basis vectors.

Because the basis is orthonormal the determinant is a product of the norms of the computed change for each basis vector. Therefore,

$$\det\left(\frac{\partial \mathrm{PT}_{\boldsymbol{x}\to\boldsymbol{y}}(\boldsymbol{v})}{\partial \boldsymbol{v}}\right) = 1^n = 1.$$

Additionally, the following two properties hold:

$$\left\|\mathrm{d}\exp_{\boldsymbol{x}}^K\left(\frac{\boldsymbol{u}}{\|\boldsymbol{u}\|_{\mathcal{L}}}\right)\right\|_{\mathcal{L}}^2 = \left\|\sinh\left(\frac{\|\boldsymbol{u}\|_{\mathcal{L}}}{R}\right)\frac{\boldsymbol{x}}{R} + \cosh\left(\frac{\|\boldsymbol{u}\|_{\mathcal{L}}}{R}\right)\frac{\boldsymbol{u}}{\|\boldsymbol{u}\|_{\mathcal{L}}}\right\|_{\mathcal{L}}^2$$

$$= \sinh^2\left(\frac{\|\boldsymbol{u}\|_{\mathcal{L}}}{R}\right)\frac{\|\boldsymbol{x}\|_{\mathcal{L}}^2}{R^2} + \cosh^2\left(\frac{\|\boldsymbol{u}\|_{\mathcal{L}}}{R}\right)\frac{\|\boldsymbol{u}\|_{\mathcal{L}}^2}{\|\boldsymbol{u}\|_{\mathcal{L}}^2}$$

$$= -\sinh^2\left(\frac{\|\boldsymbol{u}\|_{\mathcal{L}}}{R}\right) + \cosh^2\left(\frac{\|\boldsymbol{u}\|_{\mathcal{L}}}{R}\right) = 1.$$

and

$$\left\|\mathrm{d}\exp_{\boldsymbol{x}}^K(\boldsymbol{\xi})\right\|_{\mathcal{L}}^2 = \left\|\frac{R\sinh\left(\frac{\|\boldsymbol{u}\|_{\mathcal{L}}}{R}\right)}{\|\boldsymbol{u}\|_{\mathcal{L}}}\boldsymbol{\xi}\right\|_{\mathcal{L}}^2$$

$$= \frac{R^2\sinh^2\left(\frac{\|\boldsymbol{u}\|_{\mathcal{L}}}{R}\right)}{\|\boldsymbol{u}\|_{\mathcal{L}}^2}\|\boldsymbol{\xi}\|_{\mathcal{L}}^2$$

$$= \frac{R^2\sinh^2\left(\frac{\|\boldsymbol{u}\|_{\mathcal{L}}}{R}\right)}{\|\boldsymbol{u}\|_{\mathcal{L}}^2}.$$

Therefore, we obtain

$$\det\left(\frac{\partial \exp_{\boldsymbol{x}}^K(\boldsymbol{u})}{\partial \boldsymbol{u}}\right) = 1\cdot\left(\frac{R\sinh\left(\frac{\|\boldsymbol{u}\|_{\mathcal{L}}}{R}\right)}{\|\boldsymbol{u}\|_{\mathcal{L}}}\right)^{n-1}.$$

Finally,

$$\det\left(\frac{\partial f(\boldsymbol{v})}{\partial \boldsymbol{v}}\right) = \det\left(\frac{\partial \exp_{\boldsymbol{\mu}}^K(\boldsymbol{u})}{\partial \boldsymbol{u}}\right)\cdot\det\left(\frac{\partial \mathrm{PT}_{\boldsymbol{\mu}_0\to\boldsymbol{\mu}}^K(\boldsymbol{v})}{\partial \boldsymbol{v}}\right) = \left(\frac{R\sinh\left(\frac{\|\boldsymbol{u}\|_{\mathcal{L}}}{R}\right)}{\|\boldsymbol{u}\|_{\mathcal{L}}}\right)^{n-1}.$$

$\square$

**Theorem B.2** (Probability density function of $\mathcal{WN}(\boldsymbol{z};\boldsymbol{\mu},\boldsymbol{\Sigma})$ in $\mathbb{S}_K^n$).

$$\log\mathcal{WN}(\boldsymbol{z};\boldsymbol{\mu},\boldsymbol{\Sigma}) = \log\mathcal{N}(\boldsymbol{v};\boldsymbol{0},\boldsymbol{\Sigma}) - (n-1)\log\left(\frac{R\left|\sin\left(\frac{\|\boldsymbol{u}\|_2}{R}\right)\right|}{\|\boldsymbol{u}\|_2}\right),$$

where $\boldsymbol{u} = \log_{\boldsymbol{\mu}}^K(\boldsymbol{z})$, $\boldsymbol{v} = \mathrm{PT}_{\boldsymbol{\mu}\to\boldsymbol{\mu}_0}^K(\boldsymbol{u})$, and $R = 1/\sqrt{K}$.

*Proof.* The theorem is very similar to Theorem B.1. The difference is that in this one, our manifold changes from $\mathbb{H}_K^n$ to $\mathbb{S}_K^n$, hence $K > 0$. Our tranformation function has the form $f = \exp_{\boldsymbol{\mu}}^K \circ \mathrm{PT}_{\boldsymbol{\mu}_0\to\boldsymbol{\mu}}^K$, and $f^{-1} = \mathrm{PT}_{\boldsymbol{\mu}\to\boldsymbol{\mu}_0}^K \circ \log_{\boldsymbol{\mu}}^K$.

The derivative of parallel transport $PT_{\boldsymbol{x}\to\boldsymbol{y}}^K(\boldsymbol{v})$ for any $\boldsymbol{x},\boldsymbol{y} \in \mathbb{S}_K^n$ and $\boldsymbol{v} \in \mathcal{T}_{\boldsymbol{x}}\mathbb{S}_K^n$ is a map $\mathrm{d}\mathrm{PT}_{\boldsymbol{x}\to\boldsymbol{y}}^K(\boldsymbol{v}) : \mathcal{T}_{\boldsymbol{v}}(\mathcal{T}_{\boldsymbol{x}}\mathbb{S}_K^n)$. Using the orthonormal basis (with respect to the Lorentz product) $\{\boldsymbol{\xi}_1,\ldots\boldsymbol{\xi}_n\}$, we can compute the determinant by computing the change with respect to each basis vector.

$$\mathrm{d}\mathrm{PT}_{\boldsymbol{x}\to\boldsymbol{y}}^K(\boldsymbol{\xi}) = \left.\frac{\partial}{\partial\epsilon}\right|_{\epsilon=0}\mathrm{PT}_{\boldsymbol{x}\to\boldsymbol{y}}^K(\boldsymbol{v}+\epsilon\boldsymbol{\xi})$$

$$
= \frac{\partial}{\partial \epsilon}\Big|_{\epsilon=0} \left[ (\boldsymbol{v} + \epsilon \boldsymbol{\xi}) - \frac{\langle \boldsymbol{y}, \boldsymbol{v} + \epsilon \boldsymbol{\xi} \rangle_2}{R^2 + \langle \boldsymbol{x}, \boldsymbol{y} \rangle_2}(\boldsymbol{x} + \boldsymbol{y}) \right]
$$

$$
= \left[ \boldsymbol{\xi} - \frac{\langle \boldsymbol{y}, \boldsymbol{\xi} \rangle_2}{R^2 + \langle \boldsymbol{x}, \boldsymbol{y} \rangle_2}(\boldsymbol{x} + \boldsymbol{y}) \right]_{\epsilon=0}
$$

$$
= \mathrm{PT}_{\boldsymbol{x} \to \boldsymbol{y}}^K(\boldsymbol{\xi}).
$$

Since parallel transport preserves norms and vectors in the orthonormal basis have norm 1, the change is $\left\| \mathrm{d}\, \mathrm{PT}_{\boldsymbol{x} \to \boldsymbol{y}}^K(\boldsymbol{\xi}) \right\|_2 = \left\| \mathrm{PT}_{\boldsymbol{x} \to \boldsymbol{y}}^K(\boldsymbol{\xi}) \right\|_2 = 1$.

For computing the determinant of the exponential map Jacobian, we choose the orthonormal basis $\{\boldsymbol{\xi}_1 = \boldsymbol{u}/\left\| \boldsymbol{u} \right\|_2, \boldsymbol{\xi}_2, \ldots, \boldsymbol{\xi}_n\}$, where we just completed the basis based on the first vector. We again look at the change with respect to each basis vector. For the basis vector $\boldsymbol{\xi}_1$:

$$
\mathrm{d}\exp_{\boldsymbol{x}}^K(\boldsymbol{\xi}_1) =
$$

$$
= \frac{\partial}{\partial \epsilon}\Big|_{\epsilon=0} \exp_{\boldsymbol{x}}^K\left( \boldsymbol{u} + \epsilon \frac{\boldsymbol{u}}{\left\| \boldsymbol{u} \right\|_2} \right)
$$

$$
= \frac{\partial}{\partial \epsilon}\Big|_{\epsilon=0} \left[ \cos\left( \frac{\left| \left\| \boldsymbol{u} \right\|_2 + \epsilon \right|}{R} \right) \boldsymbol{x} + \frac{R \sin\left( \frac{\left| \left\| \boldsymbol{u} \right\|_2 + \epsilon \right|}{R} \right)}{\left\| \boldsymbol{u} \right\|_2 \left| \left\| \boldsymbol{u} \right\|_2 + \epsilon \right|} \left( \left\| \boldsymbol{u} \right\|_2 + \epsilon \right) \boldsymbol{u} \right]
$$

$$
= \left[ -\frac{\left( \left\| \boldsymbol{u} \right\|_2 + \epsilon \right) \sin\left( \frac{\left| \left\| \boldsymbol{u} \right\|_2 + \epsilon \right|}{R} \right)}{R \left| \left\| \boldsymbol{u} \right\|_2 + \epsilon \right|} \boldsymbol{x} + \frac{\cos\left( \frac{\left| \left\| \boldsymbol{u} \right\|_2 + \epsilon \right|}{R} \right)}{\left\| \boldsymbol{u} \right\|_2} \boldsymbol{u} \right]_{\epsilon=0}
$$

$$
= \cos\left( \frac{\left\| \boldsymbol{u} \right\|_2}{R} \right) \frac{\boldsymbol{u}}{\left\| \boldsymbol{u} \right\|_2} - \sin\left( \frac{\left\| \boldsymbol{u} \right\|_2}{R} \right) \frac{\boldsymbol{x}}{R},
$$

where the second equality is due to

$$
\left\| \boldsymbol{u} + \epsilon \frac{\boldsymbol{u}}{\left\| \boldsymbol{u} \right\|_2} \right\|_2 = \left\| \left( 1 + \frac{\epsilon}{\left\| \boldsymbol{u} \right\|_2} \right) \boldsymbol{u} \right\|_2 = \left| 1 + \frac{\epsilon}{\left\| \boldsymbol{u} \right\|_2} \right| \left\| \boldsymbol{u} \right\|_2 = \left| \left\| \boldsymbol{u} \right\|_2 + \epsilon \right|.
$$

For every other basis vector $\boldsymbol{\xi}_k$ where $k > 1$:

$$
\mathrm{d}\exp_{\boldsymbol{x}}^K(\boldsymbol{\xi}) =
$$

$$
= \frac{\partial}{\partial \epsilon}\Big|_{\epsilon=0} \exp_{\boldsymbol{x}}^K(\boldsymbol{u} + \epsilon \boldsymbol{\xi})
$$

$$
= \frac{\partial}{\partial \epsilon}\Big|_{\epsilon=0} \left[ \cos\left( \frac{\left\| \boldsymbol{u} + \epsilon \boldsymbol{\xi} \right\|_2}{R} \right) \boldsymbol{x} + \frac{R \sin\left( \frac{\left\| \boldsymbol{u} + \epsilon \boldsymbol{\xi} \right\|_2}{R} \right)}{\left\| \boldsymbol{u} + \epsilon \boldsymbol{\xi} \right\|_2}(\boldsymbol{u} + \epsilon \boldsymbol{\xi}) \right]
$$

$$
= \frac{\partial}{\partial \epsilon}\Big|_{\epsilon=0} \left[ \cos\left( \frac{\sqrt{\left\| \boldsymbol{u} \right\|_2^2 + \epsilon^2}}{R} \right) \boldsymbol{x} + \frac{R \sin\left( \frac{\sqrt{\left\| \boldsymbol{u} \right\|_2^2 + \epsilon^2}}{R} \right)}{\sqrt{\left\| \boldsymbol{u} \right\|_2^2 + \epsilon^2}}(\boldsymbol{u} + \epsilon \boldsymbol{\xi}) \right]
$$

$$
= \Bigg[ \frac{\epsilon \cos\left( \frac{\sqrt{\left\| \boldsymbol{u} \right\|_2^2 + \epsilon^2}}{R} \right)}{\left\| \boldsymbol{u} \right\|_2^2 + \epsilon^2}(\boldsymbol{u} + \epsilon \boldsymbol{\xi})
$$

$$
+ \frac{(R^2 \left\| \boldsymbol{u} \right\|_2^2 \boldsymbol{\xi} - R^2 \epsilon \boldsymbol{u} - \epsilon(\left\| \boldsymbol{u} \right\|_2^2 + \epsilon^2)\boldsymbol{x}) \sin\left( \frac{\sqrt{\left\| \boldsymbol{u} \right\|_2^2 + \epsilon^2}}{R} \right)}{R(\left\| \boldsymbol{u} \right\|_2^2 + \epsilon^2)^{3/2}} \Bigg]_{\epsilon=0}
$$

$$
= \frac{R^2 \left\| \boldsymbol{u} \right\|_2^2 \sin\left( \frac{\left\| \boldsymbol{u} \right\|_2}{R} \right)}{R(\left\| \boldsymbol{u} \right\|_2^2)^{3/2}} \boldsymbol{\xi} = \frac{R \sin\left( \frac{\left\| \boldsymbol{u} \right\|_2}{R} \right)}{\left\| \boldsymbol{u} \right\|_2} \boldsymbol{\xi},
$$

where the third equality holds because

$$
\begin{aligned}
\|\boldsymbol{u} + \epsilon\boldsymbol{\xi}\|_2^2 &= \|\boldsymbol{u}\|_2^2 + \epsilon^2 \|\boldsymbol{\xi}\|_2^2 - 2\langle \boldsymbol{u}, \epsilon\boldsymbol{\xi}\rangle_2 \\
&= \|\boldsymbol{u}\|_2^2 + \epsilon^2 - 2\epsilon\langle \boldsymbol{u}, \boldsymbol{\xi}\rangle_2 \\
&= \|\boldsymbol{u}\|_2^2 + \epsilon^2,
\end{aligned}
$$

where the last equality relies on the fact that the basis is orthogonal, and $\boldsymbol{u}$ is parallel to $\boldsymbol{\xi}_1 = \boldsymbol{u}/\|\boldsymbol{u}\|_2$, hence it is orthogonal to all the other basis vectors.

Because the basis is orthonormal the determinant is a product of the norms of the computed change for each basis vector. Therefore,

$$
\det\left(\frac{\partial \operatorname{PT}_{\boldsymbol{x}\to\boldsymbol{y}}(\boldsymbol{v})}{\partial \boldsymbol{v}}\right) = 1^n = 1.
$$

Additionally, the following two properties hold:

$$
\begin{aligned}
\left\|\operatorname{d}\exp_{\boldsymbol{x}}^K\left(\frac{\boldsymbol{u}}{\|\boldsymbol{u}\|_2}\right)\right\|_2^2 &= \left\|\cos\left(\frac{\|\boldsymbol{u}\|_2}{R}\right)\frac{\boldsymbol{u}}{\|\boldsymbol{u}\|_2} - \sin\left(\frac{\|\boldsymbol{u}\|_2}{R}\right)\frac{\boldsymbol{x}}{R}\right\|_2^2 \\
&= \sin^2\left(\frac{\|\boldsymbol{u}\|_2}{R}\right)\frac{\|\boldsymbol{x}\|_2^2}{R^2} + \cos^2\left(\frac{\|\boldsymbol{u}\|_2}{R}\right)\frac{\|\boldsymbol{u}\|_2^2}{\|\boldsymbol{u}\|_2^2} \\
&= \sin^2\left(\frac{\|\boldsymbol{u}\|_2}{R}\right) + \cos^2\left(\frac{\|\boldsymbol{u}\|_2}{R}\right) = 1.
\end{aligned}
$$

and

$$
\begin{aligned}
\left\|\operatorname{d}\exp_{\boldsymbol{x}}^K(\boldsymbol{\xi})\right\|_2^2 &= \left\|\frac{R\sin\left(\frac{\|\boldsymbol{u}\|_2}{R}\right)}{\|\boldsymbol{u}\|_2}\boldsymbol{\xi}\right\|_2^2 \\
&= \frac{R^2\sin^2\left(\frac{\|\boldsymbol{u}\|_2}{R}\right)}{\|\boldsymbol{u}\|_2^2}\|\boldsymbol{\xi}\|_2^2 \\
&= \frac{R^2\sin^2\left(\frac{\|\boldsymbol{u}\|_2}{R}\right)}{\|\boldsymbol{u}\|_2^2}.
\end{aligned}
$$

Therefore, we obtain

$$
\det\left(\frac{\partial\exp_{\boldsymbol{x}}^K(\boldsymbol{u})}{\partial\boldsymbol{u}}\right) = 1\cdot\left(\frac{R\left|\sin\left(\frac{\|\boldsymbol{u}\|_2}{R}\right)\right|}{\|\boldsymbol{u}\|_2}\right)^{n-1}.
$$

Finally,

$$
\det\left(\frac{\partial f(\boldsymbol{v})}{\partial\boldsymbol{v}}\right) = \det\left(\frac{\partial\exp_{\boldsymbol{\mu}}^K(\boldsymbol{u})}{\partial\boldsymbol{u}}\right)\cdot\det\left(\frac{\partial\operatorname{PT}_{\boldsymbol{\mu}_0\to\boldsymbol{\mu}}^K(\boldsymbol{v})}{\partial\boldsymbol{v}}\right) = \left(\frac{R\left|\sin\left(\frac{\|\boldsymbol{u}\|_2}{R}\right)\right|}{\|\boldsymbol{u}\|_2}\right)^{n-1}.
$$

$\square$

**Theorem B.3** (Probability density function of $\mathcal{WN}(\boldsymbol{z};\boldsymbol{\mu},\boldsymbol{\Sigma})$ in $\mathbb{P}_K^n$).

$$
\log\mathcal{WN}_{\mathbb{P}_K^n}(\boldsymbol{z};\boldsymbol{\mu},\boldsymbol{\Sigma}) = \log\mathcal{WN}_{\mathbb{H}_K^n}(\rho_K^{-1}(\boldsymbol{z});\rho_K^{-1}(\boldsymbol{\mu}),\boldsymbol{\Sigma}).
$$

*Proof.* Follows from Theorem B.1 and A.3.

Also proven by (Mathieu et al., 2019) in a slightly different form for a scalar scale parameter $\mathcal{WN}(\boldsymbol{z};\boldsymbol{\mu},\sigma^2\boldsymbol{I})$. Given

$$
\log\mathcal{N}(\boldsymbol{z};\boldsymbol{\mu},\sigma^2\boldsymbol{I}) = -\frac{d_{\mathbb{E}}(\boldsymbol{\mu},\boldsymbol{z})^2}{2\sigma^2} - \frac{n}{2}\log\left(2\pi\sigma^2\right)
$$

$$\log \mathcal{WN}(\boldsymbol{z}; \boldsymbol{\mu}, \sigma^2 \boldsymbol{I}) = - \frac{d_{\mathbb{P}}^K(\boldsymbol{\mu}, \boldsymbol{z})^2}{2\sigma^2} - \frac{n}{2} \log\left(2\pi\sigma^2\right)$$
$$+ (n-1) \log\left(\frac{\sqrt{-K} d_{\mathbb{P}}^K(\boldsymbol{\mu}, \boldsymbol{z})}{\sinh(\sqrt{-K} d_{\mathbb{P}}^K(\boldsymbol{\mu}, \boldsymbol{z}))}\right).$$

$\square$

**Theorem B.4** (Probability density function of $\mathcal{WN}(\boldsymbol{z}; \boldsymbol{\mu}, \boldsymbol{\Sigma})$ in $\mathbb{D}_K^n$)**.**

$$\log \mathcal{WN}_{\mathbb{D}_K^n}(\boldsymbol{z}; \boldsymbol{\mu}, \boldsymbol{\Sigma}) = \log \mathcal{WN}_{\mathbb{S}_K^n}(\rho_K^{-1}(\boldsymbol{z}); \rho_K^{-1}(\boldsymbol{\mu}), \boldsymbol{\Sigma}).$$

*Proof.* Follows from Theorem B.2 and A.3 adapted from $\mathbb{P}$ to $\mathbb{D}$. $\square$

## C   RELATED WORK

**Universal models of geometry**   Duality between spaces of constant curvature was first noticed by Lambert (1770), and later gave rise to various theorems that have the same or similar forms in all three geometries, like the law of sines (Bolyai, 1832)

$$\frac{\sin A}{p_K(a)} = \frac{\sin B}{p_K(b)} = \frac{\sin C}{p_K(c)},$$

where $p_K(r) = 2\pi \sin_K(r)$ denotes the circumference of a circle of radius $r$ in a space of constant curvature $K$, and

$$\sin_K(x) = x - \frac{Kx^3}{3!} + \frac{K^2 x^5}{5!} - \ldots = \sum_{i=0}^{\infty} \frac{(-1)^i K^i x^{2i+1}}{(2i+1)!}.$$

Other unified formulas for the law of cosines, or recently, a unified Pythagorean theorem has also been proposed (Foote, 2017):

$$A(c) = A(a) + A(b) - \frac{K}{2\pi} A(a)A(b),$$

where $A(r)$ is the area of a circle of radius $r$ in a space of constant curvature $K$. Unfortunately, in this formulation $A(r)$ still depends on the sign of $K$ w.r.t. the choice of trigonometric functions in its definition.

There also exist approaches defining a universal geometry of constant curvature spaces. Li et al. (2001, Chapter 4) define a unified model of all three geometries using the null cone (light cone) of a Minkowski space. The term "Minkowski space" comes from special relativity and is usually denoted as $\mathbb{R}^{1,n}$, similar to the ambient space of what we defined as $\mathbb{H}^n$, with the Lorentz scalar product $\langle \cdot, \cdot \rangle_{\mathcal{L}}$. The hyperboloid $\mathbb{H}^n$ corresponds to the positive (upper, future) null cone of $\mathbb{R}^{1,n}$. All the other models can be defined in this space using the appropriate stereographic projections and pulling back the metric onto the specific sub-manifold. Unfortunately, we found the formalism not useful for our application, apart from providing a very interesting theoretical connection among the models.

**Concurrent VAE approaches**   The variational autoencoder was originally proposed in Kingma & Welling (2014) and concurrently in Rezende et al. (2014). One of the most common improvements on the VAE in practice is the choice of the encoder and decoder maps, ranging from linear parametrizations of the posterior to feed-forward neural networks, convolutional neural networks, etc. For different data domains, extensions like the GraphVAE (Simonovsky & Komodakis, 2018) using graph convolutional neural networks for the encoder and decoder were proposed.

The basic VAE framework was mostly improved upon by using autoregressive flows (Chen et al., 2014) or small changes to the ELBO loss function (Matthey et al., 2017; Burda et al., 2016). An important work in this area is $\beta$-VAE, which adds a simple scalar multiplicative constant to the KL divergence term in the ELBO, and has shown to improve both sample quality and (if $\beta > 1$) disentanglement of different dimensions in the latent representation. For more information on disentanglement, see Locatello et al. (2018).

**Geometric deep learning**   One of the emerging trends in deep learning has been to leverage non-Euclidean geometry to learn representations, originally emerging from knowledge-base and graph representation learning (Bronstein et al., 2017).

Recently, several approaches to learning representations in Euclidean spaces have been generalized to non-Euclidean spaces (Dhingra et al., 2018; Ganea et al., 2018b; Nickel & Kiela, 2017). Since then, this research direction has grown immensely and accumulated more approaches, mostly for hyperbolic spaces, like Ganea et al. (2018a); Nickel & Kiela (2018); Tifrea et al. (2019); Law et al. (2019). Similarly, spherical spaces have also been leveraged for learning non-Euclidean representations (Batmanghelich et al., 2016; Wilson & Hancock, 2010).

To be able to learn representations in these spaces, new Riemannian optimization methods were required as well (Wilson & Leimeister, 2018; Bonnabel, 2013; Bécigneul & Ganea, 2019).

The generalization to products of constant curvature Riemannian manifolds is only natural and has been proposed by Gu et al. (2019). They evaluated their approach by directly optimizing a distance-based loss function using Riemannian optimization in products of spaces on graph reconstruction and word analogy tasks, in both cases reaping the benefits of non-Euclidean geometry, especially when learning lower-dimensional representations. Further use of product spaces with constant curvature components to train Graph Convolutional Networks was concurrently with this work done by Bachmann et al. (2020).

**Geometry in VAEs**   One of the first attempts at leveraging geometry in VAEs was Arvanitidis et al. (2018). They examine how a Euclidean VAE benefits both in sample quality and latent representation distribution quality when employing a non-Euclidean Riemannian metric in the latent space using kernel transformations.

Hence, a potential improvement area of VAEs could be the choice of the posterior family and prior distribution. However, the Gaussian (Normal) distribution works very well in practice, as it is the maximum entropy probability distribution with a known variance, and imposes no constraints on higher-order moments (skewness, kurtosis, etc.) of the distribution. Recently, non-Euclidean geometry has been used in learning variational autoencoders as well. Generalizing Normal distributions to these spaces is in general non-trivial..

Two similar approaches, Davidson et al. (2018) and Xu & Durrett (2018), used the von Mises-Fischer distribution on the unit hypersphere to generalize VAEs to spherical spaces. The von Mises-Fischer distribution is again a maximum entropy probability distribution on the unit hypersphere, but only has a spherical covariance parameter, which makes it less general than a Gaussian distribution.

Conversely, two approaches, Mathieu et al. (2019) and Nagano et al. (2019), have generalized VAEs to hyperbolic spaces – both the Poincaré ball and the hyperboloid, respectively. They both adopt a non-maximum entropy probability distribution called the Wrapped Normal. Additionally, Mathieu et al. (2019) also derive the Riemannian Normal, which is a maximum entropy distribution on the Poincaré disk, but in practice performs similar to the Wrapped Normal, especially in higher dimensions.

Our approach generalizes on the afore-mentioned geometrical VAE work, by employing a "products of spaces" approach similar to Gu et al. (2019) and unifying the different approaches into a single framework for all spaces of constant curvature.

## D   EXTENDED FUTURE WORK

Even though we have shown that one can approximate the true posterior very well with Normal-like distributions in Riemannian manifolds of constant curvature, there remain several promising directions of explorations.

First of all, an interesting extension of this work would be to try mixed-curvature VAEs on graph data, e.g. link prediction on social networks, as some of our models might be well suited for sparse and structured data. Another very beneficial extension would be to investigate why the obtained results have a relatively big variance across runs and try to reduce it. However, this is a problem that affects the Euclidean VAE as well, even if not as flagrantly.

Secondly, we have empirically noticed that it seems to be significantly harder to optimize our models in spherical spaces – they seem more prone to divergence. In discussions, other researchers have also observed similar behavior, but a more thorough investigation is not available at the moment. We have side-stepped some optimization problems by introducing products of spaces – previously, it has been reported that both spherical and hyperbolic VAEs generally do not scale well to dimensions greater than 20 or 40. For those cases, we could successfully optimize a subdivided space $(\mathbb{S}^2)^{36}$ instead of one big manifold $\mathbb{S}^{72}$. However, that also does not seem to be a conclusive rule. Especially in higher dimensions, we have noticed that our VAEs $(\mathbb{S}^2)^{36}$ with learnable curvature and $\mathbb{D}_1^{72}$ with fixed curvature seem to consistently diverge. In a few cases $\mathbb{S}^{72}$ with fixed curvature and even the product $(\mathbb{E}^2)^{12} \times (\mathbb{H}^2)^{12} \times (\mathbb{S}^2)^{12}$ with learnable curvature seemed to diverge quite often as well.

The most promising future direction seems to be the use of "Normalizing Flows" for variational inference as presented by Rezende & Mohamed (2015) and Gemici et al. (2016). More recently, it was also combined with "autoregressive flows" in Huang et al. (2018). Using normalizing flows, one should be able to achieve the desired level of complexity of the latent distribution in a VAE, which should, similarly to our work, help to approximate the true posterior of the data better. The advantage of normalizing flows is the flexibility of the modeled distributions, at the expense of being more computationally expensive.

Finally, another interesting extension would be to extend the defined geometrical models to allow for training generative adversarial networks (GANs) (Goodfellow et al., 2014) in products of constant curvature spaces and benefit from the better sharpness and quality of samples that GANs provide. Finally, one could synthesize the above to achieve adversarially trained autoencoders in Riemannian manifolds similarly to Pan et al. (2018); Kim et al. (2017); Makhzani et al. (2015) and aim to achieve good sample quality and a well-formed latent space at the same time.

# E    EXTENDED RESULTS

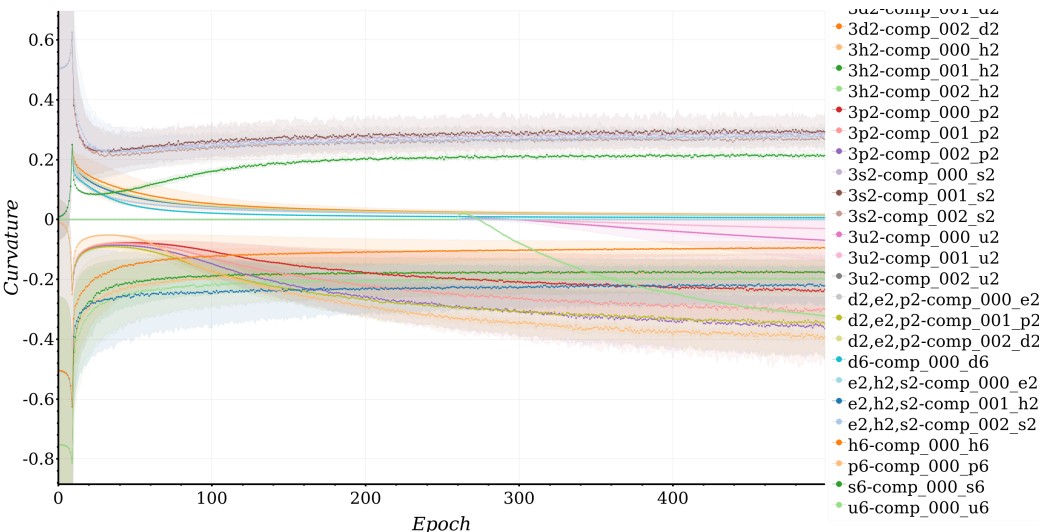

Figure 1: Learned curvature across epochs (with standard deviation) with latent space dimension of 6, diagonal covariance parametrization, on the MNIST dataset.

Table 6: Summary of results (mean and standard-deviation) with latent space dimension of 6, spherical covariance parametrization, on the BDP dataset.

| Model | LL | ELBO | BCE | KL |
|---|---|---|---|---|
| $(\mathbb{S}_1^2)^3$ | $-55.89_{\pm 0.36}$ | $-56.72_{\pm 0.40}$ | $51.01_{\pm 0.31}$ | $5.72_{\pm 0.09}$ |
| $\mathbb{S}_1^6$ | $-55.81_{\pm 0.35}$ | $-56.57_{\pm 0.44}$ | $51.16_{\pm 0.78}$ | $5.41_{\pm 0.42}$ |
| $(\text{vMF } \mathbb{S}_1^2)^3$ | $-57.87_{\pm 1.52}$ | $-58.64_{\pm 1.63}$ | $53.96_{\pm 2.16}$ | $4.68_{\pm 0.53}$ |
| $\text{vMF } \mathbb{S}_1^6$ | $-58.78_{\pm 0.83}$ | $-60.74_{\pm 2.29}$ | $56.03_{\pm 2.64}$ | $4.71_{\pm 0.48}$ |
| $(\mathbb{D}_1^2)^3$ | $-56.01_{\pm 0.24}$ | $-56.67_{\pm 0.31}$ | $51.02_{\pm 0.40}$ | $5.65_{\pm 0.10}$ |
| $\mathbb{D}_1^6$ | $-55.78_{\pm 0.07}$ | $-56.38_{\pm 0.06}$ | $50.85_{\pm 0.20}$ | $5.53_{\pm 0.24}$ |
| $(\mathbb{E}^2)^3$ | $-56.34_{\pm 0.45}$ | $-56.94_{\pm 0.50}$ | $51.32_{\pm 0.55}$ | $5.62_{\pm 0.19}$ |
| $\mathbb{E}^6$ | $-56.28_{\pm 0.56}$ | $-56.99_{\pm 0.59}$ | $51.58_{\pm 0.69}$ | $5.41_{\pm 0.29}$ |
| $(\mathbb{H}_{-1}^2)^3$ | $-56.08_{\pm 0.52}$ | $-56.80_{\pm 0.54}$ | $50.94_{\pm 0.38}$ | $5.86_{\pm 0.25}$ |
| $\mathbb{H}_{-1}^6$ | $-56.18_{\pm 0.32}$ | $-57.10_{\pm 0.21}$ | $51.48_{\pm 0.47}$ | $5.62_{\pm 0.31}$ |
| $(\mathbb{P}_{-1}^2)^3$ | $-55.98_{\pm 0.62}$ | $-56.49_{\pm 0.62}$ | $50.96_{\pm 0.61}$ | $5.52_{\pm 0.31}$ |
| $\mathbb{P}_{-1}^6$ | $-56.74_{\pm 0.55}$ | $-57.61_{\pm 0.74}$ | $52.01_{\pm 0.71}$ | $5.60_{\pm 0.24}$ |
| $(\mathcal{RN} \ \mathbb{P}_{-1}^2)^3$ | $-54.99_{\pm 0.12}$ | $-55.90_{\pm 0.13}$ | $52.42_{\pm 0.71}$ | $3.48_{\pm 0.60}$ |
| $(\mathbb{S}^2)^3$ | $-56.05_{\pm 0.21}$ | $-56.69_{\pm 0.36}$ | $51.07_{\pm 0.21}$ | $5.61_{\pm 0.22}$ |
| $(\text{vMF } \mathbb{S}^2)^3$ | $-57.56_{\pm 0.88}$ | $-57.80_{\pm 0.89}$ | $52.68_{\pm 1.62}$ | $5.12_{\pm 0.84}$ |
| $\mathbb{S}^6$ | $-56.06_{\pm 0.51}$ | $-56.65_{\pm 0.64}$ | $50.93_{\pm 0.38}$ | $5.72_{\pm 0.40}$ |
| $\text{vMF } \mathbb{S}^6$ | $-58.21_{\pm 0.92}$ | $-59.87_{\pm 1.51}$ | $54.99_{\pm 1.79}$ | $4.88_{\pm 0.39}$ |
| $(\mathbb{D}^2)^3$ | $-56.06_{\pm 0.36}$ | $-56.69_{\pm 0.54}$ | $50.95_{\pm 0.40}$ | $5.74_{\pm 0.17}$ |
| $\mathbb{D}^6$ | $-56.10_{\pm 0.25}$ | $-56.69_{\pm 0.17}$ | $50.90_{\pm 0.19}$ | $5.79_{\pm 0.03}$ |
| $(\mathbb{H}^2)^3$ | $-55.80_{\pm 0.32}$ | $-56.72_{\pm 0.16}$ | $51.14_{\pm 0.39}$ | $5.58_{\pm 0.28}$ |
| $\mathbb{H}^6$ | $-56.03_{\pm 0.21}$ | $-56.82_{\pm 0.20}$ | $50.99_{\pm 0.16}$ | $5.83_{\pm 0.27}$ |
| $(\mathbb{P}^2)^3$ | $-56.29_{\pm 0.05}$ | $-57.11_{\pm 0.22}$ | $51.41_{\pm 0.19}$ | $5.69_{\pm 0.30}$ |
| $\mathbb{P}^6$ | $-56.40_{\pm 0.31}$ | $-57.13_{\pm 0.25}$ | $51.17_{\pm 0.33}$ | $5.96_{\pm 0.27}$ |
| $(\mathcal{RN} \ \mathbb{P}^2)^3$ | $-56.25_{\pm 0.56}$ | $-57.26_{\pm 0.45}$ | $53.16_{\pm 1.07}$ | $4.11_{\pm 0.64}$ |
| $\mathbb{D}^2 \times \mathbb{E}^2 \times \mathbb{P}^2$ | $-55.87_{\pm 0.22}$ | $-56.35_{\pm 0.22}$ | $50.67_{\pm 0.57}$ | $5.69_{\pm 0.43}$ |
| $\mathbb{D}_1^2 \times \mathbb{E}^2 \times \mathbb{P}_{-1}^2$ | $-56.06_{\pm 0.41}$ | $-56.86_{\pm 0.65}$ | $51.23_{\pm 0.67}$ | $5.64_{\pm 0.11}$ |
| $\mathbb{D}^2 \times \mathbb{E}^2 \times (\mathcal{RN} \ \mathbb{P}^2)$ | $-56.35_{\pm 0.82}$ | $-57.06_{\pm 0.78}$ | $51.89_{\pm 0.71}$ | $5.17_{\pm 0.12}$ |
| $\mathbb{D}_1^2 \times \mathbb{E}^2 \times (\mathcal{RN} \ \mathbb{P}_{-1}^2)$ | $-56.17_{\pm 0.43}$ | $-56.75_{\pm 0.56}$ | $51.80_{\pm 0.80}$ | $4.95_{\pm 0.32}$ |
| $\mathbb{E}^2 \times \mathbb{H}^2 \times \mathbb{S}^2$ | $-55.92_{\pm 0.42}$ | $-56.54_{\pm 0.45}$ | $51.13_{\pm 0.74}$ | $5.41_{\pm 0.40}$ |
| $\mathbb{E}^2 \times \mathbb{H}_{-1}^2 \times \mathbb{S}_1^2$ | $-56.04_{\pm 0.57}$ | $-56.71_{\pm 0.77}$ | $51.09_{\pm 0.86}$ | $5.62_{\pm 0.12}$ |
| $\mathbb{E}^2 \times \mathbb{H}^2 \times (\text{vMF } \mathbb{S}^2)$ | $-55.82_{\pm 0.43}$ | $-56.32_{\pm 0.47}$ | $51.10_{\pm 0.67}$ | $5.21_{\pm 0.20}$ |
| $\mathbb{E}^2 \times \mathbb{H}_{-1}^2 \times (\text{vMF } \mathbb{S}_1^2)$ | $-55.77_{\pm 0.51}$ | $-56.34_{\pm 0.65}$ | $51.33_{\pm 0.57}$ | $5.01_{\pm 0.17}$ |
| $(\mathbb{U}^2)^3$ | $-55.56_{\pm 0.15}$ | $-56.05_{\pm 0.32}$ | $50.68_{\pm 0.23}$ | $5.37_{\pm 0.10}$ |
| $\mathbb{U}^6$ | $-55.84_{\pm 0.38}$ | $-56.46_{\pm 0.41}$ | $50.66_{\pm 0.38}$ | $5.81_{\pm 0.18}$ |

Table 7: Summary of results (mean and standard-deviation) with latent space dimension of 6, spherical covariance parametrization, on the MNIST dataset.

| Model | LL | ELBO | BCE | KL |
|---|---|---|---|---|
| $(\mathbb{S}_1^2)^3$ | $-96.77_{\pm 0.26}$ | $-101.66_{\pm 0.32}$ | $87.04_{\pm 0.49}$ | $14.62_{\pm 0.18}$ |
| $(\text{vMF } \mathbb{S}_1^2)^3$ | $-97.72_{\pm 0.22}$ | $-102.98_{\pm 0.15}$ | $87.77_{\pm 0.18}$ | $15.21_{\pm 0.07}$ |
| $\mathbb{S}_1^6$ | $-96.71_{\pm 0.17}$ | $-101.55_{\pm 0.30}$ | $86.90_{\pm 0.30}$ | $14.65_{\pm 0.10}$ |
| $\text{vMF } \mathbb{S}_1^6$ | $-97.03_{\pm 0.14}$ | $-102.12_{\pm 0.26}$ | $87.42_{\pm 0.28}$ | $14.69_{\pm 0.03}$ |
| $(\mathbb{D}_1^2)^3$ | $-97.84_{\pm 0.10}$ | $-102.75_{\pm 0.22}$ | $88.43_{\pm 0.12}$ | $14.33_{\pm 0.13}$ |
| $\mathbb{D}_1^6$ | $-98.21_{\pm 0.23}$ | $-103.02_{\pm 0.14}$ | $88.44_{\pm 0.05}$ | $14.58_{\pm 0.11}$ |
| $(\mathbb{E}^2)^3$ | $-97.04_{\pm 0.14}$ | $-101.44_{\pm 0.18}$ | $86.77_{\pm 0.22}$ | $14.67_{\pm 0.22}$ |
| $\mathbb{E}^6$ | $-97.16_{\pm 0.15}$ | $-101.67_{\pm 0.14}$ | $87.17_{\pm 0.26}$ | $14.50_{\pm 0.20}$ |
| $(\mathbb{H}_{-1}^2)^3$ | $-97.31_{\pm 0.09}$ | $-102.20_{\pm 0.29}$ | $87.81_{\pm 0.23}$ | $14.39_{\pm 0.13}$ |
| $\mathbb{H}_{-1}^6$ | $-97.10_{\pm 0.44}$ | $-101.89_{\pm 0.33}$ | $87.32_{\pm 0.22}$ | $14.56_{\pm 0.20}$ |
| $(\mathbb{P}_{-1}^2)^3$ | $-97.56_{\pm 0.04}$ | $-102.33_{\pm 0.22}$ | $87.93_{\pm 0.32}$ | $14.40_{\pm 0.10}$ |
| $(\mathcal{RN}\ \mathbb{P}_{-1}^2)^3$ | $-92.54_{\pm 0.19}$ | $-97.19_{\pm 0.21}$ | $88.42_{\pm 0.20}$ | $8.76_{\pm 0.04}$ |
| $\mathbb{P}_{-1}^6$ | $-97.80_{\pm 0.05}$ | $-102.60_{\pm 0.04}$ | $88.14_{\pm 0.08}$ | $14.46_{\pm 0.07}$ |
| $(\mathbb{S}^2)^3$ | $-96.46_{\pm 0.12}$ | $-101.30_{\pm 0.17}$ | $86.79_{\pm 0.25}$ | $14.51_{\pm 0.09}$ |
| $(\text{vMF } \mathbb{S}^2)^3$ | $-97.62_{\pm 0.30}$ | $-102.72_{\pm 0.37}$ | $87.48_{\pm 0.37}$ | $15.24_{\pm 0.03}$ |
| $\mathbb{S}^6$ | $-96.72_{\pm 0.15}$ | $-101.39_{\pm 0.16}$ | $86.69_{\pm 0.15}$ | $14.70_{\pm 0.13}$ |
| $\text{vMF } \mathbb{S}^6$ | $-96.72_{\pm 0.18}$ | $-101.55_{\pm 0.21}$ | $86.82_{\pm 0.23}$ | $14.73_{\pm 0.02}$ |
| $(\mathbb{D}^2)^3$ | $-97.68_{\pm 0.24}$ | $-102.51_{\pm 0.44}$ | $88.11_{\pm 0.34}$ | $14.41_{\pm 0.11}$ |
| $\mathbb{D}^6$ | $-97.72_{\pm 0.15}$ | $-102.31_{\pm 0.16}$ | $87.70_{\pm 0.22}$ | $14.61_{\pm 0.06}$ |
| $(\mathbb{H}^2)^3$ | $-97.37_{\pm 0.13}$ | $-102.07_{\pm 0.24}$ | $87.56_{\pm 0.30}$ | $14.51_{\pm 0.11}$ |
| $\mathbb{H}^6$ | $-97.47_{\pm 0.16}$ | $-102.18_{\pm 0.20}$ | $87.64_{\pm 0.23}$ | $14.53_{\pm 0.07}$ |
| $(\mathbb{P}^2)^3$ | $-97.62_{\pm 0.05}$ | $-102.34_{\pm 0.16}$ | $87.92_{\pm 0.16}$ | $14.43_{\pm 0.06}$ |
| $(\mathcal{RN}\ \mathbb{P}^2)^3$ | $-94.16_{\pm 0.68}$ | $-98.65_{\pm 0.66}$ | $89.27_{\pm 0.79}$ | $9.38_{\pm 0.15}$ |
| $\mathbb{P}^6$ | $-97.71_{\pm 0.24}$ | $-102.55_{\pm 0.21}$ | $88.24_{\pm 0.23}$ | $14.32_{\pm 0.04}$ |
| $\mathbb{D}^2 \times \mathbb{E}^2 \times \mathbb{P}^2$ | $-97.48_{\pm 0.18}$ | $-102.22_{\pm 0.29}$ | $87.85_{\pm 0.17}$ | $14.37_{\pm 0.13}$ |
| $\mathbb{D}_1^2 \times \mathbb{E}^2 \times \mathbb{P}_{-1}^2$ | $-97.58_{\pm 0.13}$ | $-102.23_{\pm 0.15}$ | $87.75_{\pm 0.15}$ | $14.49_{\pm 0.15}$ |
| $\mathbb{D}^2 \times \mathbb{E}^2 \times (\mathcal{RN}\ \mathbb{P}^2)$ | $-96.43_{\pm 0.47}$ | $-101.31_{\pm 0.51}$ | $88.82_{\pm 0.50}$ | $12.50_{\pm 0.03}$ |
| $\mathbb{D}_1^2 \times \mathbb{E}^2 \times (\mathcal{RN}\ \mathbb{P}_{-1}^2)$ | $-96.18_{\pm 0.21}$ | $-100.91_{\pm 0.31}$ | $88.58_{\pm 0.47}$ | $12.33_{\pm 0.19}$ |
| $\mathbb{E}^2 \times \mathbb{H}^2 \times \mathbb{S}^2$ | $-96.80_{\pm 0.20}$ | $-101.60_{\pm 0.33}$ | $87.13_{\pm 0.19}$ | $14.47_{\pm 0.17}$ |
| $\mathbb{E}^2 \times \mathbb{H}_{-1}^2 \times \mathbb{S}_1^2$ | $-96.76_{\pm 0.09}$ | $-101.48_{\pm 0.13}$ | $86.99_{\pm 0.17}$ | $14.49_{\pm 0.05}$ |
| $\mathbb{E}^2 \times \mathbb{H}^2 \times (\text{vMF } \mathbb{S}^2)$ | $-96.56_{\pm 0.27}$ | $-101.49_{\pm 0.28}$ | $86.58_{\pm 0.36}$ | $14.91_{\pm 0.14}$ |
| $\mathbb{E}^2 \times \mathbb{H}_{-1}^2 \times (\text{vMF } \mathbb{S}_1^2)$ | $-96.76_{\pm 0.39}$ | $-101.82_{\pm 0.13}$ | $87.08_{\pm 0.06}$ | $14.74_{\pm 0.13}$ |
| $(\mathbb{U}^2)^3$ | $-97.12_{\pm 0.04}$ | $-101.68_{\pm 0.06}$ | $87.13_{\pm 0.14}$ | $14.55_{\pm 0.16}$ |
| $\mathbb{U}^6$ | $-97.26_{\pm 0.16}$ | $-102.05_{\pm 0.18}$ | $87.54_{\pm 0.21}$ | $14.51_{\pm 0.11}$ |

Table 8: Summary of results (mean and standard-deviation) with latent space dimension of 6, diagonal covariance parametrization, on the MNIST dataset.

| Model | LL | ELBO | BCE | KL |
|---|---|---|---|---|
| $(\mathbb{S}_1^2)^3$ | $-96.57_{\pm 0.04}$ | $-101.34_{\pm 0.12}$ | $86.88_{\pm 0.17}$ | $14.45_{\pm 0.10}$ |
| $\mathbb{S}_1^6$ | $-96.51_{\pm 0.09}$ | $-101.29_{\pm 0.18}$ | $86.71_{\pm 0.20}$ | $14.58_{\pm 0.13}$ |
| $(\mathbb{D}_1^2)^3$ | $-97.81_{\pm 0.14}$ | $-102.58_{\pm 0.23}$ | $88.31_{\pm 0.25}$ | $14.27_{\pm 0.02}$ |
| $\mathbb{D}_1^6$ | $-97.89_{\pm 0.10}$ | $-102.65_{\pm 0.10}$ | $88.39_{\pm 0.16}$ | $14.26_{\pm 0.08}$ |
| $(\mathbb{E}^2)^3$ | $-96.94_{\pm 0.34}$ | $-101.34_{\pm 0.41}$ | $86.89_{\pm 0.36}$ | $14.44_{\pm 0.11}$ |
| $\mathbb{E}^6$ | $-96.88_{\pm 0.16}$ | $-101.36_{\pm 0.08}$ | $86.90_{\pm 0.14}$ | $14.46_{\pm 0.07}$ |
| $(\mathbb{H}_{-1}^2)^3$ | $-97.19_{\pm 0.32}$ | $-102.06_{\pm 0.28}$ | $87.63_{\pm 0.37}$ | $14.42_{\pm 0.10}$ |
| $\mathbb{H}_{-1}^6$ | $-97.38_{\pm 0.73}$ | $-102.22_{\pm 0.95}$ | $87.75_{\pm 0.59}$ | $14.47_{\pm 0.37}$ |
| $(\mathbb{P}_{-1}^2)^3$ | $-97.57_{\pm 0.12}$ | $-102.22_{\pm 0.18}$ | $87.83_{\pm 0.30}$ | $14.39_{\pm 0.13}$ |
| $\mathbb{P}_{-1}^6$ | $-97.33_{\pm 0.15}$ | $-102.02_{\pm 0.35}$ | $87.71_{\pm 0.36}$ | $14.31_{\pm 0.04}$ |
| $(\mathbb{S}^2)^3$ | $-96.78_{\pm 0.35}$ | $-101.43_{\pm 0.24}$ | $86.93_{\pm 0.28}$ | $14.50_{\pm 0.05}$ |
| $\mathbb{S}^6$ | $-96.44_{\pm 0.20}$ | $-101.18_{\pm 0.36}$ | $86.74_{\pm 0.38}$ | $14.44_{\pm 0.05}$ |
| $(\mathbb{D}^2)^3$ | $-97.61_{\pm 0.19}$ | $-102.37_{\pm 0.26}$ | $87.96_{\pm 0.21}$ | $14.41_{\pm 0.06}$ |
| $\mathbb{D}^6$ | $-97.53_{\pm 0.22}$ | $-102.31_{\pm 0.38}$ | $87.97_{\pm 0.37}$ | $14.34_{\pm 0.08}$ |
| $(\mathbb{H}^2)^3$ | $-96.86_{\pm 0.31}$ | $-101.61_{\pm 0.30}$ | $87.13_{\pm 0.30}$ | $14.48_{\pm 0.08}$ |
| $\mathbb{H}^6$ | $-96.90_{\pm 0.26}$ | $-101.48_{\pm 0.35}$ | $87.18_{\pm 0.48}$ | $14.30_{\pm 0.15}$ |
| $(\mathbb{P}^2)^3$ | $-97.52_{\pm 0.02}$ | $-102.30_{\pm 0.07}$ | $88.11_{\pm 0.07}$ | $14.19_{\pm 0.12}$ |
| $\mathbb{P}^6$ | $-97.26_{\pm 0.16}$ | $-102.00_{\pm 0.17}$ | $87.58_{\pm 0.16}$ | $14.42_{\pm 0.08}$ |
| $\mathbb{D}^2 \times \mathbb{E}^2 \times \mathbb{P}^2$ | $-97.37_{\pm 0.14}$ | $-102.12_{\pm 0.19}$ | $87.78_{\pm 0.23}$ | $14.34_{\pm 0.12}$ |
| $\mathbb{D}_1^2 \times \mathbb{E}^2 \times \mathbb{P}_{-1}^2$ | $-97.29_{\pm 0.16}$ | $-101.86_{\pm 0.16}$ | $87.54_{\pm 0.17}$ | $14.32_{\pm 0.04}$ |
| $\mathbb{E}^2 \times \mathbb{H}^2 \times \mathbb{S}^2$ | $-96.71_{\pm 0.19}$ | $-101.34_{\pm 0.16}$ | $86.91_{\pm 0.17}$ | $14.43_{\pm 0.06}$ |
| $\mathbb{E}^2 \times \mathbb{H}_{-1}^2 \times \mathbb{S}_1^2$ | $-96.66_{\pm 0.27}$ | $-101.46_{\pm 0.44}$ | $87.02_{\pm 0.38}$ | $14.44_{\pm 0.08}$ |
| $(\mathbb{U}^2)^3$ | $-97.06_{\pm 0.13}$ | $-101.66_{\pm 0.19}$ | $87.22_{\pm 0.12}$ | $14.44_{\pm 0.07}$ |
| $\mathbb{U}^6$ | $-96.90_{\pm 0.10}$ | $-101.68_{\pm 0.07}$ | $87.27_{\pm 0.11}$ | $14.42_{\pm 0.12}$ |

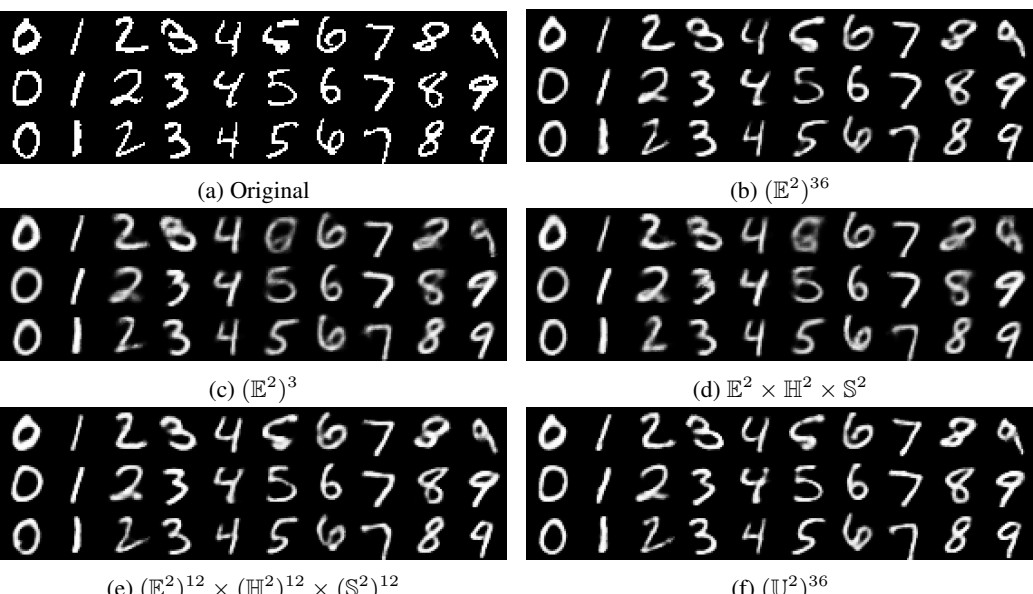

(a) Original     (b) $(\mathbb{E}^2)^{36}$

(c) $(\mathbb{E}^2)^3$     (d) $\mathbb{E}^2 \times \mathbb{H}^2 \times \mathbb{S}^2$

(e) $(\mathbb{E}^2)^{12} \times (\mathbb{H}^2)^{12} \times (\mathbb{S}^2)^{12}$     (f) $(\mathbb{U}^2)^{36}$

Figure 2: Qualitative comparison of reconstruction quality of randomly selected runs of a selection of well-performing models on MNIST test set digits.

Table 9: Summary of results (mean and standard-deviation) with latent space dimension of 12, diagonal covariance parametrization, on the MNIST dataset.

| Model | LL | ELBO | BCE | KL |
|---|---|---|---|---|
| $(\mathbb{S}_1^2)^6$ | $-79.92_{\pm 0.21}$ | $-84.88_{\pm 0.14}$ | $62.83_{\pm 0.21}$ | $22.06_{\pm 0.07}$ |
| $\mathbb{S}_1^{12}$ | $-80.72_{\pm 0.34}$ | $-85.73_{\pm 0.36}$ | $63.86_{\pm 0.32}$ | $21.87_{\pm 0.04}$ |
| $(\mathbb{D}_1^2)^6$ | $-80.53_{\pm 0.10}$ | $-85.59_{\pm 0.08}$ | $63.62_{\pm 0.12}$ | $21.97_{\pm 0.16}$ |
| $\mathbb{D}_1^{12}$ | $-80.81_{\pm 0.12}$ | $-86.40_{\pm 0.17}$ | $64.42_{\pm 0.19}$ | $21.98_{\pm 0.06}$ |
| $(\mathbb{E}^2)^6$ | $-79.51_{\pm 0.10}$ | $-83.91_{\pm 0.12}$ | $61.84_{\pm 0.06}$ | $22.07_{\pm 0.13}$ |
| $\mathbb{E}^{12}$ | $-79.51_{\pm 0.09}$ | $-83.95_{\pm 0.06}$ | $61.66_{\pm 0.10}$ | $22.29_{\pm 0.04}$ |
| $(\mathbb{H}_{-1}^2)^6$ | $-80.54_{\pm 0.23}$ | $-86.05_{\pm 0.52}$ | $63.78_{\pm 0.26}$ | $22.27_{\pm 0.26}$ |
| $\mathbb{H}_{-1}^{12}$ | $-79.37_{\pm 0.14}$ | $-84.76_{\pm 0.08}$ | $62.32_{\pm 0.05}$ | $22.44_{\pm 0.10}$ |
| $(\mathbb{P}_{-1}^2)^6$ | $-80.39_{\pm 0.07}$ | $-85.46_{\pm 0.15}$ | $63.48_{\pm 0.22}$ | $21.98_{\pm 0.17}$ |
| $\mathbb{P}_{-1}^{12}$ | $-80.88_{\pm 0.20}$ | $-85.87_{\pm 0.45}$ | $63.66_{\pm 0.59}$ | $22.21_{\pm 0.17}$ |
| $(\mathbb{S}^2)^6$ | $-79.95_{\pm 0.14}$ | $-84.90_{\pm 0.25}$ | $62.83_{\pm 0.34}$ | $22.07_{\pm 0.17}$ |
| $\mathbb{S}^{12}$ | $-79.99_{\pm 0.27}$ | $-84.78_{\pm 0.26}$ | $62.89_{\pm 0.29}$ | $21.89_{\pm 0.18}$ |
| $(\mathbb{D}^2)^6$ | $-80.40_{\pm 0.09}$ | $-85.38_{\pm 0.08}$ | $63.49_{\pm 0.12}$ | $21.89_{\pm 0.18}$ |
| $\mathbb{D}^{12}$ | $-80.37_{\pm 0.16}$ | $-85.26_{\pm 0.19}$ | $63.24_{\pm 0.15}$ | $22.02_{\pm 0.13}$ |
| $(\mathbb{H}^2)^6$ | $-80.13_{\pm 0.08}$ | $-85.22_{\pm 0.24}$ | $63.32_{\pm 0.34}$ | $21.90_{\pm 0.10}$ |
| $\mathbb{H}^{12}$ | $-79.77_{\pm 0.10}$ | $-84.58_{\pm 0.15}$ | $62.49_{\pm 0.10}$ | $22.09_{\pm 0.20}$ |
| $(\mathbb{P}^2)^6$ | $-80.31_{\pm 0.08}$ | $-85.35_{\pm 0.10}$ | $63.57_{\pm 0.17}$ | $21.79_{\pm 0.07}$ |
| $\mathbb{P}^{12}$ | $-80.66_{\pm 0.09}$ | $-85.55_{\pm 0.03}$ | $63.55_{\pm 0.17}$ | $22.00_{\pm 0.14}$ |
| $(\mathbb{D}^2)^2 \times (\mathbb{E}^2)^2 \times (\mathbb{P}^2)^2$ | $-80.30_{\pm 0.31}$ | $-85.22_{\pm 0.40}$ | $63.52_{\pm 0.48}$ | $21.70_{\pm 0.11}$ |
| $(\mathbb{D}_1^2)^2 \times (\mathbb{E}^2)^2 \times (\mathbb{P}_{-1}^2)^2$ | $-80.14_{\pm 0.11}$ | $-85.00_{\pm 0.08}$ | $62.99_{\pm 0.16}$ | $22.01_{\pm 0.24}$ |
| $\mathbb{D}^4 \times \mathbb{E}^4 \times \mathbb{P}^4$ | $-80.17_{\pm 0.11}$ | $-84.95_{\pm 0.27}$ | $62.87_{\pm 0.39}$ | $22.08_{\pm 0.18}$ |
| $\mathbb{D}_1^4 \times \mathbb{E}^4 \times \mathbb{P}_{-1}^4$ | $-80.14_{\pm 0.20}$ | $-84.99_{\pm 0.26}$ | $63.06_{\pm 0.26}$ | $21.92_{\pm 0.08}$ |
| $(\mathbb{E}^2)^2 \times (\mathbb{H}^2)^2 \times (\mathbb{S}^2)^2$ | $-79.59_{\pm 0.25}$ | $-84.43_{\pm 0.20}$ | $62.68_{\pm 0.20}$ | $21.75_{\pm 0.20}$ |
| $(\mathbb{E}^2)^2 \times (\mathbb{H}_{-1}^2)^2 \times (\mathbb{S}_1^2)^2$ | $-79.87_{\pm 0.45}$ | $-84.82_{\pm 0.61}$ | $62.66_{\pm 0.42}$ | $22.17_{\pm 0.20}$ |
| $\mathbb{E}^4 \times \mathbb{H}^4 \times \mathbb{S}^4$ | $-79.69_{\pm 0.14}$ | $-84.45_{\pm 0.12}$ | $62.64_{\pm 0.28}$ | $21.81_{\pm 0.21}$ |
| $\mathbb{E}^4 \times \mathbb{H}_{-1}^4 \times \mathbb{S}_1^4$ | $-79.77_{\pm 0.09}$ | $-84.75_{\pm 0.03}$ | $62.68_{\pm 0.25}$ | $22.07_{\pm 0.24}$ |
| $(\mathbb{U}^2)^6$ | $-79.61_{\pm 0.06}$ | $-84.13_{\pm 0.04}$ | $61.92_{\pm 0.22}$ | $22.21_{\pm 0.23}$ |
| $\mathbb{U}^{12}$ | $-80.01_{\pm 0.30}$ | $-84.86_{\pm 0.51}$ | $62.90_{\pm 0.63}$ | $21.96_{\pm 0.16}$ |

Table 10: Summary of results (mean and standard-deviation) with latent space dimension of 72, diagonal covariance parametrization, on the MNIST dataset.

| Model | LL | ELBO | BCE | KL |
|---|---|---|---|---|
| $(\mathbb{S}_1^2)^{36}$ | $-78.43_{\pm 0.44}$ | $-84.99_{\pm 0.49}$ | $56.88_{\pm 0.28}$ | $28.11_{\pm 0.56}$ |
| $(\mathbb{D}_1^2)^{36}$ | $-76.03_{\pm 0.17}$ | $-83.04_{\pm 0.25}$ | $54.35_{\pm 0.15}$ | $28.69_{\pm 0.17}$ |
| $(\mathbb{E}^2)^{36}$ | $-74.53_{\pm 0.06}$ | $-80.05_{\pm 0.10}$ | $50.91_{\pm 0.17}$ | $29.15_{\pm 0.07}$ |
| $\mathbb{E}^{72}$ | $-74.42_{\pm 0.06}$ | $-80.09_{\pm 0.12}$ | $51.45_{\pm 0.30}$ | $28.63_{\pm 0.20}$ |
| $(\mathbb{H}_{-1}^2)^{36}$ | $-77.92_{\pm 0.32}$ | $-84.76_{\pm 0.55}$ | $56.85_{\pm 0.60}$ | $27.91_{\pm 0.42}$ |
| $\mathbb{H}_{-1}^{72}$ | $-77.30_{\pm 0.12}$ | $-86.98_{\pm 0.09}$ | $58.04_{\pm 0.29}$ | $28.94_{\pm 0.25}$ |
| $(\mathbb{P}_{-1}^2)^{36}$ | $-76.11_{\pm 0.08}$ | $-82.63_{\pm 0.19}$ | $53.89_{\pm 0.36}$ | $28.74_{\pm 0.30}$ |
| $\mathbb{P}_{-1}^{72}$ | $-77.50_{\pm 0.05}$ | $-84.53_{\pm 0.13}$ | $55.80_{\pm 0.20}$ | $28.73_{\pm 0.18}$ |
| $\mathbb{S}^{72}$ | $-75.24_{\pm 0.01}$ | $-81.39_{\pm 0.14}$ | $53.03_{\pm 0.27}$ | $28.36_{\pm 0.16}$ |
| $(\mathbb{D}^2)^{36}$ | $-75.66_{\pm 0.06}$ | $-81.94_{\pm 0.09}$ | $53.32_{\pm 0.16}$ | $28.61_{\pm 0.11}$ |
| $\mathbb{D}^{72}$ | $-77.11_{\pm 2.21}$ | $-83.94_{\pm 2.81}$ | $54.94_{\pm 2.55}$ | $29.00_{\pm 1.31}$ |
| $(\mathbb{H}^2)^{36}$ | $-77.87_{\pm 0.02}$ | $-83.95_{\pm 0.02}$ | $55.71_{\pm 0.35}$ | $28.24_{\pm 0.36}$ |
| $\mathbb{H}^{72}$ | $-75.03_{\pm 0.11}$ | $-81.23_{\pm 0.14}$ | $52.63_{\pm 0.10}$ | $28.61_{\pm 0.11}$ |
| $(\mathbb{P}^2)^{36}$ | $-75.77_{\pm 0.12}$ | $-82.07_{\pm 0.02}$ | $53.65_{\pm 0.38}$ | $28.43_{\pm 0.39}$ |
| $\mathbb{P}^{72}$ | $-75.71_{\pm 0.08}$ | $-81.95_{\pm 0.09}$ | $53.29_{\pm 0.14}$ | $28.67_{\pm 0.05}$ |
| $(\mathbb{D}^2)^{12} \times (\mathbb{E}^2)^{12} \times (\mathbb{P}^2)^{12}$ | $-77.40_{\pm 0.55}$ | $-83.35_{\pm 0.41}$ | $53.90_{\pm 0.40}$ | $29.45_{\pm 0.12}$ |
| $(\mathbb{D}_1^2)^{12} \times (\mathbb{E}^2)^{12} \times (\mathbb{P}_{-1}^2)^{12}$ | $-75.36_{\pm 0.23}$ | $-81.53_{\pm 0.42}$ | $53.02_{\pm 0.39}$ | $28.51_{\pm 0.45}$ |
| $\mathbb{D}^{24} \times \mathbb{E}^{24} \times \mathbb{P}^{24}$ | $-75.11_{\pm 0.05}$ | $-80.99_{\pm 0.07}$ | $52.48_{\pm 0.19}$ | $28.52_{\pm 0.16}$ |
| $(\mathbb{E}^2)^{12} \times (\mathbb{H}_{-1}^2)^{12} \times (\mathbb{S}_1^2)^{12}$ | $-77.53_{\pm 0.34}$ | $-83.95_{\pm 0.40}$ | $55.54_{\pm 0.43}$ | $28.42_{\pm 0.08}$ |
| $\mathbb{E}^{24} \times \mathbb{H}^{24} \times \mathbb{S}^{24}$ | $-75.04_{\pm 0.16}$ | $-81.17_{\pm 0.18}$ | $52.61_{\pm 0.32}$ | $28.55_{\pm 0.38}$ |
| $(\mathbb{U}^2)^{36}$ | $-74.64_{\pm 0.08}$ | $-80.52_{\pm 0.10}$ | $52.04_{\pm 0.10}$ | $28.48_{\pm 0.07}$ |
| $\mathbb{U}^{72}$ | $-75.46_{\pm 0.09}$ | $-81.76_{\pm 0.09}$ | $53.27_{\pm 0.18}$ | $28.49_{\pm 0.18}$ |

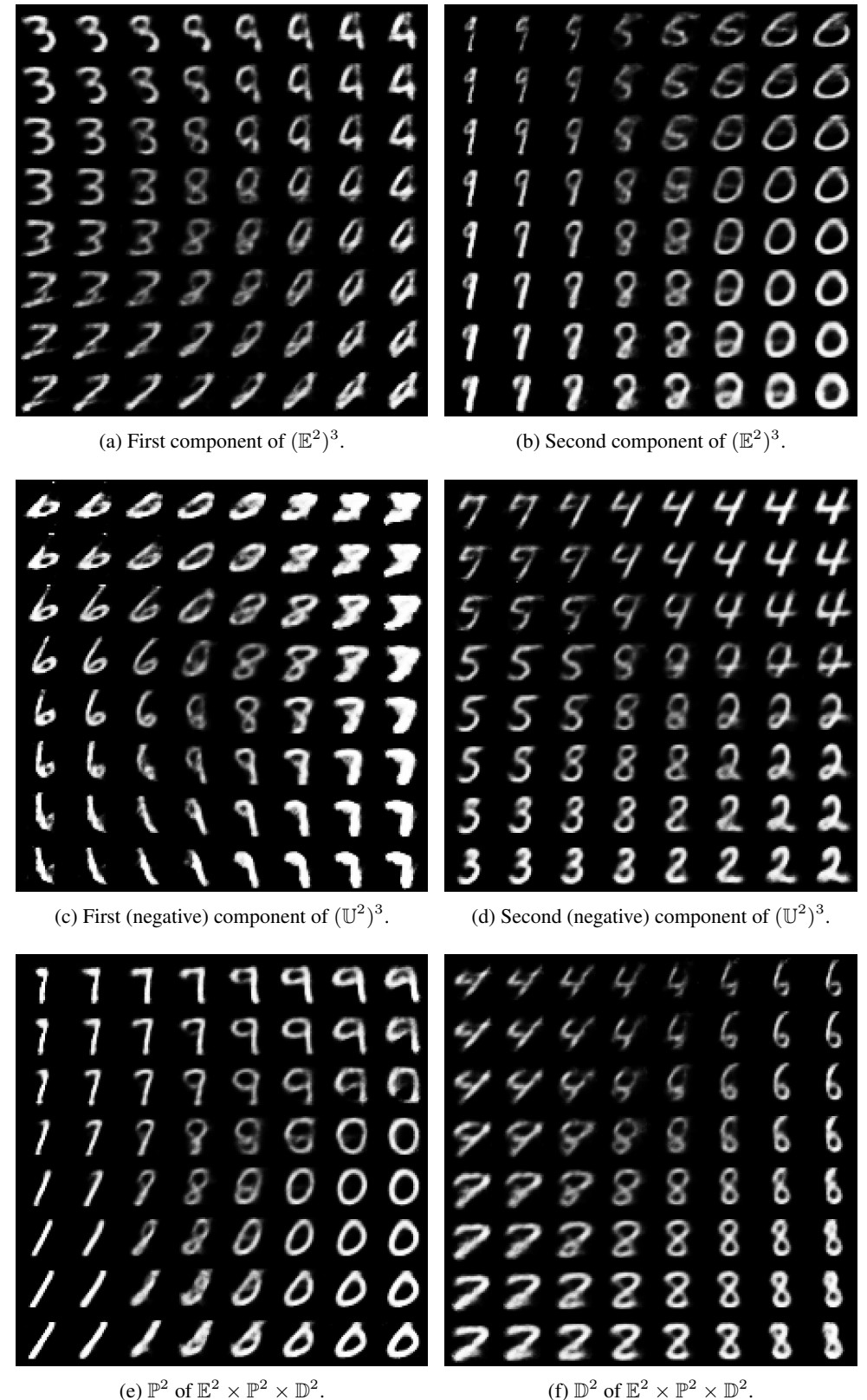

(a) First component of $(\mathbb{E}^2)^3$.

(b) Second component of $(\mathbb{E}^2)^3$.

(c) First (negative) component of $(\mathbb{U}^2)^3$.

(d) Second (negative) component of $(\mathbb{U}^2)^3$.

(e) $\mathbb{P}^2$ of $\mathbb{E}^2 \times \mathbb{P}^2 \times \mathbb{D}^2$.

(f) $\mathbb{D}^2$ of $\mathbb{E}^2 \times \mathbb{P}^2 \times \mathbb{D}^2$.

Figure 3: Samples from various models of a grid search around **0** of a single component's latent space on MNIST test digits.

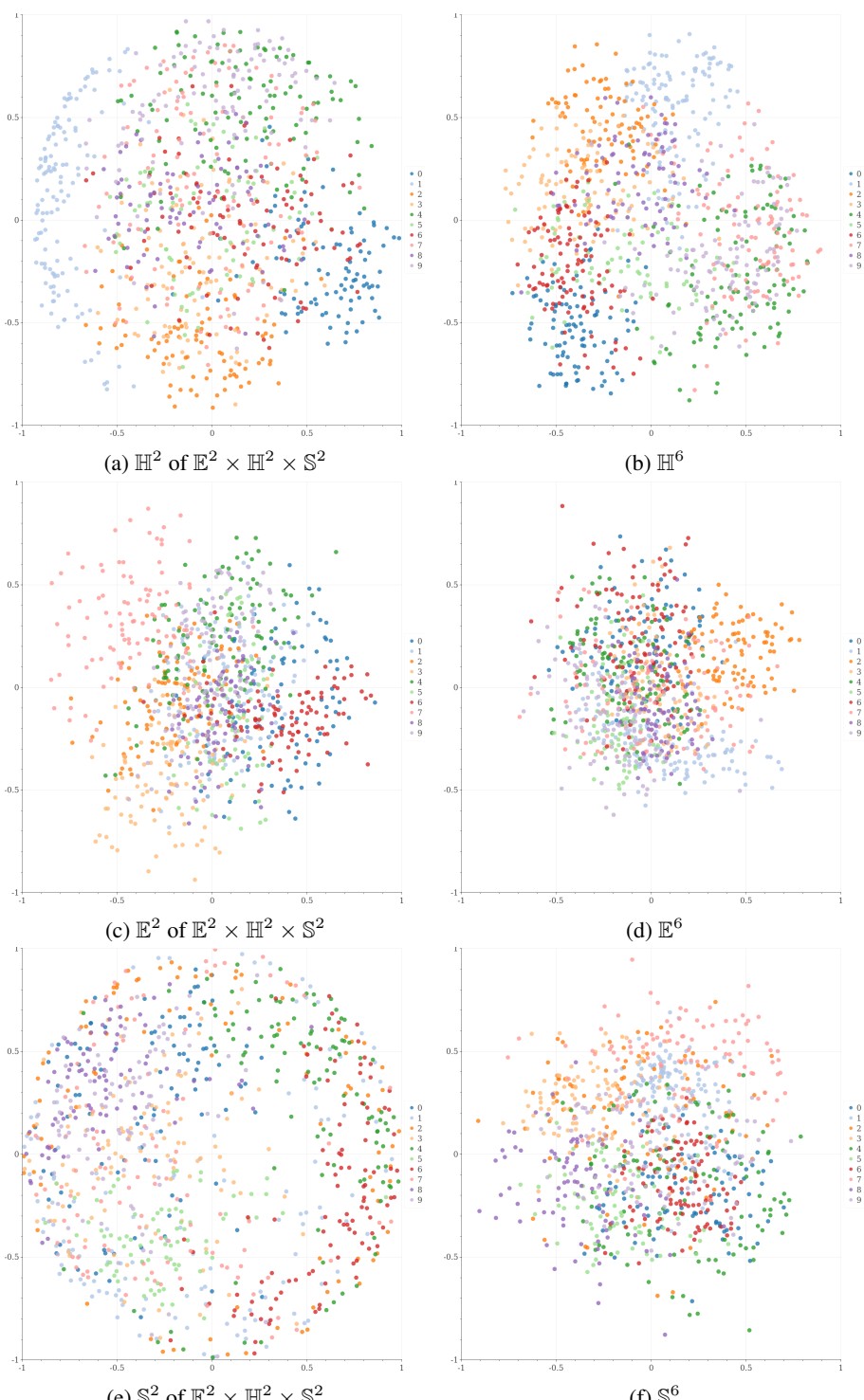

Figure 4: Illustrative latent space visualization of a randomly selected run of the models $\mathbb{E}^2 \times \mathbb{H}^2 \times \mathbb{S}^2$, $\mathbb{E}^6$, $\mathbb{H}^6$, and $\mathbb{S}^6$ on MNIST with spherical covariance. $\mathbb{E}^2$ is visualized directly, $\mathbb{S}^2$ is visualized using a Lambert azimuthal equal-area projection (Snyder, 1987, Chapter 24), $\mathbb{H}_2$ is transformed to the Poincaré ball model using a stereographic conformal projection. All other latent space sizes were first projected using the respective transformation (to Poincaré ball, Lambert projection) if applicable, and then projected to $\mathbb{R}^2$ using Principal Component Analysis (Abdi & Williams, 2010, PCA) and visualized directly.

Table 11: Summary of results (mean and standard-deviation) with latent space dimension of 6, diagonal covariance parametrization, on the Omniglot dataset.

| Model | LL | ELBO | BCE | KL |
|---|---|---|---|---|
| $(\mathbb{S}_1^2)^3$ | $-136.80_{\pm 1.31}$ | $-141.68_{\pm 1.52}$ | $131.73_{\pm 5.65}$ | $9.95_{\pm 4.33}$ |
| $\mathbb{S}_1^6$ | $-136.69_{\pm 0.94}$ | $-141.46_{\pm 0.92}$ | $129.52_{\pm 0.74}$ | $11.94_{\pm 0.19}$ |
| $(\mathbb{D}_1^2)^3$ | $-136.21_{\pm 0.12}$ | $-140.44_{\pm 0.17}$ | $128.93_{\pm 0.14}$ | $11.51_{\pm 0.04}$ |
| $\mathbb{D}_1^6$ | $-137.42_{\pm 1.20}$ | $-141.95_{\pm 1.94}$ | $130.70_{\pm 2.18}$ | $11.25_{\pm 0.26}$ |
| $(\mathbb{E}^2)^3$ | $-136.08_{\pm 0.21}$ | $-140.46_{\pm 0.24}$ | $128.85_{\pm 0.34}$ | $11.62_{\pm 0.14}$ |
| $\mathbb{E}^6$ | $-136.05_{\pm 0.29}$ | $-140.50_{\pm 0.35}$ | $128.95_{\pm 0.41}$ | $11.55_{\pm 0.14}$ |
| $(\mathbb{H}_{-1}^2)^3$ | $-137.14_{\pm 0.13}$ | $-141.87_{\pm 0.16}$ | $130.18_{\pm 0.21}$ | $11.69_{\pm 0.10}$ |
| $\mathbb{H}_{-1}^6$ | $-137.09_{\pm 0.06}$ | $-142.22_{\pm 0.19}$ | $130.37_{\pm 0.21}$ | $11.85_{\pm 0.12}$ |
| $(\mathbb{P}_{-1}^2)^3$ | $-136.16_{\pm 0.20}$ | $-140.63_{\pm 0.32}$ | $129.29_{\pm 0.34}$ | $11.34_{\pm 0.03}$ |
| $\mathbb{P}_{-1}^6$ | $-135.86_{\pm 0.20}$ | $-140.36_{\pm 0.19}$ | $128.92_{\pm 0.23}$ | $11.44_{\pm 0.16}$ |
| $(\mathbb{S}^2)^3$ | $-136.14_{\pm 0.27}$ | $-140.68_{\pm 0.32}$ | $128.98_{\pm 0.27}$ | $11.70_{\pm 0.13}$ |
| $\mathbb{S}^6$ | $-136.20_{\pm 0.44}$ | $-140.76_{\pm 0.45}$ | $129.10_{\pm 0.37}$ | $11.66_{\pm 0.13}$ |
| $(\mathbb{D}^2)^3$ | $-136.13_{\pm 0.17}$ | $-140.59_{\pm 0.15}$ | $129.10_{\pm 0.20}$ | $11.49_{\pm 0.12}$ |
| $\mathbb{D}^6$ | $-136.30_{\pm 0.08}$ | $-140.74_{\pm 0.14}$ | $129.35_{\pm 0.16}$ | $11.39_{\pm 0.05}$ |
| $(\mathbb{H}^2)^3$ | $-136.17_{\pm 0.09}$ | $-140.65_{\pm 0.17}$ | $129.26_{\pm 0.07}$ | $11.39_{\pm 0.16}$ |
| $\mathbb{H}^6$ | $-136.24_{\pm 0.32}$ | $-140.92_{\pm 0.33}$ | $129.48_{\pm 0.27}$ | $11.45_{\pm 0.12}$ |
| $(\mathbb{P}^2)^3$ | $-136.09_{\pm 0.07}$ | $-140.41_{\pm 0.08}$ | $129.04_{\pm 0.05}$ | $11.37_{\pm 0.08}$ |
| $\mathbb{P}^6$ | $-136.05_{\pm 0.44}$ | $-140.42_{\pm 0.47}$ | $129.04_{\pm 0.53}$ | $11.38_{\pm 0.07}$ |
| $\mathbb{D}^2 \times \mathbb{E}^2 \times \mathbb{P}^2$ | $-135.89_{\pm 0.40}$ | $-140.28_{\pm 0.42}$ | $128.75_{\pm 0.40}$ | $11.53_{\pm 0.04}$ |
| $\mathbb{D}_1^2 \times \mathbb{E}^2 \times \mathbb{P}_{-1}^2$ | $-136.01_{\pm 0.31}$ | $-140.52_{\pm 0.35}$ | $129.02_{\pm 0.27}$ | $11.50_{\pm 0.11}$ |
| $\mathbb{E}^2 \times \mathbb{H}^2 \times \mathbb{S}^2$ | $-135.93_{\pm 0.48}$ | $-140.51_{\pm 0.53}$ | $128.85_{\pm 0.48}$ | $11.66_{\pm 0.14}$ |
| $\mathbb{E}^2 \times \mathbb{H}_{-1}^2 \times \mathbb{S}_1^2$ | $-136.34_{\pm 0.41}$ | $-141.02_{\pm 0.46}$ | $129.24_{\pm 0.47}$ | $11.78_{\pm 0.10}$ |
| $(\mathbb{U}^2)^3$ | $-136.21_{\pm 0.07}$ | $-140.65_{\pm 0.30}$ | $129.14_{\pm 0.34}$ | $11.52_{\pm 0.15}$ |
| $\mathbb{U}^6$ | $-136.04_{\pm 0.17}$ | $-140.43_{\pm 0.14}$ | $129.07_{\pm 0.27}$ | $11.36_{\pm 0.13}$ |

Table 12: Summary of results (mean and standard-deviation) with latent space dimension of 72, diagonal covariance parametrization, on the Omniglot dataset.

| Model | LL | ELBO | BCE | KL |
|---|---|---|---|---|
| $(\mathbb{S}_1^2)^{36}$ | $-112.33_{\pm0.14}$ | $-118.94_{\pm0.14}$ | $91.04_{\pm0.37}$ | $27.90_{\pm0.23}$ |
| $(\mathbb{D}_1^2)^{36}$ | $-108.66_{\pm0.24}$ | $-116.06_{\pm0.18}$ | $85.95_{\pm0.16}$ | $30.11_{\pm0.04}$ |
| $(\mathbb{E}^2)^{36}$ | $-105.96_{\pm0.33}$ | $-112.41_{\pm0.35}$ | $79.80_{\pm0.72}$ | $32.61_{\pm0.41}$ |
| $\mathbb{E}^{72}$ | $-105.89_{\pm0.16}$ | $-112.40_{\pm0.17}$ | $79.52_{\pm0.19}$ | $32.89_{\pm0.20}$ |
| $(\mathbb{H}_{-1}^2)^{36}$ | $-112.22_{\pm0.11}$ | $-119.06_{\pm0.15}$ | $91.30_{\pm0.47}$ | $27.76_{\pm0.35}$ |
| $\mathbb{H}_{-1}^{72}$ | $-111.19_{\pm0.42}$ | $-120.49_{\pm0.35}$ | $91.11_{\pm0.73}$ | $29.38_{\pm0.40}$ |
| $(\mathbb{P}_{-1}^2)^{36}$ | $-109.05_{\pm0.09}$ | $-115.99_{\pm0.10}$ | $85.81_{\pm0.42}$ | $30.18_{\pm0.34}$ |
| $\mathbb{P}_{-1}^{72}$ | $-111.24_{\pm0.28}$ | $-118.36_{\pm0.24}$ | $89.53_{\pm0.38}$ | $28.84_{\pm0.18}$ |
| $\mathbb{S}^{72}$ | $-109.39_{\pm0.32}$ | $-116.42_{\pm0.32}$ | $87.22_{\pm0.58}$ | $29.20_{\pm0.28}$ |
| $(\mathbb{D}^2)^{36}$ | $-108.89_{\pm0.36}$ | $-115.65_{\pm0.45}$ | $85.29_{\pm0.74}$ | $30.37_{\pm0.30}$ |
| $\mathbb{D}^{72}$ | $-108.81_{\pm0.08}$ | $-115.71_{\pm0.09}$ | $85.68_{\pm0.10}$ | $30.03_{\pm0.09}$ |
| $(\mathbb{H}^2)^{36}$ | $-112.21_{\pm0.28}$ | $-118.74_{\pm0.30}$ | $91.03_{\pm0.76}$ | $27.71_{\pm0.47}$ |
| $\mathbb{H}^{72}$ | $-108.62_{\pm0.40}$ | $-115.54_{\pm0.30}$ | $85.18_{\pm0.62}$ | $30.37_{\pm0.34}$ |
| $(\mathbb{P}^2)^{36}$ | $-108.78_{\pm0.66}$ | $-115.54_{\pm0.70}$ | $85.16_{\pm1.38}$ | $30.38_{\pm0.69}$ |
| $\mathbb{P}^{72}$ | $-109.66_{\pm0.61}$ | $-116.50_{\pm0.68}$ | $87.09_{\pm1.43}$ | $29.42_{\pm0.75}$ |
| $(\mathbb{D}^2)^{12} \times (\mathbb{E}^2)^{12} \times (\mathbb{P}^2)^{12}$ | $-107.02_{\pm1.56}$ | $-115.62_{\pm1.76}$ | $88.52_{\pm8.24}$ | $27.10_{\pm6.48}$ |
| $(\mathbb{D}_1^2)^{12} \times (\mathbb{E}^2)^{12} \times (\mathbb{P}_{-1}^2)^{12}$ | $-108.06_{\pm0.47}$ | $-114.92_{\pm0.39}$ | $83.95_{\pm0.58}$ | $30.97_{\pm0.22}$ |
| $(\mathbb{E}^2)^{12} \times (\mathbb{H}^2)^{12} \times (\mathbb{S}^2)^{12}$ | $-114.85_{\pm0.38}$ | $-120.98_{\pm0.15}$ | $95.12_{\pm0.49}$ | $25.86_{\pm0.40}$ |
| $(\mathbb{E}^2)^{12} \times (\mathbb{H}_{-1}^2)^{12} \times (\mathbb{S}_1^2)^{12}$ | $-110.28_{\pm0.37}$ | $-116.90_{\pm0.42}$ | $87.71_{\pm0.82}$ | $29.19_{\pm0.41}$ |
| $(\mathbb{U}^2)^{36}$ | $-105.98_{\pm0.05}$ | $-112.70_{\pm0.19}$ | $79.85_{\pm0.80}$ | $32.85_{\pm0.61}$ |
| $\mathbb{U}^{72}$ | $-106.58_{\pm0.12}$ | $-113.68_{\pm0.11}$ | $81.53_{\pm0.34}$ | $32.15_{\pm0.36}$ |

Table 13: Summary of results (mean and standard-deviation) with latent space dimension of 6, diagonal covariance parametrization, on the CIFAR dataset. All $nan$ standard deviation values below indicate the repeated experiment was not stable enough to produce a meaningful estimate of spread.

| Model | LL | ELBO | BCE | KL |
|---|---|---|---|---|
| $\mathbb{S}_1^6$ | $-1893.16_{\pm nan}$ | $-1901.32_{\pm nan}$ | $1885.97_{\pm nan}$ | $15.35_{\pm nan}$ |
| $\mathbb{D}_1^6$ | $-1891.69_{\pm nan}$ | $-1897.77_{\pm nan}$ | $1884.12_{\pm nan}$ | $13.64_{\pm nan}$ |
| $\mathbb{E}^6$ | $-1896.19_{\pm2.54}$ | $-1905.75_{\pm3.19}$ | $1889.97_{\pm2.88}$ | $15.78_{\pm0.32}$ |
| $\mathbb{H}_{-1}^6$ | $-1888.23_{\pm2.12}$ | $-1896.56_{\pm2.93}$ | $1882.05_{\pm2.65}$ | $14.51_{\pm0.34}$ |
| $\mathbb{P}_{-1}^6$ | $-1893.27_{\pm0.61}$ | $-1902.67_{\pm0.74}$ | $1887.44_{\pm0.83}$ | $15.23_{\pm0.16}$ |
| $\mathbb{D}^6$ | $-1893.85_{\pm0.36}$ | $-1902.67_{\pm0.69}$ | $1887.37_{\pm0.74}$ | $15.30_{\pm0.08}$ |
| $\mathbb{S}^6$ | $-1889.76_{\pm1.62}$ | $-1897.31_{\pm1.71}$ | $1882.55_{\pm1.48}$ | $14.76_{\pm0.24}$ |
| $\mathbb{P}^6$ | $-1891.40_{\pm2.14}$ | $-1899.68_{\pm2.74}$ | $1884.58_{\pm2.56}$ | $15.10_{\pm0.18}$ |
| $\mathbb{D}^2 \times \mathbb{E}^2 \times \mathbb{P}^2$ | $-1899.90_{\pm4.60}$ | $-1904.63_{\pm1.46}$ | $1889.13_{\pm1.38}$ | $15.50_{\pm0.08}$ |
| $\mathbb{E}^2 \times \mathbb{H}^2 \times \mathbb{S}^2$ | $-1895.46_{\pm0.92}$ | $-1897.57_{\pm0.94}$ | $1882.84_{\pm0.70}$ | $14.73_{\pm0.24}$ |
| $(\mathbb{U}^2)^3$ | $-1895.09_{\pm4.27}$ | $-1904.46_{\pm5.21}$ | $1888.89_{\pm4.71}$ | $15.57_{\pm0.51}$ |

