# OpenReview forum: "Mixed-curvature Variational Autoencoders"
_ICLR.cc/2020/Conference — Accept (Poster)_

### Official Review · AnonReviewer3 · 2019-10-22
**Official Blind Review #3**

**Rating:** 8

**Review:**

This paper introduces a general formulation of the notion of a VAE with a latent space composed by a curved manifold. It follows the current trend of learning representations on curved spaces by proposing a formulation of the latent distributions of the VAE in a variety of fixed-curvature spaces, and introduces an approach to learn the curvature of the space itself. Extensive mathematical derivations are provided, as well as experiments illustrating the impact of various choices of latent manifolds on the performance of the VAE.

I believe this work should be accepted, as while the numerical results are not particularly impressive, it provides some clear foundational work for further exploration of the use of non-euclidean latent spaces in VAEs.

This paper provides extensive and detailed theoretical grounding for their work, ensuring that it is a well-founded extension the VAE formalism. It explores numerous alternatives and compares them, providing detailed experimental results on 4 datasets. The appendices provided a much welcome refreshing on non-euclidean geometry, as well as more details & experimental results.

The paper is already quite dense, especially with the appendices, however there are a few points that could still be detailed in my opinion:

First of all, what were the observation models used for the reconstruction loss in the experiments? I suspect a bernouilli likelhood was used for the binarized dataset, but what about the other ones, and notably CIFAR? Was it a Gaussian observation, a discretized logistic, ...? Was its variance learned? This kind of information is in my opinion crucial for assessing a construction to the latent space of VAE model, as it can have a lot of influence on the kind of information the model will try to store in its latent space.

Secondly, for the model using product of spaces, do you observe some preference of the VAE to store more information in some of the sub-component? This can be explored by comparing the values of the KL term in each of these subspaces.

Third, the VAE with a factorized Gaussian euclidean latent space has a well-known tendency to sparcify its latent representations: unneeded dimensions of the latent space are ignored by the decoder and set to the prior by the encoder. This allows one to not worry too much about the size of the latent space as long as it is "large enough". Does this property remain in curved spaces? Especially in the case the VAE on MNIST with a 72-dimensional latent, as I suspect the 6 and 12 dimensional spaces are not "large enough" for this phenomenon to appear.

**Experience Assessment:**

I have published one or two papers in this area.

**Review Assessment: Checking Correctness Of Derivations And Theory:**

I assessed the sensibility of the derivations and theory.

**Review Assessment: Checking Correctness Of Experiments:**

I assessed the sensibility of the experiments.

**Review Assessment: Thoroughness In Paper Reading:**

I read the paper at least twice and used my best judgement in assessing the paper.

---

> ### Author Response · Authors · 2019-11-13
> **Response to review #3**
>
> First of all, we would like to thank Reviewer #3 for all the time and effort spent on understanding our work. We are happy to see that the mathematical foundations of our work were understood and appreciated.
>
> A point-by-point reply follows below:
>
> - All reconstruction loss terms for all experiments used Bernoulli log-likelihood, except for CIFAR-10 and the Binary Diffusion Process dataset, which used Gaussian log-likelihood with a fixed standard deviation of 1, as did the original authors of the dataset in Mathieu et al. (2019). We have trained CIFAR-10 (images rescaled to values in [-1, 1]) with a Gaussian log-likelihood as above, as well as with a binary cross entropy loss (images rescaled to values in [0, 1]), which loosely corresponds to a “continuous” Bernoulli observation model. Changing the observation model in CIFAR did not have a big impact on log-likelihood comparisons between models, just a small one on sample quality. All of these details are readily available in the accompanying source code. Additionally, we have added this information into the text. For more information on the values of the KL term and reconstruction term of the ELBO in experiments, please see “Extended results” in Appendix E.
>
> - The KL values of components do differ – across different models and even across different runs of the same model. The sum of KL terms behaves stably and similarly to the standard Euclidean VAE. We were not able to establish any meaningful connection between the model preferring a specific type of component (hyperbolic, spherical, or Euclidean) due to the KL being higher in that subspace compared to other component types. This applies even if we fix curvature and do not attempt to learn it. Despite no immediate apparent connections, this might be interesting to investigate more in future work.
>
> - Sparsity does appear and is indeed verifiable in the context of e.g. small 2-dimensional components. Even with fixed or learnable curvatures, this phenomenon does occur in latent spaces which are “big enough” for the given task, as suggested. It does not seem to be significantly more or less occurring than in a standard Euclidean VAE. As for the question if sparsity appears inside the dimensions of a single big component, this is not straightforward to answer, and would need extensive further investigation, because in spaces of non-zero constant curvature, dimensions are correlated.
>
> Thank you for all the feedback,
>
> The authors

---

### Official Review · AnonReviewer1 · 2019-10-23
**Official Blind Review #1**

**Rating:** 8

**Review:**

Summary:
This paper is about developing VAEs in non-Euclidean spaces. Fairly recently, ML researchers have developed non-Euclidean embeddings, initially in hyperbolic space (constant negative curvature), and then in product spaces that have varying curvatures. These ideas were developed for embeddings, and recent attempts have been made to build entire models that operate in non-Euclidean spaces. The authors develop VAEs for the product spaces case.

There's largely two aspects here: one is to be able to write down the equivalents for the operations in models (e.g., the equivalent of adding or multiplying matrices and vectors in Euclidean space have to be lifted to other spaces which no longer have a linear structure). The other are VAE-specific choices, particularly choosing a normal distribution on the manifolds. The authors consider several of these choices and then run a variety of experiments on small latent-dimension cases for VAEs. These reveal that sometimes non-Euclidean and in particular product spaces improve performance.


Strengths, Weakness, Recommendation
I like what the authors are trying to do here; embeddings and discriminative models on non-Euclidean spaces have been developed, offer credible benefits, and generative models are the next step. The authors push forward the machinery needed to do this, and the results seem like there's something there.

On the other hand, the entire work seems quite preliminary. It's hard to say what the takeaway is, or any suggestions for users. The paper is written in a pretty frustrating way. There's an enormous amount of stuff in a sprawling appendix (there are 43 results in the first appendix?!), and checking all of these details will take a great deal of time.

Overall, I recommended weak accept, since a lot of these issues seem like they can be cleaned up.

EDIT: I increased my score based on the authors' response.

Comments:
- The approach taken here is quite similar to another ICLR submission this year, which basically does the same thing but applies these operations to GCNs instead of VAEs.

- A better way to define curvature is just to talk about the sectional curvature, instead of the Gaussian curvature the authors mention at the beginning of section 2. Fortunately for the constant case all of these definitions will be the same.

- It's not quite clear in Section 2.1 why we should care about the fact that you can't fully take K->0 there---why does this hurt anything? You can approximate flat curvature arbitrarily well even without K exactly 0.

- On a similar theme, what's the point of doing the product of {E,S,D,H,P}, instead of just {E,S,H} or {E,D,P}? Seems a bit weird to consider all 5, given the equivalence between S-D and H-P.

- In 2.3, the products of spaces section, the distance decomposition in the 2nd paragraph should have squares (it's an l2): d_M(x,y)^2 = \sum_{i=1}^k d_{M_k_i^n_i)^2(x^i,y^i).

- The discussion in 2.3 should be expanded and made more concrete (some of these you can write out the expressions for), and more pros and cons explained, e.g., which theoretical properties are lost for the wrapped distributions?

- On page 6, I don't understand the first problem with the learnable curvature approach. Why is there no gradient w.r.t to K? Isn't the idea that you'll write this thing as a piecewise function (presumably it's continuous, since that's why the authors built those models that deform to flat), and differentiate the whole thing? Why wouldn't there be a gradient at ELBO(K)? Is it not differentiable at K=0? That doesn't follow directly from just saying the curvature is 0.

-  What's the intuition for the component learning algorithm using 2 dimensions for each of the spaces?

- The experiment section was written in a way where I couldn't understand why the choices being made were there. Why 6 and 12 dimensions here? More clarity here would be great. Also, are there any other models to compare against for these datasets? I'm not a VAE expert; what do other models typically obtain in the authors' regime?


**Experience Assessment:**

I have published one or two papers in this area.

**Review Assessment: Checking Correctness Of Derivations And Theory:**

I assessed the sensibility of the derivations and theory.

**Review Assessment: Checking Correctness Of Experiments:**

I assessed the sensibility of the experiments.

**Review Assessment: Thoroughness In Paper Reading:**

I read the paper at least twice and used my best judgement in assessing the paper.

---

> ### Author Response · Authors · 2019-11-13
> **Response to review #1**
>
> Firstly, we would like to thank Reviewer #1 for all the time and effort invested into understanding and reviewing our work.
>
> We agree that our work is dense and that is why any result that is not directly essential to the concept of a Mixed-curvature VAE like theoretical properties, proofs, or derivations of formulas has been already moved to Appendices A and B. We believe the provided details will not be at the expense of clarity or make verification of results harder, but should on the contrary make reproducibility and verification easier by outlining the necessary steps more clearly. We have shortened the appendices even more.
>
> A point-by-point response to the comments follows below:
>
> - Thanks for pointing out the relevant submission on Graph Convolutional Networks. We have added a mention of this in the related work section of our work. While the underlying geometry is similar to ours, they face several challenges that are different.
>
> - We have updated our definitions of curvature and verified that all following claims still hold.
>
> - You are correct, one can approximate flat spaces (0 curvature) arbitrarily well in any of the mentioned spaces. However, in the hyperboloid and hypersphere (as mentioned at the end of the first paragraph in Section 2.1), the Euclidean norms of points in these spaces go to infinity as $K \to 0$. Therefore, when learning curvature, we wouldn’t be able to change signs of curvature in these spaces. Additionally, the distance and the metric tensors do not converge to their Euclidean variants as $K \to 0$ for these spaces, hence the spaces themselves do not converge to $\mathbb{R}^d$. On the other hand, for the Poincare ball and the projected hypersphere this holds.
>
> - Prior work on product spaces (Gu et al., 2019) considered the manifolds {E, S, H}, mostly due to the favorable optimization properties. However, they never attempted to learn curvature in a sign-agnostic way, which presents a challenge: the norm of points diverges as $K \to 0$. For this reason, we introduce additional manifolds {P, D}. In our experiments, we either use products of {E, S, H} or products of {E, D, P} depending on if we’re learning curvature sign-agnostically or not, because as you correctly pointed out, S is isometric to D and H is isometric to P, in terms of their distance functions. All of this is motivated and explained in Section 2.1 and extensively detailed in Appendix A.
>
> - We have fixed the mistake in the L2 distance decomposition, thank you for spotting this!
>
> (continued below)

---

> > ### Author Response · Authors · 2019-11-13
> > **Response to review #1 (continued)**
> >
> > - We will assume that the reviewer meant Section 2.4, not 2.3, as that one does not deal with wrapped distributions. Essentially, Wrapped Normal distributions are very computationally efficient to sample from and also efficient for computing the log probability of a sample, as detailed by Nagano et al. (2019). The Riemannian Normal distributions (based on geodesic distance in the manifold directly) could also be used, however they are more computationally expensive for sampling, because the only methods available are based on rejection sampling (Mathieu et al., 2019). A detailed description of Wrapped Normal distributions follows Section 2.4.1, and constructions of all the other mentioned approaches (e.g. restriction-based distributions) are detailed in prior work extensively, like Mathieu et al. (2019), Nagano et al. (2019) Davidson et al. (2018), Xu & Durrett (2018), and several other older sources which we cite in our work. A quite comprehensive trade-off discussion is also present in Mathieu et al. (2019), Appendix B.1. We only aim to give an overview and brief motivation for the choice, which we have slightly improved in the text.
> >
> > - As stated at the end of Section 2.4.1, all the operations in the projected constantly curved spaces (Poincare ball and the projected hypersphere) converge to their Euclidean counterparts as $K \to 0$. Therefore, ELBO(K) in these spaces also converges to the classic Gaussian VAE ELBO as $K \to 0$, and hence ELBO(K) can be reformulated as a continuous function w.r.t. $K$. at all points. Differentiability w.r.t. K is straightforward at all points except for $K=0$. At $K=0$, all operations in these spaces are differentiable, which can be verified. Hence ELBO(K) is differentiable, because it is a differentiable composition of differentiable functions. Thank you for pointing out the impreciseness, we have removed the claim from the paper. Fortunately, our method does not strictly depend on differentiability at 0.
> >
> > - Components of dimension 2 are the smallest non-trivial examples of these spaces. They can be easily plotted and inspected. Also, a product space constructed of components of dimension 2 has the most curvature parameters possible overall, because they are one per component. Moreover, as detailed by Tifrea et al. (2019, Section 5), products of H^2 hyperbolic spaces are isometric with the space of Gaussian distributions with diagonal covariance matrices while a single H^n hyperbolic space is isometric to a single Gaussian distribution with a spherical covariance matrix.
> >
> > - The latent space dimensions in the experimental evaluation were chosen as dimensions similar to prior work (Mathieu et al. 2019, Nagano et al. 2019, Davidson et al. (2018)). They mostly used smaller dimensions (e.g. 5, 10, 15). We changed the numbers slightly so that they are divisible by 2 (divisible by the smallest component dimension) and also divisible by 3 (the number of types of components, i.e. {E, S, H} or {E, D, P}). The models H^n, P^n, S^n are all equivalent to prior art models, as stated in the last paragraph of Section 4 before the header “Binary Diffusion Process”. There are also other conceptually differing VAE models that could be used with our product space formulation, as we mentioned in the “Extended Future Work” in Appendix D. We have also attempted to use other models (e.g. Beta-VAE), but we did not observe any significant deviations from the presented results, when compared across different product space curvatures.
> >
> > Thank you for all the feedback,
> >
> > The authors

---

> > > ### Comment · AnonReviewer1 · 2019-11-14
> > > **In response**
> > >
> > > Thank you for answering my questions. The revised paper looks much better.
> > >
> > > - The E/D/P versus E/S/H thing makes sense--- I see that you compare the product with the same dimensions for these two spaces in the experiments. what's the takeaway in general? Does the sign-agnostic aspect significantly help?
> > >
> > > - One more thing: how do we interpret the number in Table 13, where the values often have +/- nan error bars? Should I just assume the whole value is too unstable to interpret?

---

> > > > ### Author Response · Authors · 2019-11-15
> > > > **Response to "In response"**
> > > >
> > > > Thank you for the quick reply and the extensive feedback.
> > > >
> > > > - As can be seen from the experiments, the sign agnostic models (universal component, denoted $\mathbb{U}$) do perform better in higher dimensions than models with fixed signs, which we also mentioned in Section 4, in the second paragraph of the Summary. Another important benefit is that the sign-agnostic models are computationally more efficient, since for a given amount of components in our product space, we do not have to try out an exponential amount of different "sign configurations", but only one that is allowed to dynamically learn the best such sign configuration.
> > > >
> > > > On the other hand, the E/S/H models are less prone to numerical instabilities and are also more efficient to optimize in practice, as noted in [1] and in the second paragraph of Appendix A.2. However, they cannot do sign-agnostic curvature learning (Section 2.1, first paragraph). The same dimensions are comparable across these models in experiments, because they have the same degrees of freedom.
> > > >
> > > > - Nan values: Thank you for spotting this. We have removed Table 14 in the Appendix, as it has never been mentioned in the paper and contained only preliminary results. For the two rows in Table 13 with "nan" standard deviations, we did not obtain meaningful results across multiple runs due to optimization instability. We have clarified this in the table's caption.
> > > >
> > > > Thank you,
> > > >
> > > > The authors
> > > >
> > > >
> > > > [1] Learning Continuous Hierarchies in the Lorentz Model of Hyperbolic Geometry by Nickel & Kiela (ICML 2018)

---

### Official Review · AnonReviewer2 · 2019-10-28
**Official Blind Review #2**

**Rating:** 6

**Review:**

Summary: This paper devised a framework towards modeling probability distributions in products of spaces with constant curvature and showed how to generalize the VAE to learn latent representations on such product spaces using Gaussian-like priors generalized for this case. Empirically the authors evaluate the VAEs on four different datasets (a synthetic tree dataset, binarized MNIST, Omniglot, and CIFAR-10) for various choices of product spaces (fixed curvature and learnable curvature) and choices of latent space dimensionality.

Evaluation:
Overall this seems to be a nice work, with balanced discussion of the empirical results, and is clearly written.
--Past works have considered VAEs on single constant curvature spaces and hence it is well-motivated to consider a more flexible model that enables usage of products of such spaces.
--Empirical evaluations seems fair as far as I can tell, but I am not familiar with benchmarks for VAEs. It was interesting to see the variability in best performing models, e.g. cases in which the mixed curvature models did well vs. the Euclidean one.
--Paper is quite readable, though in a few parts seems to delve a bit unnecessarily into geometric formalism/definitions (e.g. I did not really follow or appreciate the relevance of gyrovector distances).
--Main text is 10 pages long and I'm not sure the extra length is necessary.
--I would have appreciated a more clearly delineated discussion on how the technical details of this work overlap with past papers, both those that have investigated product spaces (Gu et al 2019) and single curvature spaces in VAEs (spherical & hyperbolic)? How did the latter approaches deal with modified prior distributions and/or smoothly recovering the Euclidean K=0 limit? As a result, I'm a bit unsure as to the novelty or technical obstacles that are overcome in the proposed framework in comparison to these.

**Experience Assessment:**

I do not know much about this area.

**Review Assessment: Checking Correctness Of Derivations And Theory:**

I did not assess the derivations or theory.

**Review Assessment: Checking Correctness Of Experiments:**

I assessed the sensibility of the experiments.

**Review Assessment: Thoroughness In Paper Reading:**

I read the paper at least twice and used my best judgement in assessing the paper.

---

> ### Author Response · Authors · 2019-11-13
> **Response to review #2**
>
> Firstly, we would like to thank Reviewer #2 for all the time and effort invested into understanding and reviewing our work.
>
> Below follows a point-by-point response:
>
> - To the best of our knowledge, we used several standard benchmark datasets for evaluating VAE models. We have attempted to use other models (e.g. Beta-VAE), but we did not observe any significant deviations from the presented results, when compared across different product space curvatures.
>
> - Gyrovector spaces are a key step towards defining VAEs in a unified framework across differently curved spaces (Poincare ball, Euclidean space, Projected spherical space), which then leads to being able to learn the curvature of such a space irrespective of the sign.
>
> - We have attempted to shorten the geometric background as much as possible while not sacrificing understandability by moving a lot of important definitions and properties required for reproducing the theoretical and empirical results of this paper into Appendices A and B already.
>
> - The only similarity to the paper of Gu et al. is the motivation and use of product spaces as opposed to single component spaces. They focus on graph embeddings, whereas we attempt to learn VAEs in these spaces. Gu et al. additionally only use the spaces {E, S, H} and learn curvature only with a fixed sign.
> As far as we are aware, none of the single-component prior works (hyperbolic/spherical/Euclidean VAEs) ever attempted to learn curvature as a parameter of the model directly. Hence, they did not have to obtain derivations of the Wrapped Normal distribution’s log-likelihood or even the operations in the space for different values of curvature K. Several of the formulas simplify significantly if we assume a fixed $K=1$ or $K=-1$. One notable exception to this is the Poincare VAE approach of Mathieu et al. (2019), where they derive the necessary formulas and attempt to train several $\mathbb{P}_c^n$ VAEs with different values of $c \in [0.1, 1.4]$, always fixed during the entire process of training and evaluation. Therefore, all the operations and log-likelihoods were derived from scratch for {S, H, D}, proven (see Appendix A and B), and checked by comparing to prior work formulas by substituting $K=1$ (or $-1$) as an additional verification step.
>
> Thank you for all the feedback,
>
> The authors

---

### Decision · Program_Chairs · 2019-12-19

**Decision:**

Accept (Poster)

**Comment:**

This paper studies generalizations of Variational Autoencoders to Non-Euclidean domains, modeled as products of constant curvature Riemannian manifolds. The framework allows to simultaneously learn the latent representations as well as the curvature of the latent domain.

Reviewers were unanimous at highlighting the significance of this work at developing non-Euclidean tools for generative modeling. Despite the somewhat preliminary nature of the empirical evaluation, there was consensus that the paper puts forward interesting tools that might spark future research in this direction. Given those positive assessments, the AC recommends acceptance.